# Self-preservation strategy for approaching global warming targets in the post-Paris Agreement era

Yi-Ming Wei [1,2,3✉], Rong Han[1,3], Ce Wang [1,2], Biying Yu [1,2,3✉], Qiao-Mei Liang [1,2,3✉], Xiao-Chen Yuan[1,2,3✉], Junjie Chang[1], Qingyu Zhao[2,3], Hua Liao[1,2,3], Baojun Tang[1,2,3], Jinyue Yan [4], Lijing Cheng [5] & Zili Yang[6]

A strategy that informs on countries' potential losses due to lack of climate action may facilitate global climate governance. Here, we quantify a distribution of mitigation effort whereby each country is economically better off than under current climate pledges. This effort-sharing optimizing approach applied to a 1.5 °C and 2 °C global warming threshold suggests self-preservation emissions trajectories to inform NDCs enhancement and long-term strategies. Results show that following the current emissions reduction efforts, the whole world would experience a washout of benefit, amounting to almost 126.68–616.12 trillion dollars until 2100 compared to 1.5 °C or well below 2 °C commensurate action. If countries are even unable to implement their current NDCs, the whole world would lose more benefit, almost 149.78–791.98 trillion dollars until 2100. On the contrary, all countries will be able to have a significant positive cumulative net income before 2100 if they follow the self-preservation strategy.

[1] Center for Energy and Environmental Policy Research, Beijing Institute of Technology, Beijing 100081, China. [2] School of Management and Economics, Beijing Institute of Technology, Beijing 100081, China. [3] Beijing Key Lab of Energy Economics and Environmental Management, Beijing 100081, China. [4] Energy Process Division, Royal Institute of Technology, SE-10044 Stockholm, Sweden. [5] International Center for Climate and Environment Sciences, Institute of Atmospheric Physics, Chinese Academy of Sciences, Beijing 100029, China. [6] Department of Economics, State University of New York at Binghamton, Binghamton, NY 13902-6000, USA. ✉email: wei@bit.edu.cn; yubiyingjapan@yahoo.co.jp; liangqiaomei@bit.edu.cn; yuanxc@bit.edu.cn

To facilitate global climate governance, Paris Agreement requires the ratified parties to update their nationally determined contributions (NDCs) every 5 years[1]. However, the recent 24th Conference of Parties in Katowice, Poland (COP24)[2] and 25th Conference of Parties in Madrid, Spain (COP25) ended with limited progress. According to the IPCC Special Report on the impacts of global warming of 1.5 °C, global temperatures are likely to reach 1.5 °C between 2030 and 2052, which would cause dramatic damage[3]. Early and quick action will provide better chance to close the widening emissions gap[4], even though a large amount of abatement cost would occur in the short term. Inaction to climate change will lead to substantial socioeconomic losses, implying the occurrence of a broader cost than sufficient action. In this sense, providing information for countries about their economic losses due to insufficient action against climate change and check if they had net income (avoided climate damage minus abatement cost) when they achieve the 1.5 °C or well below 2 °C target would be helpful for countries to make a self-preservation decision.

Prior literature has proposed global or national strategies for climate change mitigation. Some studies focused on the emissions gap between the NDCs and emission scenarios consistent with the global warming thresholds (1.5 °C or well below 2 °C target)[4,5]. As current efforts are insufficient to meet the warming targets, some scholars have modeled equity principles to allocate the emissions gap across countries from a top–down perspective[6–12]. Since the emission allocations in most of these studies are not always cost-optimal[8], several studies allocated the global cost optimal emissions consistent with the 1.5 °C goal to countries or regions following equity approaches[8,13–17]. However, they only considered the global emission abatement cost and mostly ignore the potential benefits of avoiding the climate damage. In contrast to the effort-sharing studies, other studies have considered not only mitigation costs but also the benefit of mitigation (i.e., avoided climate impacts) to find the optimal emission mitigation pathways by optimizing global or regional social welfare[18–23]. Although this group of research indicates that countries may have potential benefits attributing to their efforts on limiting climate warming, their derived strategies do not always consider the global warming thresholds and the equity. Therefore, what is lack to reach the warming targets is a beneficial strategy that can balance the long-term benefits obtained by climate mitigation and the short-term abatement costs for each country, and take into account the equitable effort sharing. Thus, we present a farsighted self-preservation strategy, contributing to straightforward benefits that countries would otherwise lose by inaction or insufficient action, compared to 1.5 °C or 2 °C commensurate action.

Normally, the cost and benefit are strongly determined by the progress of technological development and the degree of climate damage. Recent studies have indicated that the climate damage could be much higher than previously estimated;[24–27] thus, climate mitigation (avoided climate loss) could be extremely beneficial. Additionally, if the low-carbon technologies (such as carbon capture and storage, renewable energy utilization, and negative emissions technologies) could be rapidly developed, it will result in a lower cost for emissions reduction, which will make countries more capable in mitigating climate change[3,28]. Resulting from the uncertainties of climate damage and technology development, we simulated a global cooperative situation to obtain the optimal emission trajectories toward the 1.5 °C or well below 2 °C target. Under this framework, the self-preservation strategy indicates that the temperature-limiting goals could be reached along with net income, as compared to the current reduction efforts (i.e., policy-as-usual situation). NDC is regarded as the policy-as-usual effort for countries with ambitious NDCs; while for countries that actually do more than they have committed in the NDCs, the business-as-usual (BaU) situation is regarded as policy-as-usual effort (see Methods section). Net income means that the cumulative benefits from the extra avoided climate impacts should exceed the extra mitigation costs, compared to the current climate polices at both the global and national levels. The break-even point between mitigation costs and benefits for each country can be further identified.

China's Climate Change Integrated Assessment Model (C[3]IAM)[29] was used in this study to explore the self-preservation strategy for each country with the uncertainties of climate damage and low-carbon technologies (see Methods section). To take into account the equity among regions, we introduce the effort-sharing approaches to determine the social welfare weights that can represent the relative importance in the utility and the relative mitigation burden. An integrated social welfare weight indicator is constructed for each region by combining the estimated social welfare weights obtained from the existing mainstream effort-sharing principles, including responsibility (grandfathering and historical responsibility) defined by multiple entities, such as developing countries and developed countries, capability (ability to pay) to assign more affluent countries with more efforts, and equality (equal per capita allocation) to ensure each region's equitable burden sharing in response to climate mitigation. And then the integrated social welfare weight is used in the global welfare maximization function to improve the equity of allocation results in the cost-benefit analysis. The optimal emission pathways for each region will then be determined under its given integrated social welfare weight, and its own climate damage and abatement cost functions through the C[3]IAM. As the existing NDCs are ambiguous in terms of their definition and coverage[30], a uniform accounting criterion for NDCs is developed and the policy-as-usual pathways are constructed. By comparing the benefits and costs between optimal emission pathways and policy-as-usual pathways, the optimal emission trajectories were derived that could realize the warming targets and bring net incomes to every region. In C[3]IAM, we implement the effort-sharing and cost-benefit analysis at regional level (in total 12 regions); and then the allocation results are further downscaled to the country level to inform the national actions.

Overall, this study presents a better emission reduction strategy than current NDCs in terms of the potential net income from climate mitigation. Results informs on an economically effective action for countries to update their NDCs in the post-Paris Agreement era.

## Results

**Self-preservation strategies identification.** The scenario setting considers four aspects, including warming thresholds, low-carbon technology costs, climate damage, and equity principles. With respect to warming, we focus on the average atmospheric temperature change in 2100. The warming thresholds in this study are in line with the Paris Agreement. If the rise in temperature in 2100 is <2 °C and if every region gain compared to the policy-as-usual scenario, the optimal emission scenario is regarded as a self-preservation scenario for 2 °C, and similar for 1.5 °C. Overshoots are allowed for the self-preservation scenario. Meanwhile, different levels of low-carbon technology costs and climate damage have been considered to reflect the uncertainty of low-carbon technology development and climate risk. According to the uncertainty level of climate damage and technology development, there is a package of specific optimal emission scenarios (Fig. 1a). The changes in low-carbon technology costs and climate damage are set in accordance with previous studies[20,21,27,31]. We define the level of climate damage by using the ratio of economic

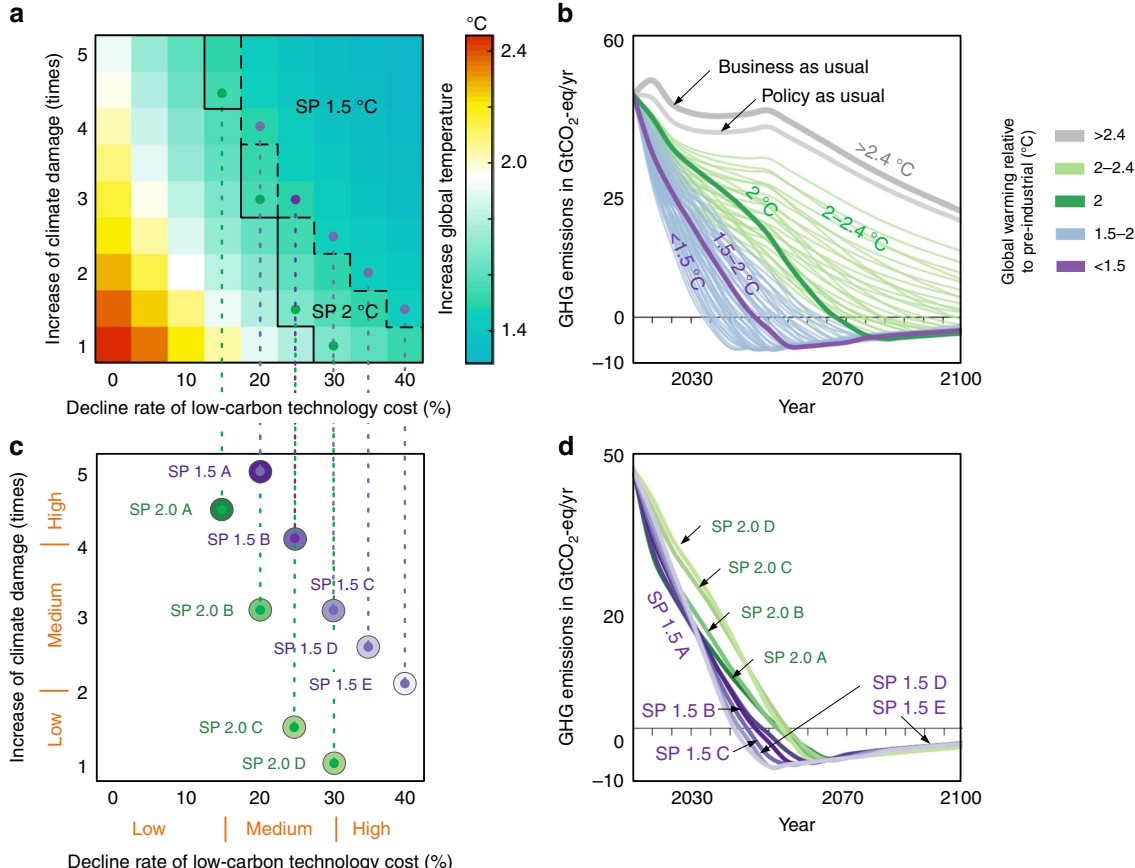

**Fig. 1 Deriving the representative self-preservation strategies. a** Changes in temperature by 2100 under various scenarios; each cell in this matrix represents a scenario under a certain level of low-carbon technology development and climate damage; the boundary line in **a** represents a self-preservation scenario consistent with 1.5 °C or 2 °C target, respectively. For the vertical axis, following the literature[20], we define the level with climate damage to be ~1.6% of the global GDP at a 2.62 °C warming in 2100 as the reference level of climate damage (i.e., 1). Please see the reference level in Supplementary Fig. 1. The increase in climate damage (times) means the times of climate damage coefficients used in the damage function compared to the reference level of climate damage for the given temperature rise (see Supplementary Fig. 1 for the exact economic damage for each region). For the horizontal axis, 0–40% means the decline rate of low-carbon technology cost every 5 years, and the reference level (i.e., 0) means the decline rate of low-carbon technology cost keeps constant as the base year 2015. **b** Global GHG emissions pathways and corresponding changes in temperature under the scenarios in **a**, plus two reference paths (business-as-usual (BaU) and policy-as-usual). BaU scenario describes the situation without losses caused by climate change and the NDCs. Policy-as-usual scenario is constructed based on current NDCs and assumes the efforts of NDCs will continue (see Methods section). **c** The representative self-preservation scenarios consistent with the 2 °C or 1.5 °C target. **d** Global GHG emissions pathways of the self-preservation scenarios in **c**. The names of scenarios are explained as: SP means self-preservation, 2.0 or 1.5 means the temperature-limiting goal. The starting point of the horizontal axis in **b**, **d** is the year 2015. Colors from dark to light green and purple represent SP 2.0 scenarios from A to D and SP 1.5 scenarios from A to E, respectively.

damage in GDP. The enlarged coefficients (increase times of climate damage shown in Fig. 1a) of damage function in the model are used to characterize different levels of climate damage. The values reported in the Nordhaus' research[20] are set as the reference level (1 in Fig. 1a, c), which show climate damage to be 1.6% of GDP at a 2.62 °C warming in 2100. Given the temperature rise, the economic damages are assumed to be different times as large as the reference level. Supplementary Fig. 1 shows the changes in economic damage under different uncertainties with temperature rise for each region. In this study, we define high, medium and low level of climate damage, which corresponds to the increase times of climate damage coefficients used in the damage function being four to five times, two to four times, and less than two times, respectively. In addition, we define three levels of technological development, that is the slow development with low decline rate of low-carbon technology costs being <15% every 5 years, medium development with medium decline rate of low-carbon technology costs being 15–30% every 5 years, and

rapid development with high declining rate of low-carbon technology costs being 30–40% every 5 years. The capability and responsibility of developing and vulnerable countries are fully considered by introducing equity principles to determine their social welfare weights in the global welfare function. The integrated social welfare weight of each region is shown in Supplementary Table 1.

In all the optimal emission scenarios (Fig. 1a), the percentage of self-preservation scenarios consistent with the well below 2 °C and 1.5 °C targets could reach 51.9%, a majority of that are under conditions of medium to high rate of decline of low-carbon technology costs. This implies that if society were to experience a medium to rapid technological development (the decline rate every 5 years of the low-carbon technology costs can reach 15% or more), a self-preservation strategy with straightforward benefits could always be found. However, only scenarios under the conditions of >1.5 times the climate damage (relative to the reference level) and >20% decline in low-carbon technology costs,

which account for 35.8% in the all optimal emissions scenarios, can achieve the 1.5 °C target. Figure 1b shows a highly diverse greenhouse gas (GHG) emissions trajectory, where GHG emissions of the optimal scenarios that range from −3.39 to 13.95 GtCO$_2$-eq and temperature increases that range from 1.3 to 2.5 °C are likely to occur in 2100. The pathway under the policy-as-usual scenario shows that the current NDCs are not in line with the well below 2 °C target.

We selected nine representative self-preservation scenarios that have the highest welfares under each level of low-carbon technology cost for further analysis; of the nine, four would reach the well below 2 °C target under different levels of climate damage (namely SP 2.0 s, including SP 2.0 A, SP 2.0 B, SP 2.0 C, and SP 2.0 D) and five would reach the 1.5 °C target (named SP 1.5 s, including SP 1.5 A, SP 1.5 B, SP 1.5 C, SP 1.5 D, and SP 1.5 E; Fig. 1a, c). The GHG emissions in the four 2 °C scenarios would rapidly drop after 2035, and net zero or negative emissions would be obtained ~2055 (indicted using green lines in Fig. 1d). In contrast, the 1.5 °C scenarios need a sharp decline in GHG emissions from now so that negative emissions can be obtained ~2045–2050 (indicated by purple lines in Fig. 1d).

**Cost and benefits of self-preservation strategies**. Figure 2a and Fig. 3a suggest that, as compared to the current reduction efforts, the global cumulative benefits would outweigh the additional costs before 2100. On average, there will be 336.0 trillion dollars and 422.1 trillion dollars of accumulated net income if the 2 °C target and 1.5 °C target is realized until 2100 (constant prices in 2011, purchasing power parity method; hereafter, all monetary amounts use the same constant prices). This net income would be obtained after 2065 and 2070 under SP 1.5 s and SP 2.0 s, respectively. Meanwhile, all regions and countries can achieve a positive cumulative net income by 2100 (Fig. 2b, c and Fig. 3b, c). Specifically, all regions and countries except the USA, the Russia Federation, Japan, the EU, other branches of umbrella group (OBU) countries, and the Eastern European and Commonwealth of Independent States (EES) countries can achieve a positive cumulative net income before 2080 and before 2070 if the 2 °C and 1.5 °C target is met, respectively (Fig. 2c, Fig. 3c, Supplementary Fig. 2). Additionally, positive cumulative net income would be obtained earlier under the SP 1.5 s than under the SP 2.0 s (Supplementary Fig. 2). If the warming limit targets are realized, then Indonesia, China, the EU, India, and Nigeria will have a larger cumulative net income (on average, 37.2 trillion dollars until 2100) than the global average (2.5 trillion dollars) in the case of the 2 °C target. India, Nigeria, China, the EU, Indonesia, and the USA will have a larger cumulative net income (on average, 39.9 trillion dollars until 2100) than the global average (3.2 trillion dollars) after achieving the 1.5 °C target. The negative net income for some countries at an early stage is <0.57% of the national annual GDP. Vulnerable countries like Colombia, Venezuela, Algeria, and Ethiopia could also reach break-even point between 2030 and 2070 with 1.23–2.75, 0.87–1.95, 1.55–3.79, and 3.36–8.21 trillion dollars of cumulative net income in 2100 to achieve the 1.5 °C target, respectively (shown in Supplementary Fig. 2).

However, to achieve the break-even point, upfront investment is needed. We estimated the cumulative mitigation costs ahead of break-even point by comparing the emissions reduction costs implied by self-preservation strategy with that of policy-as-usual scenario. The resulting global upfront investment amounts to ~18.12–113.70 trillion dollars for achieving global temperature-limiting targets, of which G20 Economies are responsible for the largest share (up to 91%). Figure 4 presents the break-even dates and amount of upfront investments ahead of break-even point of

G20 Economies and selected vulnerable countries following self-preservation strategy. Results show that the timing of break-even points of the USA, Russian Federation, Canada, and Australia will occur in the end of this century, and before 2035 for South Africa and Saudi Arabia. Developing economies in G20 (i.e., China, Brazil, Mexico, Indonesia, Turkey, Argentina, India, and Saudi Arabia) need ~4.73–30.66 trillion dollars of financial investment before turning into profits. Regarding the vulnerable countries, for example, Colombia, Venezuela, Algeria, and Ethiopia, the average upfront investment ranges from 48.62 to 352.61 billion dollars for achieving warming targets. The above results of upfront investment are able to inform the maximum financial transfers across countries to some extent.

Though the large amount of upfront investments before break-even points are required for approaching the global warming targets, if only following the current reduction efforts (Fig. 5a, b), the whole world would experience a washout of benefit, which is estimated to be as high as 126.68–616.12 trillion dollars and 264.11–610.16 trillion dollars until 2100, as compared to well below 2 °C and 1.5 °C commensurate action, respectively, and ~1.21–5.86 times and 2.51–5.80 times of global GDP in 2015, respectively. Therein, India and the Middle East and Africa (MAF) will have lager washout benefits compared to their own current national GDP. What's worse, if even the current NDCs cannot be fully achieved (e.g., the USA quit from the Paris Agreement), the whole world would tend to lose out on more benefit, ranging between 149.78 and 791.98 trillion dollars until 2100, which is ~1.42–7.53 times of the current global GDP (2015; Fig. 5c, d).

**Economically effective actions in the post-Paris agreement era**. Compared to the current NDCs, the self-preservation scenarios require countries worldwide to cut more GHG emissions in 2030, with an extra global reduction of 19.16–29.14 GtCO$_2$-eq to achieve the 2 °C target and 28.21–29.75 GtCO$_2$-eq reduction to achieve the 1.5 °C target (Fig. 6a). To achieve the 2 °C target, emission gaps are smaller at the early stages under the situation of lower climate damage (SP 2.0 C and SP 2.0 D). At the national level, countries except those in the other developed countries in Western Europe (OWE), MAF, and Asia are required to mitigate more GHG emissions than their current NDCs for the 2 °C target (Fig. 6b, c, Supplementary Figs. 3 and 5). To reach the 1.5 °C target, all countries need to further reduce their emissions in 2030 compared with the current NDCs level. Much more improvements need to be made by countries, including Japan (on average 101% extra emissions reduction required for SP 1.5 s), USA (93% extra), Russia Federation (85% extra), the EU (72% extra), China (65% extra), and OBU (63% extra; Fig. 6b, Supplementary Figs. 4 and 6). The average GHG emissions of China, the USA, the EU, RUS, and Japan need to become negative before mid-century under both SP 2.0 and SP 1.5 scenarios. India's average GHG emissions need to become negative before 2065 for achieving 2 °C target, which is almost 10 years later compared with the timing for 1.5 °C target. Among these major emitters, the timing of net-zero emissions of the USA and Japan (2035–2040) is 10 years earlier than China (2045–2050), and 23 years earlier than India (2060–2065) for achieving 2 °C target. The gap of net-zero points between these countries has narrowed for 1.5 °C target (Table 1). The emission reduction actions required for each country in all self-preservation scenarios for achieving the 2 °C and 1.5 °C targets are given in Supplementary Tables 2–3, respectively, which can serve as an economically effective action strategy in the post-Paris Agreement era.

To realize the self-preservation strategy, effective policies are required. The marginal abatement cost (MAC) is an important

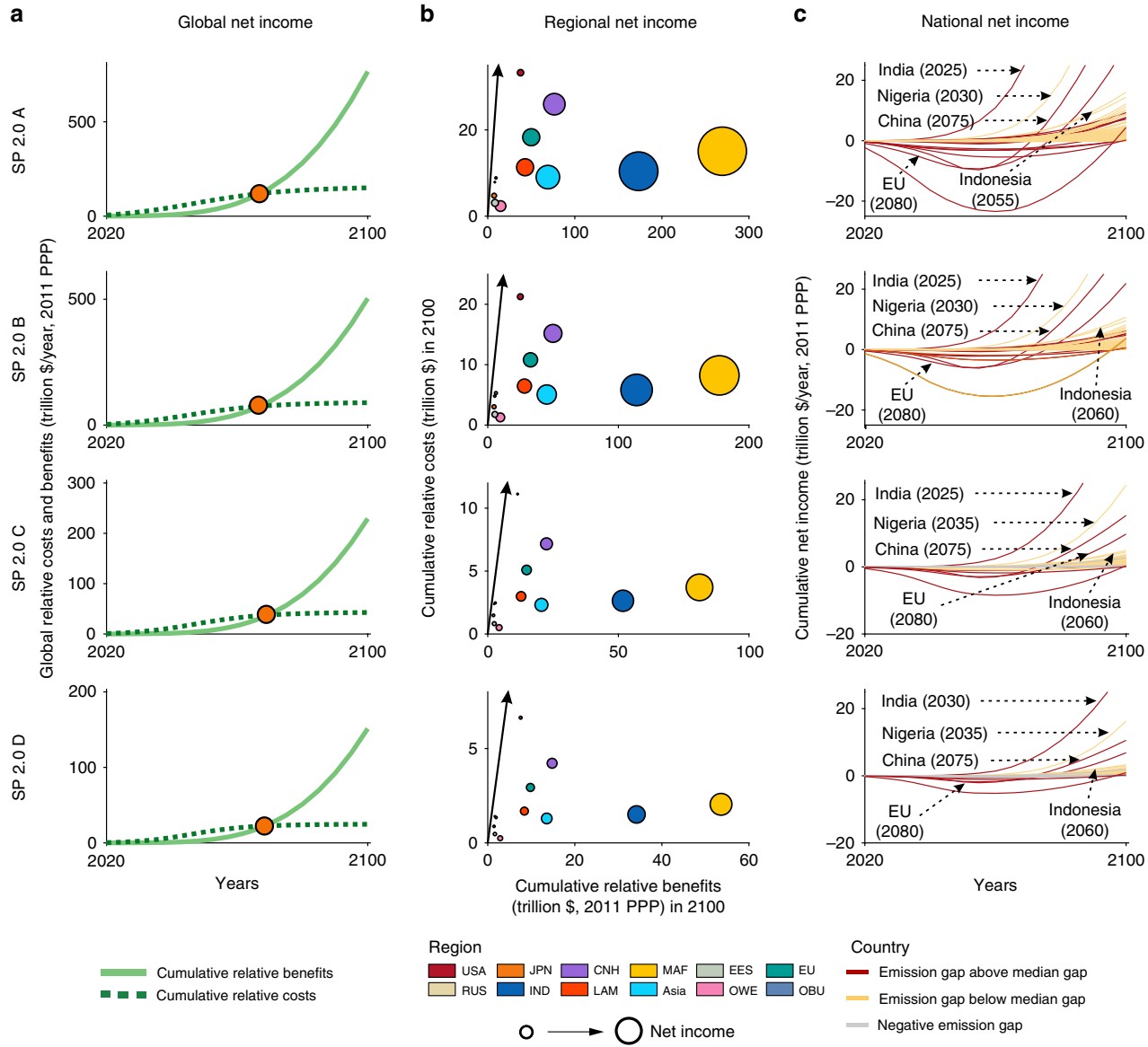

**Fig. 2 Net income between policy-as-usual scenario following the current reduction efforts and SP 2.0 s at the global, regional, and national levels.**
**a** Global cumulative relative costs and benefits under 2 °C target. The circles indicate the turning point where the benefits exceed the costs. The cumulative relative benefits (light green line) mean the avoided cost of climate impact. Cumulative relative costs (dark green dotted line) mean the cost of climate mitigation. **b** Regional cumulative relative benefits and costs in 2100 under 2 °C target. The black line indicates that the benefits and costs are equal. Countries or regions on the right side of the black line have positive net income. The size of the bubble refers to regions' cumulative net income in 2100. Different colors represent different regions. USA the United States, CHN China, JPN Japan, IND India, EU the European Union, Asia Asia excluding China, India, and Japan, RUS Russia Federation, MAF the Middle East and Africa, EES Eastern European and Commonwealth of Independent States countries (except the Russian Federation), LAM Latin America, OBU other branches of umbrella group, i.e., Canada, Australia, and New Zealand, OWE other developed countries in Western Europe. **c** National net income from 2020 to 2100 under 2 °C target. Unit, trillion dollars per year. The emission gap in **c** means the difference in the GHG emissions between current NDCs and self-preservation scenarios. The positive emissions gap indicates the further required GHG emissions reduction. Numbers in parentheses refer to year of break-even points.

factor that can influence the stringency of climate change policy. In order to improve the feasibility, the stringency of climate change policies should be consistent with the corresponding MAC, as shown in Fig. 7. Compared with other studies containing MAC analysis, our results are within the existing interval. Moreover, higher marginal costs do not necessarily imply higher total policy costs[32]. Thus, the self-preservation strategies are feasible from this point of view. From the time perspective, all regions need to start by tightening their policies year by year at the early stage. At a later stage, most of them can relax their strict policies. The timings for policy relaxation differ

among regions. Specifically, for the 2 °C target, Japan, the USA, Russia Federation, OBU, and the EU can relax their stringency before 2040; OWE, China, EES countries except the Russian Federation, and Latin America (LAM) need to continuously tighten their policy until 2045–2050. India, Asia, and the MAF need to continuously strengthen their policies at least till 2060–2065. To achieve the 1.5 °C target, all regions need to increase their policy stringency much faster than they would under the 2 °C target in the early stage. The substantial difference in MAC across regions is in accordance to the different mitigation efforts of regions, implying the necessity of

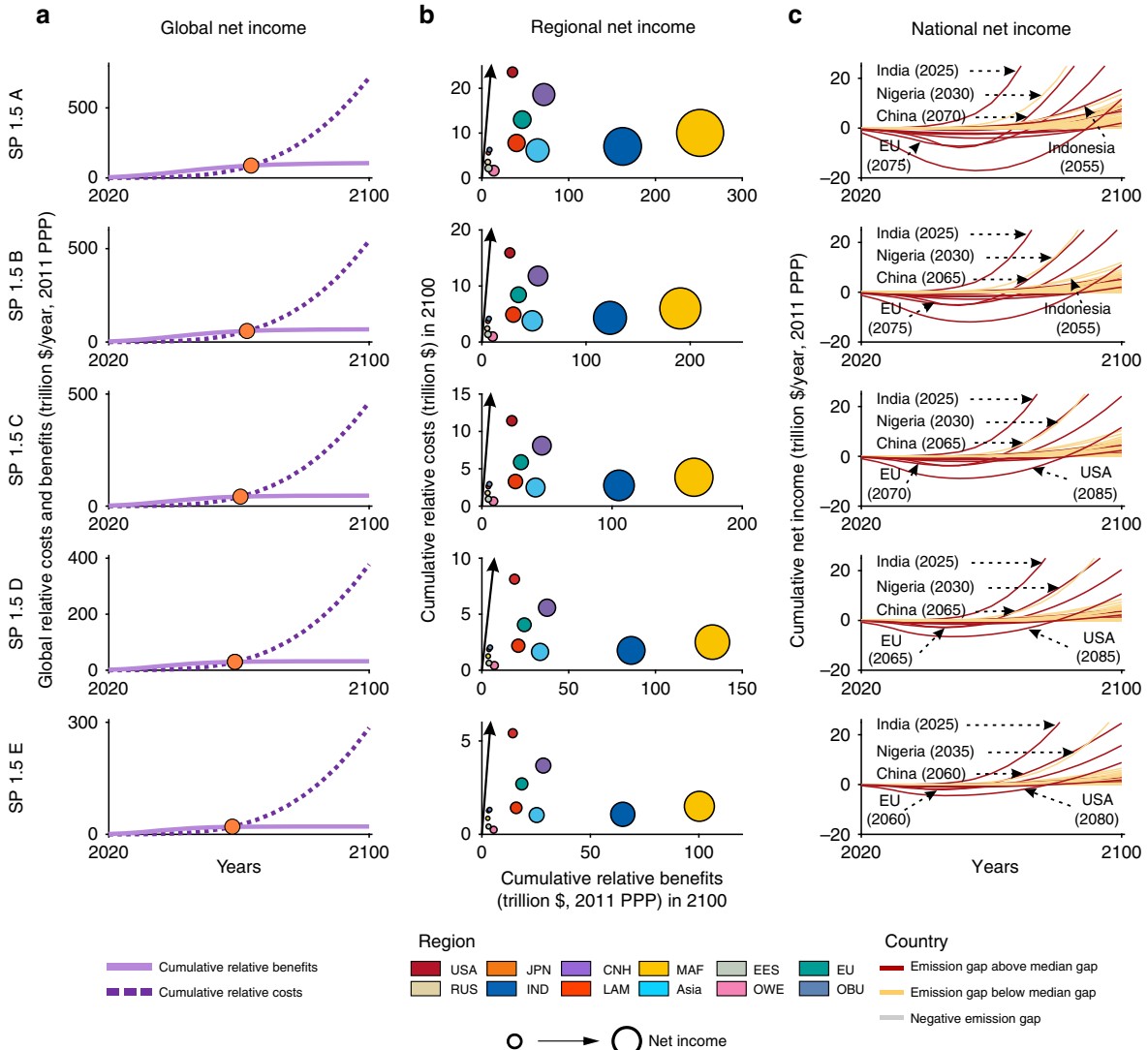

**Fig. 3 Net income between policy-as-usual scenario following the current reduction efforts and SP 1.5 s at the global, regional, and national levels.**
**a** Global cumulative relative costs and benefits under 1.5 °C target. The circles indicate the turning point where the benefits exceed the costs. The cumulative relative benefits (light purple line) mean the avoided cost of climate impact. Cumulative relative costs (dark purple dotted line) mean the cost of climate mitigation. **b** Regional cumulative relative benefits and costs in 2100 under 1.5 °C target. The black line indicates that the benefits and costs are equal. Countries or regions on the right side of the black line have positive net income. The size of the bubble refers to regions' cumulative net income in 2100. Different colors represent different regions. **c** National net income from 2020 to 2100 under 1.5 °C target. Unit, trillion dollars per year. The emission gap in **c** means the difference in the GHG emissions between current NDCs and self-preservation scenario. The positive emissions gap indicates the further required GHG emissions reduction. Numbers in parentheses refer to year of break-even points.

establishing international emissions trading scheme, in order to reduce the total abatement cost.

## Discussions
The post-Paris Agreement climate governance requires countries to undertake greater emission mitigation in the subsequent rounds of pledging for achieving warming targets[33]. Thus, our study presents self-preservation strategies in order to improve current emissions reduction efforts to achieve a well below 2 °C or 1.5 °C target, while highlighting the self-inflicted losses that countries commit to by not enhancing their NDC sufficiently. We found that all countries would gain by raising their target and by aligning with the 1.5 °C or well below 2 °C objective. Even when society experiences a relatively slow development of low-carbon technologies, a self-preservation strategy could still be found. Under conditions of higher climate damage and rapid low-carbon technology development, there will be larger benefits by limiting

global warming in all countries. Even the relatively vulnerable countries, which are mainly located in the MAF and LAM, would have a cumulative net income in 2100. As for the well below 2 °C target, countries in the MAF could reach break-even point between 2030 and 2035, with a cumulative net income of 0.84–4.17 trillion dollars in 2100, which is 0.63–3.12% of the cumulative GDP. Furthermore, countries in the LAM could reach break-even point between 2070 and 2075, with 0.25–1.17 trillion dollars of cumulative net income in 2100, equivalent to 0.26–1.24% of GDP. The break-even point could be reached earlier and they could have more benefits by achieving 1.5 °C target, which is 2030 (1.62–3.96 trillion dollars) and 2060–2065 (0.53 to 1.19 trillion dollars) for the MAF and LAM, respectively. For achieving the well below 2 °C target, most countries need to improve their current NDCs modestly. For achieving the 1.5 °C target, it requires extra 28–30 GtCO₂-eq GHG reductions in 2030, globally. Each country has to enhance its current efforts

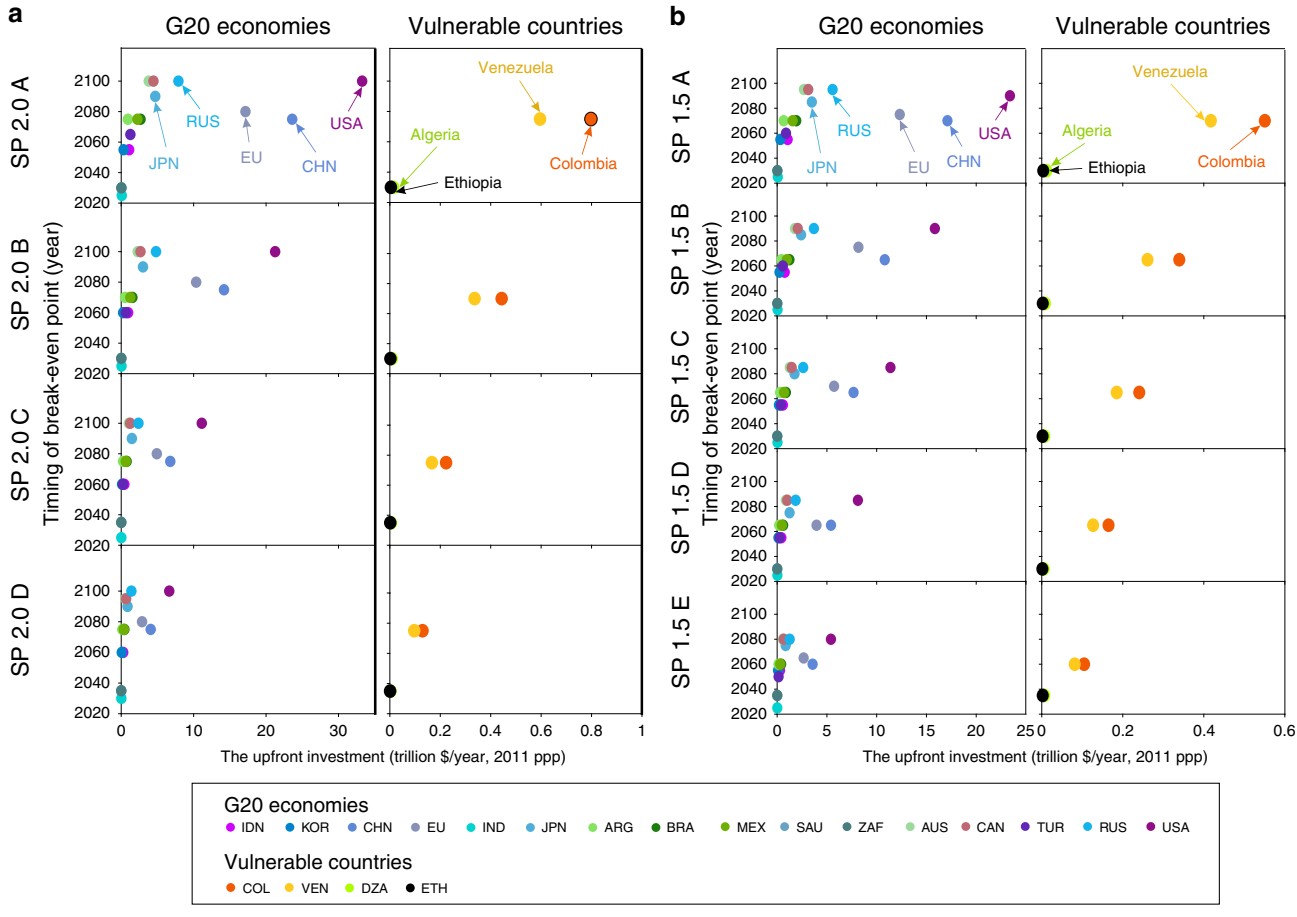

**Fig. 4 The upfront investment and timing of break-even points for G20 Economies and selected vulnerable countries following self-preservation strategy. a** The upfront investment and timing of break-even points for G20 and selected vulnerable countries under 2.0 °C target. **b** The upfront investment and timing of break-even points for G20 and selected vulnerable countries under 1.5 °C target. Different colors represent different countries. G20 Economies: IDN Indonesia, KOR the Republic of Korea, CHN China, EU the European Union, IND India, JPN Japan, ARG Argentina, BRA Brazil, MEX Mexico, SAU Saudi Arabia, ZAF South Africa, AUS Australia, CAN Canada, TUR Turkey, RUS the Russian Federation, USA the United States. Selected vulnerable countries: COL Colombia, VEN Venezuela, ETH Ethiopia. Since the other four G20 members, i.e., the United Kingdom, France, Germany, and Italy belong to the EU, the related information does not display, respectively. The lower demand of investment support for achieving 1.5 °C than that for 2 °C is attributing to the earlier break-even points and faster decline of technology cost.

significantly. Moreover, they need to accelerate technology upgradation to achieve rapid emission reductions immediately. Japan, the USA, Russia, the EU, China, and the OBU have to make greater efforts for achieving the 1.5 °C target.

Though our self-preservation strategies are able to achieve the temperature-limiting goals and are able to obtain the cumulative positive net income for all countries before 2100, as compare to the current emissions reduction efforts, many countries and regions would have a negative net income in the early stage due to the large amount of GHG abatement cost. Our analysis indicates that, the upfront investment before break-even points of G20 Economies is ~16.38–103.53 trillion dollars for achieving the temperature-limiting targets. In particular, the USA has to invest 5.41–33.27 trillion dollars. For Canada and Australia, the upfront investment is also relatively higher than other G20 Economies. And the break-even points for the USA, Canada, and Australia will occur in the end of this century. This is a severe obstacle in implementing the proposed self-preservation strategies in the real world. Some countries or regions may refuse to accept the enhanced strategy in the near term and choose to neglect the long-term climate damage. However, our results show that the amount of the negative net income is <0.57% of annual GDP for each country or region on average. Therefore, to avoid the

threat of climate damage, all countries in the world are encouraged to adopt the climate mitigation actions following our self-preservation strategy, which would allow them to reach 0.46–5.24% GDP gains in 2100.

Most importantly, implementing such a self-preservation strategy in a real word requires countries to recognize the gravity of global warming and to make breakthroughs in low-carbon technologies. Financial and technical support from developed countries is necessary for relatively vulnerable countries to implement the self-preservation strategy. In order to determine how countries should cooperate, we assume an approach that takes into account the equitable effort sharing for emission reduction. However, it leads to some non-major emitters bearing large burdens and some countries may not be acting early enough to avoid climate change, despite their interest to do so[34]. Therefore, we should recognize the special vulnerability of countries and prioritize them to receive technical and financial support, which need further analysis on how to implement it in the practice. Our study here can contribute to identifying the ceiling costs of self-preservation strategies for each country. For regions like the MAF and LAM, the upfront investment before break-even points are 1.35–9.77 and 0.06–0.31 trillion dollars for keeping below the warming thresholds, respectively. They need

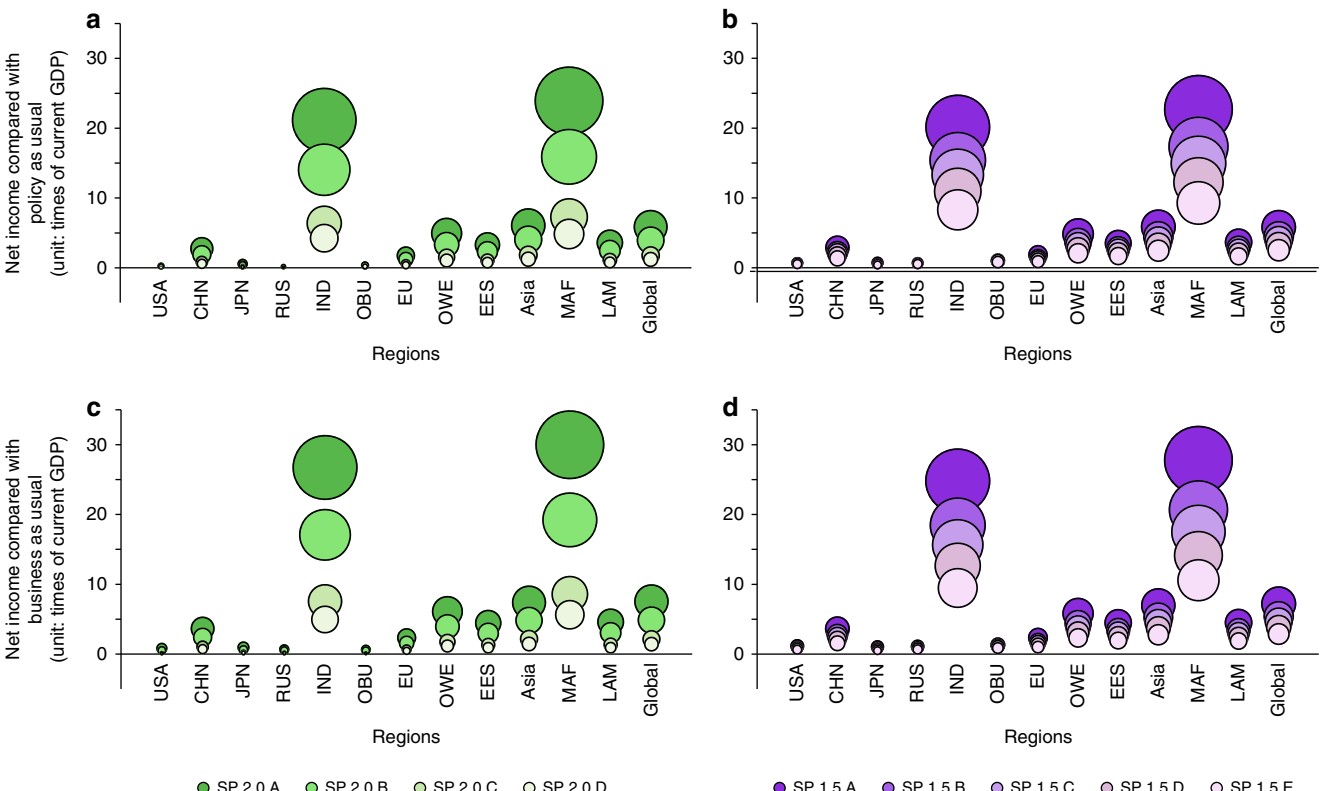

**Fig. 5 Net income of self-preservation strategies at the regional levels (unit: times of the regional or national GDP in 2015). a** Net income following the current reduction efforts (policy-as-usual scenario) of 2 °C. **b** Net income following the current reduction efforts (policy-as-usual scenario) of 1.5 °C. **c** Net income following the business-as-usual (BaU) efforts of 2 °C. **d** Net income following the BaU efforts of 1.5 °C. The size of the bubble refers to the times of net income compared with regional GDP in 2015. Colors from dark to light green and purple represent SP 2.0 scenarios from A to D and SP 1.5 scenarios from A to E, respectively.

capital and technology transfer from developed countries, which is consistent with Article 11 of the Paris Agreement. Relative vulnerable countries, for instance, Algeria and Colombia need 2.48–13.02 and 104.56–797.57 billion dollars of upfront investment for approaching the global warming targets, respectively, and turn into profit in 2030–2035 and 2060–2075. We have set the emission reduction target for each country, following the self-preservation strategy that is consistent with public perceived climate damage. The exact emissions reduction targets for each country are shown in Supplementary Tables 2 and 3. However, reaching the temperature-limiting goals require a relatively rapid declining rate of the low-carbon technology cost, which should be >15% every 5 years in order to achieve the 2 °C target and 20% to achieve the 1.5 °C target.

Compared to the existing effort-sharing studies, such as ref. [8], which collected over 70 studies that analyzed future GHG emissions allowances for different regions based on a wide range of equity allocation approaches, our results are more stringent. For example, in ref. [8], the enhanced strategy of the USA in terms of its GHG emissions reduction in 2030 is on average 44 and 64% compared with 2010 level for 2 °C and 1.5 °C targets, respectively, which are less stringent than the result of our study (79% and 94%, respectively). The reason for such difference may be because the deterministic warming targets, i.e., 2 °C and 1.5 °C in 2100 applied in this study are more stringent than the targets of ref. [8]. with a likelihood; and our results are economically optimum for each involved region rather than at the global scale. When combining different effort-sharing approaches and cost-benefit analysis, countries could benefit from avoiding potential climate impacts through more stringent mitigation efforts. Compared to

equitable allocations, our self-preservation strategies suggest a real costly effort that a country could put in and point out the net income a country could stand to gain.

Although this study contributes to displaying the real economic benefits for each country and has provided some insights for countries to reform their actions and update the NDCs in the post-Paris Agreement era, there are still a few limitations. For example, despite many adaptation strategies being proposed, more work is needed to assess the adaptation potentials and costs for managing climate change risks in C$^3$IAM model framework. In addition, the self-preservation strategy defined here is under the principle of economic benefits with the consideration of fairness for each country. Successful implementation of the self-preservation strategy is premised on improving the understanding of climate damages and the breakthroughs of low-carbon technologies. In addition to economic benefits, factors such as political attitudes, diplomacy policies, and environmental capacities are thought to be important determinants of climate mitigation actions of each country. This can be discussed in a future study.

## Methods

**Process for the model analysis**. To obtain a self-preservation strategy, we introduce an effort-sharing approach into the cost-benefit analysis. The first step is to account for the current NDCs and construct a policy-as-usual pathway for each region. Then, we calculate the effort-sharing indicators of each region by following the four mainstream effort-sharing principles, which is grandfathering, historical responsibility, ability to pay, and equal per capita allocation, and combine these indicators to define the integrated social welfare weights that will be used in the global welfare maximization function as a welfare weight of each region. Noted that the grandfathering approach determines the national efforts relying on the current

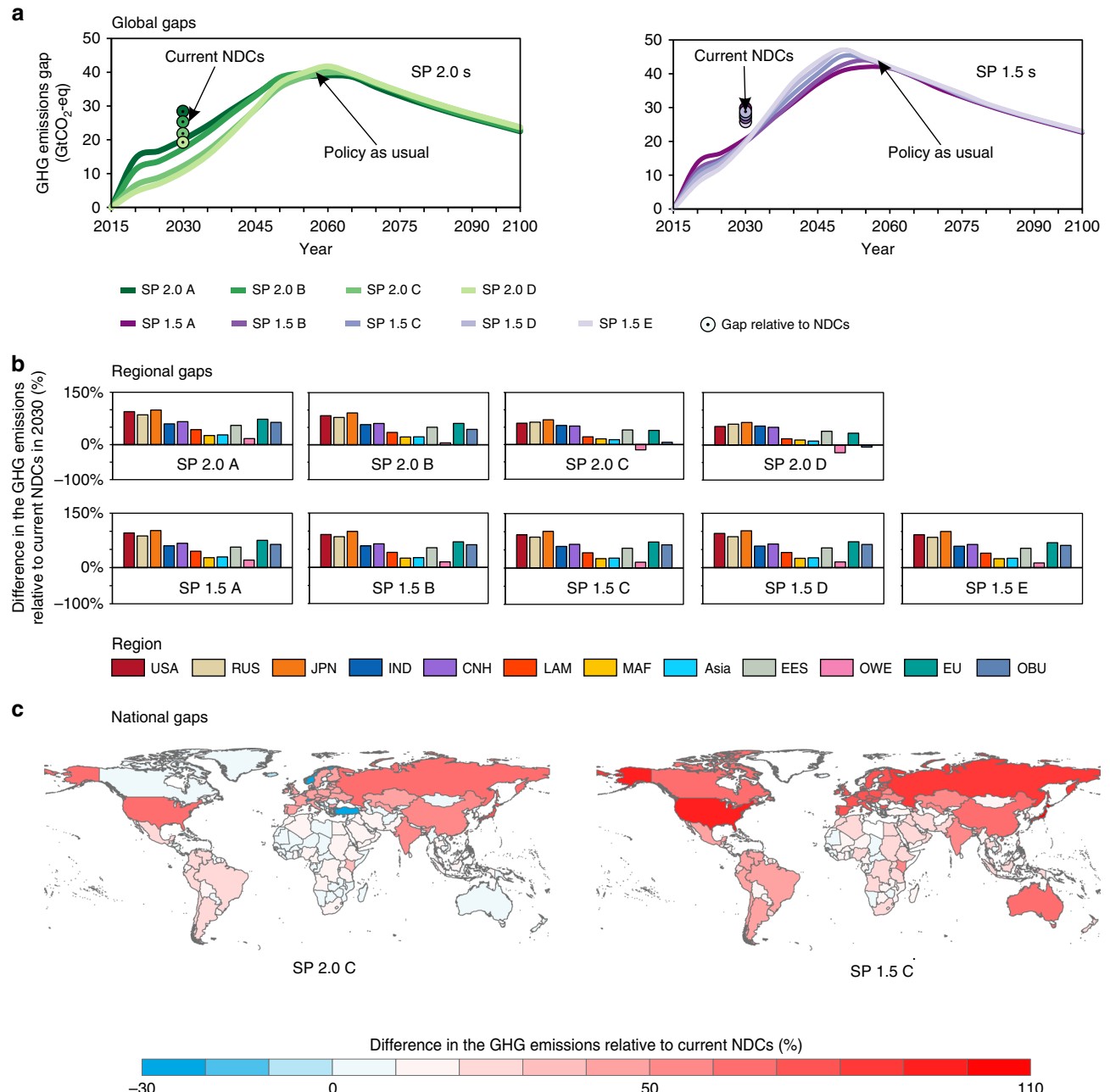

**Fig. 6 GHG emission gaps at the global, regional, and national levels. a** Emissions gaps at the global level. The lines stand for emissions gaps between policy-as-usual scenario following the current reduction efforts and self-preservation scenarios, and the points stand for emissions gaps between the current NDCs and self-preservation scenarios. Colors from dark to light green and purple represent SP 2.0 scenarios from A to D and SP 1.5 scenarios from A to E, respectively. **b** Emissions gaps between current NDCs and self-preservation scenarios at the regional level in 2030. **c** Emissions gaps between current NDCs and self-preservation scenarios at the country level in 2030. The positive emissions gap indicates the further required GHG emissions reduction.

emissions and is not conducive to countries with relatively low emissions in the base year. Thus, it is often criticized in the literature[35,36]. However, we choose to include it in the average because it represents one of the five IPCC equity categories and is implicitly followed by many of the developed countries[8,37]. The integrated social welfare weights are subsequently applied to simulate the optimal emission pathways in the cost-benefit analysis to improve the equity of mitigation efforts and allocation results across regions. Finally, the self-preservation strategies can be identified through comparing the relative benefits, and costs between optimal emission pathways and policy-as-usual pathways.

**Simulation of optimal emission mitigation pathways**. To simulate the optimal mitigation path, we applied a revised version of the global multiregional economic optimum growth model (C³IAM/EcOp), which is a submodule of the C³IAM (ref. [29]). It is established based on the theory of optimal economic growth and

consists of an economic module and a climate module. The economic module of C³IAM/EcOp is a modified version of a standard neoclassical optimal growth model. The climate module of C³IAM/EcOp links GHG emissions to concentration, radiative forcing, and temperature. In addition, C³IAM/EcOp takes into account the interaction between economic module and climate module by introducing the climate damage function and abatement cost function. We adopt the climate damage function derived from ref. [20], based on which the collective impact of many types of climate damages are included, such as damages to major sectors, such as (e.g. agriculture) adverse impacts on health, non-market damages, and estimates of the potential costs of catastrophic damages. We further compare the difference of the degrees of climate damage with the results of other existing climate impact models to define the uncertainty of climate damage.

To obtain a self-preservation strategy, we considered a grand cooperative situation. Mathematically, the object function was the global welfare, which was the

| Country | Strategy | GHG emissions in 2030 (GtCO₂-eq) | Net-zero year | Cumulative negative emissions (GtCO₂-eq) |
|---|---|---|---|---|
| China | SP 2.0 s | 5.62 (4.53 to 6.56) | 2045–2050 | −49.48 (−46.71 to −51.68) |
|  | SP 1.5 s | 4.61 (4.38 to 4.77) | 2040–2045 | −61.85 (−58.48 to −67.09) |
| India | SP 2.0 s | 3.49 (3.26 to 3.70) | 2060–2065 | −22.66 (−20.96 to −25.09) |
|  | SP 1.5 s | 3.26 (3.22 to 3.29) | 2050–2055 | −30.15 (−25.28 to −33.09) |
| EU | SP 2.0 s | 1.63 (0.93 to 2.25) | 2040–2045 | −26.85 (−25.25 to −28.87) |
|  | SP 1.5 s | 0.97 (0.85 to 1.06) | 2035–2040 | −31.45 (−30.85 to −32.46) |
| USA | SP 2.0 s | 1.37 (0.28 to 2.39) | 2035–2040 | −50.33 (−40.52 to −56.80) |
|  | SP 1.5 s | 0.37 (0.22 to 0.47) | 2035 | −47.79 (−42.04 to −52.62) |
| RUS | SP 2.0 s | 0.63 (0.33 to 0.92) | 2040–2045 | −13.38 (−14.11 to −12.75) |
|  | SP 1.5 s | 0.33 (0.29 to 0.37) | 2035 | −15.28 (−15.97 to −14.73) |
| JPN | SP 2.0 s | 0.19 (0.01 to 0.36) | 2035–2040 | −6.38 (−6.89 to −5.83) |
|  | SP 1.5 s | −0.01 (−0.02 to 0.01) | 2030–2035 | −7.24 (−7.32 to −7.10) |

Table 1 GHG emissions in 2030, timing of net-zero emissions and cumulative negative emissions of selected countries for the SP 2.0 s and SP 1.5 s, average over the four SP 2.0 strategies and five SP 1.5 strategies.

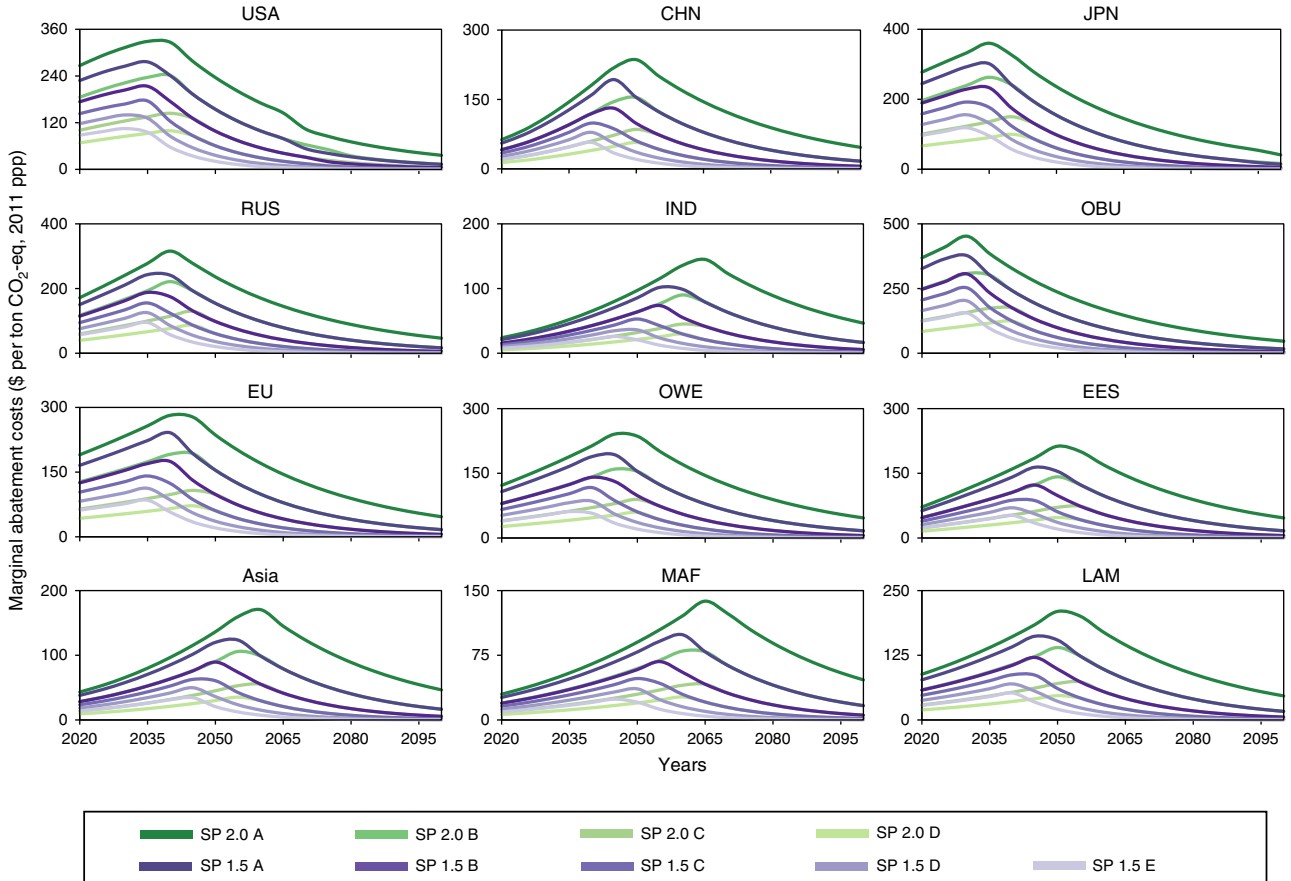

**Fig. 7 MAC by region (unit: dollars per ton CO₂-eq).** The green curve represents the SP 2.0 s, and the purple curve represents the SP 1.5 s. Colors from dark to light green and purple represent SP 2.0 scenarios from A to D and SP 1.5 scenarios from A to E, respectively.

weighted sum of welfare for different regions, as shown in Eq. (1).

$$\text{Max } W = \sum_i \varphi_i U^i \tag{1}$$

Where, $i$ stands for region, $U^i$ is the social welfare of region $i$, and $\varphi_i$ is the social welfare weight of each region, which reflects the importance of each region as well as the mitigation responsibility of each region. If the social welfare weight of one region become larger, which means the importance of social welfare of this region will be relative bigger, hence the mitigation rate of this region become relative smaller to maintain the global welfare maximization.

The social welfare function of each region is shown in Eq. (2).

$$U^i = \int_0^\infty L_i(t) Ln\left(\frac{C_i(t)}{L_i(t)}\right) e^{-\delta t} dt, \ i = 1, 2, \cdots, n \tag{2}$$

Where, $t$ stands for time period, $L_i(t)$ and $C_i(t)$ are the population and consumption of region $i$, respectively, and $\delta$ is the discounting rate.

The economic module of C³IAM/EcOp is a modified version of a standard neoclassical optimal growth model, as shown in Eqs. (3)–(6).

$$Q_i(t) = A_i(t) K_i(t)^\gamma L_i(t)^{1-\gamma}, \tag{3}$$

$$Y_i(t) = \Omega_i(t) Q_i(t), \tag{4}$$

$$C_i(t) = Y_i(t) - I_i(t), \tag{5}$$

$$\dot{K}_i(t) = I_i(t) - \delta_K K_i(t), \tag{6}$$

Where, $Q_i(t)$, $A_i(t)$, $K_i(t)$ represent gross output, technology, and capital, respectively, $\gamma$ is the capital share, $Y_i(t)$ is the net output, which is the gross output

excluding the damage and abatement costs, $\Omega_i(t)$ is the adjustment coefficient, which is a function of the damage and abatement costs, $I_i(t)$ is investment, and $\delta_K$ is the depreciation rate.

The climate module of C³IAM/EcOp depicts the physical process starting from GHG emissions changing to the GHG concentration to radiative forcing and, finally, affecting the atmospheric temperature, which is based on previous studies[20,21,38], as shown in Eqs. (7)–(15).

$$E_i(t) = \left(1 - \mu_i(t)\right)\sigma_i(t)Q_i(t) + E_i^{\text{land}}(t) \quad 0 \leq \mu_i(t) \leq 1, \tag{7}$$

$$M_i(t) = \sum_{s=0}^{t}\left\{ E_i(s)\left(\alpha_0 + \sum_k \alpha_k e^{\frac{s-t}{\tau_k}}\right) \right\}, \tag{8}$$

$$\dot{T}_1(t) = \varepsilon_{11} T_1(t) + \varepsilon_{12} T_2(t) + \varepsilon_3 F(t), \tag{9}$$

$$\dot{T}_2(t) = \varepsilon_{22}\left(T_2(t) - T_1(t)\right), \tag{10}$$

$$F(t) = \eta_1 \ln\left(\sum_{i=1}^{n} M_i(t) + \text{NAT}\right) - \eta_2 - O(t), \tag{11}$$

$$\Omega_i(t) = \left(1 - AC_i(t)\right) \times \left(1 - D_i(t)\right), \tag{12}$$

$$D_i(t) = 1 - \frac{1}{1 + a_{1,i}T_1(t) + a_{2,i}T_1(t)^2}, \tag{13}$$

$$AC_i(t) = b_{1,i}(t)\mu_i(t)^{b_{2,i}}, \tag{14}$$

$$b_{1,i}(t) = p \times (1 - g)^{t-1} \times \frac{\sigma_i}{b_{2,i}} \tag{15}$$

Where, $E_i(t)$ is the GHG emission, which is a combination of $CO_2$, $CH_4$, and $N_2O$ using the global warming potential values in the fifth IPCC Assessment Report, $\mu_i(t)$ and $\sigma_i(t)$ are the emissions reduction rate and GHG intensity, respectively, $E_i^{\text{land}}(t)$ is the land-use emissions of each region, which is exogenous, $M_i(t)$ is the GHG concentration, which is an impulse response function of GHG emissions, $\alpha_k(t)$ and $\tau_k$ are the coefficients of each period in the impulse response function, $T_1(t)$ and $T_2(t)$ are the global mean temperature of the surface and the deep oceans, respectively, $\varepsilon_{11}$, $\varepsilon_{12}$, $\varepsilon_3$, and $\varepsilon_{22}$ are the coefficients of the relationships between global mean temperature of the surface and the deep oceans, $F(t)$ represents radiative forcing due to atmospheric concentrations of GHG plus exogenous forcing ($O(t)$), NAT is the constant atmospheric concentrations of GHG, $\eta_1$ and $\eta_2$ are the coefficients in the radiative forcing function, $AC_i(t)$ and $D_i(t)$ represent the abatement cost fraction and climate damage fraction of gross output, respectively, $a_{1,i}$ and $a_{2,i}$ are the coefficients in damage function and $b_{2,i}$ is the coefficient in abatement cost function, $b_{1,i}(t)$ is the cost coefficient of the backstop technologies of region $i$, which decrease with time, and $p$, $g$, $\sigma_i$, and $b_{2,i}$ are the parameters in the backstop technology function.

**Estimation of the integrated social welfare weights.** In order to improve the equity of mitigation effort allocation in the cost-benefit analysis, we introduce the effort-sharing approaches of emissions rights to determine the social welfare weights in the global welfare maximization function (Eq. (1)) of C³IAM/EcOp. Based on this, the cost-benefit analysis is further conducted. Therefore, in our study, the mitigation effort across regions are derived on the basis of avoided climate damages and abatement costs of each region by meanwhile considering the equity in each region's social welfare weight. In order to combine the effort-sharing approaches with the cost-benefit analysis in C³IAM/EcOp, we calculated the relative mitigation responsibility of each region by adopting four major effort-sharing principles used worldwide that is principle of ability to pay, equal per capita allocation, grandfathering, and historical responsibility. Furthermore, we calculated the average of the weights of these four different methods. The resulting average weights were then used as social welfare weights in the simulation of optimal pathways. For the grandfathering principle, the permits are distributed equivalent to the baseline year emission, indicating that more emissions in baseline year would lead to lesser share of reduction burden. In this paper, GHG emissions, which are a combination of $CO_2$, $CH_4$, and $N_2O$ of each region at the 1990 level are regarded as the baseline. For historical principle, the permits are distributed equivalent to the contribution of global temperature increase over a certain period of time. This principle suggests that the reductions toward an overall emissions ceiling were to be shared among countries proportional to their relative share of responsibility for climate change. We use cumulative GHG emissions of each region from the period 1990 to 2017. For the ability to pay principle, the permits are distributed equivalent to per capita GDP, indicating that richer countries should have heavier reduction burden. We use each region's per capita GDP in the year 2017. For the equal per capita principle, the distribution of permits is in proportion to population. In this regime, the more people there are, the lesser responsibility there is to reduce emissions. We use each region's population for the year 2017.

**NDCs accounting and policy-as-usual scenario construction.** As little to no guidance or requirement was given, every aspect of the NDCs submitted was decidedly nationally[30]. The existing NDCs that have been submitted are ambiguous in terms of their definition and coverage. More than 70% of the ratified parties choose BaU scenarios as the emissions reduction reference. Few parties (only 48) have indicated their methodology for quantifying the BaU scenario, with none providing the data source. Even worse, some parties have proposed only mitigation and adaptation actions, which makes it difficult to precisely determine their future emissions[5,32,39–41]. All these strongly prohibit accurately accounting for their climate impacts. Therefore, to overcome the difficulties of accounting for emissions in the BaU scenario and to draw the NDC path for each country, we developed a Carbon Emission Extended Principle based on Structure (CEEP-S) method. First, the CEEP-S provides a transparent projection of future emissions in the BaU scenario by considering uncertain economic development (GDP) and dynamic emission intensity (GHG emissions per unit GDP); then, the NDCs are further quantified based on the BaU emissions.

The emission intensities of various GHGs (ins) are projected by a reduced-form model, as shown in Eq. (16):

$$\ln\left(ins_{ijt}\right) = \alpha_j + \alpha_{ij} + \beta_{jt} + \gamma_j \ln\left(ins_{ij,t-\tau}\right) + \eta_j \ln\left(Y_{ijt}\right) + \varepsilon_{ijt} \tag{16}$$

where $i$, $j$, and $t$ represent the country, GHG type and year, respectively; three types of GHG, including $CO_2$, $CH_4$, and $N_2O$, are considered; $\tau$ is the lag, which is set to five to reduce the influence of short-term changes[42], $ins_{ijt}$ is the emission intensity of $j$ type of GHG of country $i$ in year $t$; $\alpha$ is constant; $\alpha_{ij}$ and $\beta_{jt}$ are country fixed effects and time fixed effects, respectively; $Y_{ijt}$ is the GDP; $\gamma_j$ and $\eta_j$ are estimated coefficients; and $\varepsilon_{ijt}$ is the residual. The country fixed effects ($\alpha_{ij}$) reflect persistent differences across countries, such as fossil fuel availabilities and prices, the output mixes, the regulatory structures, tax and subsidy policies, and tastes. The time fixed effects ($\beta_{jt}$) reflect changes over time in domestic prices and changes in the technologies in use, environmental policies and standards, and the relevant taxes and subsidies.

To obtain the projection results, we initially forecasted the time fixed effects. We examined several models of the time fixed effects and report projections based on fitting the model with alternative specifications of these effects. Additionally, we chose specifications that included a linear spline part with a different growth rate prior to 1980 and after 1980, and a logarithmic trend after 1960, as shown in Eq. (17).

$$\beta_{jt} = \beta_{0j} + \beta_{1j}t + \beta_{2j}(t - 1980) \times 1[t \geq 1980] + \beta_{3j}\ln(t - 1060) \times 1[t \geq 1960] \tag{17}$$

When the original model is estimated, this alternative time fixed effect specification has the same goodness-of-fit performance.

Finally, we obtained the GHG emission projection of each country by combining Eqs. (16) and (17), which is the BaU emission path of each country. By combining this with the current NDCs to which each country has committed, the amount of emissions in the target year can be obtained.

The NDCs of each country are added to get the regional NDCs, in order to construct the policy-as-usual scenarios to find the self-preservation strategies for current NDCs. For regions whose NDCs are lower than BaU emissions, mitigation rates of the target year were obtained and a policy-as-usual emissions pathway was constructed by assuming the same mitigation rate with NDCs during the whole model period. For regions whose NDCs are higher than BaU emissions, we construct their policy-as-usual emissions pathway by using their BaU emissions pathways to avoid the situation that some countries actually do more than they have committed.

**Self-preservation strategies.** Based on our definition of self-preservation strategy, we estimated the self-preservation strategies from all optimal emissions pathways under different uncertainties of climate damage and low-carbon technology cost. The self-preservation strategy should meet the following constraints (Eqs. (18) and (19)):

$$\sum_{t=2015}^{2100} \text{Ben}_i(t) \geq \sum_{t=2015}^{2100} \text{Cos}_i(t) \, , \, \forall i \tag{18}$$

$$T_1(2100) \leq T\text{tar} \tag{19}$$

Where, $\text{Ben}_i(t)$ and $\text{Cos}_i(t)$ are the relative benefit and cost between policy-as-usual pathways and optimal pathways, respectively, which is formulated in Eqs. (20) and (21), and $T\text{tar}$ is the warming limits target, that is 1.5 °C or 2 °C target.

$$\text{Ben}_i(t) = \text{Dam}_i^{\text{PaU}}(t) - \text{Dam}_i^{\text{opt}}(t) \tag{20}$$

$$\text{Cos}_i(t) = \text{Abat}_i^{\text{opt}}(t) - \text{Abat}_i^{\text{PaU}}(t) \tag{21}$$

Where, $\text{Dam}_i(t)$ and $\text{Abat}_i(t)$ represent the climate damage and abatement cost for each region, respectively, which could be calculated using Eqs. (22) and (23).

$$\text{Dam}_i(t) = D_i(t)Q_i(t) \tag{22}$$

$$\text{Abat}_i(t) = AC_i(t)Q_i(t) \tag{23}$$

**Improved NDCs at the country level**. Due to data limitations, most of the literature on methods to guide countries in boosting their reduction ambitions has focused only on the regional level. However, it is critical for policy makers and stakeholders to find hotspots where NDC targets can be revised under various combinations of climate damage and technology development. Therefore, an improved NDC at the country level is required. To that end, we use the C$^3$IAM/DS (C$^3$IAM/downscaling) module to obtain improved NDC targets under different self-preservation scenarios for each country. To ensure a fair and efficient assignment of improved NDCs, we consider the common but differentiated responsibilities and respective capabilities. Indicators of responsibility, capability, and equality are used to downscale the gap for countries under the SP 2.0 s and SP 1.5 s.

$$\text{GapRate}_i^s = \text{GapRate}_R^s \times \omega_i \,, \forall i \in R \,, \sum_i \omega_i = 1 \qquad (24)$$

$$\omega_i = f(\text{responsibility, capability, equality}) \qquad (25)$$

$$\text{Gap}_i^s = \text{GapRate}_i^s \times \text{NDC}_i^s \qquad (26)$$

Where, $\text{GapRate}_i^s$ and $\text{GapRate}_R^s$ denote the relative GHG emissions gap between the current NDCs and the self-preservation scenario $s$ of country $i$ and region $R$ in 2030, respectively; $\omega_i$ refers to the integrated weight of country $i$; $\text{Gap}_i^s$ and $\text{NDC}_i^s$ is the GHG emission gap and current NDCs, respectively.

**Data sources**. Population, GDP, capital stock, and GHG emissions data used for effort-sharing indicators calculation and model estimation are obtained from the UN (ref. [43]), IMF (ref. [44]), CDIAC (ref. [45]), and EDGAR (ref. [46]). Future population and GDP data are from SSP2 (ref. [47]; a more middle-of-the-road development pattern of Share Socioeconomic Pathways). The main parameters, including the coefficients of climate damage in the C$^3$IAM/EcOp model are from Yang[19] and Nordhaus[20,21].

**Reporting summary**. Further information on research design is available in the Nature Research Reporting Summary linked to this article.

## Data availability

The source data underlying Figs. 1–7 and Supplementary Figs. 1–6 are provided as a Source Data file. The data that support the plots within this paper and the findings of this study are available from the corresponding author upon reasonable request.

## Code availability

The codes that support the methods of this study are available from the corresponding author upon reasonable request.

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

## Acknowledgements

The authors gratefully acknowledge the financial support of China's National Key R&D Program (2016YFA0602603), the National Natural Science Foundation of China (nos. 71822401, 71521002, 71603020, and 71704009), Beijing Natural Science Foundation (JQ19035), Huo Yingdong Education fund and the support from the Joint Development Program of Beijing Municipal Commission of Education. The authors would like to thank their colleagues for their support and acknowledge help from CEEP-BIT. Deep thanks to anonymous reviewers for their valuable comments on the manuscript, which means a lot to improve our work.

## Author contributions

Y.-M.W. and B.Y. conceived the study and performed the analysis. C.W., Q.-M.L. and Z.Y. implemented the model. X.-C.Y., R.H., and L.C. contributed to the scenario design. R.H., J.C., and Q.Z. contributed to the data collection and figure design. H.L., B.T., and J.Y. worked on the review and editing. All authors approved and contributed to writing the paper.

## Competing interests

The authors declare no competing interests.
