## [Peer Review File · Nature Communications]

Reviewers' comments:

Reviewer #1 (Remarks to the Author):

Dear authors,

Thank you very much for this interesting work and the nicely presented graphs.

I see important value in the analysis presented here but the methods are not clearly explained in the main text, and not explained in plain language in the methods section. I do not understand how the equity considerations are mixed with the importance of avoided climate damages. The main text needs to better explain what we are looking at and calling it a winwin strategy is not sufficient.

In particular, I understand that the avoided climate damage are used as a metric to allocate mitigation effort. A country that can avoid a lot of damages in a 1.5°C world compared to a 5°C world will contribute more to the mitigation effort. Is it up to the break-even points in terms of expenditure (it would be interesting to have the break-even temperature for each country and compare it to the global warming implied by the ambition of their NDC: <https://www.nature.com/articles/s41467-018-07223-9/>). So the most vulnerable countries, often the poorest with limited adaptation capacity, will have the greatest burden to mitigate. It is deeply unfair and violates the Paris Agreement principle of equity (CBDR-RC principle and article 2), the recognition of the special vulnerability of countries and their priority to receive support (when the win-win is giving them more burden). These points need to be stated clearly or disproven if you disagree. While the win-win strategy can bring an interesting message, it cannot hide its deep unfairness and its incompatibility with the Paris Agreement and the UNFCCC. Can you please clarify this point and how this winwin strategy is equitable or not?

Please note that I see great value in this work even if the results are not equitable. But if they are not, I would not present this strategy as an option for negotiations (it violates basic morality and international agreements). Instead, your results could highlight the potential economic benefits for each country to mitigate up to the allocation you derived (the break-even point). The difference with equitable allocations could then be discussed as the real 'costly effort' that a country could provide, or the net benefit that a country could gain.

Please explain clearly what is understood by a winwin situation. How is this different from the cost-optimal allocation of mitigation efforts from IAM? How is it different from burden sharing/ equity allocations? Not all readers have this expertise. Please also explain what is the potential advantage of this metric (supposedly more acceptable to parties?), and drawbacks (is that fair to ask countries for mitigation effort that reflects their vulnerability to climate change? I would argue not. Also how can it be combined with cost-optimal modelling from IAMs)? Please discuss the results here and their implications in light of the existing literature instead of simply stating that all previous studies neglects the most important fact.

What are the assumptions for non-party emitters such as IMO and ICAO?

Please also explain whether emissions from land use are accounted for. Please detail which GWP is used to aggregate GHG emissions.

I am also unsure how these results account for adaptation when calculating climate impacts. I can imagine that adaptation is sometimes cheaper than mitigation to avoid climate impacts. How can such considerations change the results presented here?

Please also explain that while a winwin strategy sounds interesting to all parties. It may be that some countries are willing to win even more, which can cause others to lose (prisoner's dilemma), or that fairness or other considerations may influence countries' negotiating positions. In general, please embed the game theory in political/diplomatic reality to highlight the advantage and drawbacks of this approach.

What are the assumptions on negative emissions (there has to be negative CO₂ to compensate non-CO₂ GHG)? Why do none of the winwin strategy feature negative emissions? It seems to result from the modelling hypothesis more than by feasibility considerations. Please discuss implications given

that most IAM scenarios find negative emissions to be necessary to achieve the 1.5°C or well below 2°C scenarios at least cost.

The literature review is a bit light. In particular, it would be interesting to compare your results with Burke et al.2019 (<https://doi.org/10.1038/s41586-018-0071-9>) who found temperature optima for various economies. In the case of your study and given my limited understanding of your methods, why would a country that is better off with some global warming (Russia or other Nordic countries) cooperate and mitigate emissions? How would it be a winwin for them?

In general, much more details and clarity are required on the methods mostly in the main text. So far, the article praises the winwin strategies without detailing their constructions. More details are needed to explain the individual scenarios: what are the warming definitions and likelihood of 1.5°C and 2°C? what are the overshoots etc.

I do not understand the inclusion of equity, both technically speaking and philosophically speaking. How is equity taking part in a winwin situation? It would be worth explaining.

Where is the data from in general? Where are the equity allocations coming from? Where is the climate damage data coming from? Where is the emissions data coming from? Please mention these in a sentence explaining the reasons of your choice.

Thanks a lot for your work. I believe that it can give interesting insights and discussions after more clarity is brought to help the reader what (s)he is looking at and how to understand the results.

Abstract

Line 13 – I do not agree the factual benefits considered in the win-win strategy depend on a subjective attitude (pessimistic). I also would refrain from phrasing such as ‘there will always be’ unless it’s clearly proven in the text. It reads oddly after putting a vague condition on the attitude of countries for that statement. What is a pessimistic attitude? It is not defined in the text.

My understanding is that the situation can be win-win based on the climate impacts linked with inaction and the cost of action. I would simply articulate the sentence around these facts.

Please state that this win-win, is a win for some countries only compared to absolute inaction and its climate impacts (so absolute unfairness). However, it is still a loss compared to potentially fairer allocations (e.g. burden sharing and equity). A ‘win’ is a comparison, and the point of comparison is not defined in the abstract.

Line 15: The added value of this paper in terms national ‘allocations’ is very vague in the abstract. Most earlier studies agree that more contributions are needed from these countries and the Paris Agreement requires them to ratchet up. Please state your results more prominently.

Page 2:

Line 21 Why does it indicate that we are at crossroads of governance? Also, the COP24 did not aim at discussing efforts. The ratcheting up of NDC is to be discussed by 2020 and COP26.

Line 31: The enhancement of NDC (not simple update) is required by the Paris Agreement, which is only confirmed in the Katowice package

Line 34: There is no 2°C target. At minimum, there is a ‘well below 2°C’. Please state what the upcoming summit is for people reading your article in the future.

Line 38 it’s ‘well below 2°C’

Line 42: Do you have a reference to back up the statement that policy makers find it harder to adopt the strategy? My experience is that they prefer actually budget allocation that they can use the way

they want (and sometimes overshoot).

Line 46: these references derive allocations for all countries to my knowledge.

Page 3

Line 51: these references are from 1996 and 2008, do you have anything more recent?

Line 53: The consequence of the lack of equity of this approach really need to be stated more clearly and deserve a few sentence. The win-win strategy allocates effort to mitigate emissions on the basis of the different exposures of countries to climate impacts. So the most vulnerable countries, often the poorest with limited adaptation capacity, will have the greatest burden to mitigate. It is deeply unfair and violates the Paris Agreement principle of equity (CBDR-RC principle and article 2), the recognition of the special vulnerability of countries and their priority to receive support (when the win-win is giving them more burden). These points need to be stated clearly or disproven if you disagree. While the win-win strategy can bring an interesting message, it cannot hide its deep unfairness and its incompatibility with the Paris Agreement and the UNFCCC.

Line 56: This is an overstatement. The article does not prove that consensus only relies on equity and a win situation over climate impacts. Many other factors, including political and diplomatic can prevent consensus. Please avoid such strong statements.

Line 56: who is 'we'? And this statement sound obvious. If there is a lack of will, obviously a stronger willingness would help.

Line 62: do you have a reference? I strongly disagree, the willingness or not to tackle climate change is driven by many factors and the economics is only a small part. Many negative cost options are not implemented, which shows that it's not the main drive.

Line 64 why using 'would'? I do not see a conditionality in that statement. I would use 'will'.

Line 65: why 'would'?

Page 4:

Line 70: please define what is 1.5°C and 2°C here. What likelihood? What overshoot? By when? Compared to what?

Line 70: The statement implies that the only possible win-win strategy is the one suggested here, and that is compared to inaction. Please indicate that other win-win strategies could exist. Many argue that equitable solutions would be beneficial for all. Some could argue that the status quo is by definition the win-win strategy, all factors accounted for (not just win over climate impacts) as it reflects the choices of governments. Some countries can consider a relative win over a given competitor more important than an absolute win (and could accept an absolute loss as relative win).

Line 74: The choice of the term 'will' reflect the legal language of the Paris Agreement but is here understood as a prediction that is not backed up (and actually unlikely, research shows that many countries are not on track to achieve their NDC:

<https://www.sciencedirect.com/science/article/pii/S030142151830750X>).

Line 75: which studies? Please provide references.

Line 78: Are the UNFCCC and the Paris Agreement not global cooperation strategies?

Line 80: Suggestion: the next two sentences could be inverted to reflect chronological order.

Page 5

Before stating results, there need to be a description of what the scenarios are made of and what are the assumptions. Beyond the absence of discussion of the parameters, the text does not event introduce what the scenarios entail. They are just called win-win, and consider climate impacts, costs and equity somehow. There are no more details at this point. The reader needs to understand that before reading about the results.

Line 89: I am not sure to understand. The uncertainty does not result in more scenarios. It it just reflects the uncertainty of a given set of scenarios. While the results may yield a greater range, it does

not imply that there is a broader 'package of specific scenarios'.

Line 92: Please briefly explain which uncertainties are covered and how.

Line 92: as a percentage of what? Of all the scenarios that the paper quantifies as winwin? Based on what range of assumptions? This is unclear. Please start by a paragraph on your model, its goals and modelling assumptions. At this stage, the reader does not know what the scenarios are made of.

Line 95: 'if we would' is not grammatically correct. I suggest 'if society does not have a slower technology development,...'

Line 95 slower than what? This section is very unclear and comparison are left without objects.

Line 97: please define what are these medium to high damages and low costs. These labels are introduced in the article but do not have a universal definition. The article needs to explain what is understood by that.

Line 102: The pathway does not seem to show that it is challenging. The pathway confirms that NDCs are not in line with 2°C.

Line 103: This section needs to come before the Results section (see my prior comments on the lack of explanation on the scenarios). Again, please explain what is understood by '2°C/1.5°C'. Is that a 50% chance? Through the century or in 2100 with possible overshoot?

Line 105: what are the characteristics of the different levels of climate damages? Could there be a table summarising the main differences?

Line 198: 90% of what? Of their current levels? Of a reduction compared to BaU? Compared to their 2030 NDC emissions? Please be more specific.

Line 202: a guide may be an overstatement. Information may be more appropriate.

Line 203, Fig4a: given that the policy as usual path is the same in each graph, there might be better and clearer representations that help comparing the different scenarios. Perhaps bundle in 2 graphs for 1.5°C and 2°C

Page 12

Line 215: what would a relaxing of policies imply? What does it mean to increase the stringency? If the article is not discussing the policies, it may be better to only discuss the needed emissions trends. The whole paragraph is unclear.

Page 13:

Line 231: The PA is not really about climate actions. It is more about

Line 232: Please explain how it's a continued success? The first test will really be at the COP26 with the outcome of the ratcheting up process. Also, how can a success 'require' countries to do something?

Line 240: Given the winwin narrative, I would phrase this as 'Most Parties would gain from raising their target and align with a 2°C objective'.

Line 246: again, it is important to explain that the net positive income is compared to inaction. And that the situation would be very different if compared to a more equitable approach, in particular for developing countries. Likewise, I would rather speak of avoided damages than net incomes.

Line 253: how do you define what is acceptable?

Line 254: it may be best to avoid overstatements. Science did not conclude that societies would be destroyed. Also, destruction is presented as the alternative to your winwin scenario. It comes across as threatening. The advantages of your scenario should speak for themselves.

Page 15

Line 260: It may be better to state that you have set these targets, rather than you can set them. Perhaps provide them in a separate table.

The last paragraph of the discussion could come as a second paragraph. It reads oddly as a concluding statement and ends with numbers.

Methods: Please introduce a general sequence of the steps before going into details.
Line 389: 'in consistent with' reads oddly.

Reviewer #2 (Remarks to the Author):

Thank you for giving me the chance to review manuscript NCOMMS-19-28971

I find it well-written yet with numerous potential (mostly minor and a few major) improvements.

- Although, to begin with, I am not very comfortable with the authors' non-disclosure of (socioeconomic/input) data, the employed model (C3IAM) is referred to as a model established for evaluating INDCs based on the SSPs. Are we to assume a specific SSP is used for this analysis? If so, how does that scenario relate to the technological and marginal abatement costs assumed here?
- Regarding the determination of the cooperative scheme, the EcOp module of the C3IAM model is selected. As a standard neoclassical optimal growth model, it calculates parties' willingness to reduce emissions as a function of associated costs and benefits. I am trying to understand what the regional/national benefits for each Party are: if these benefits (climate damage aversion) are at the Party scale, then Parties choose their emissions reductions based on how vulnerable they are? If so, this is not in line with the Paris Agreement equity principles.
- Nordhaus' 2010 paper is referred to for representing climate damage in terms of (global) GDP in 2100; is this estimate up to date?
- The authors select nine 'win-win' scenarios, relating to levels of technological costs and climate damage (and assuming given stringency), which help explore future pledges targets, but (aside from technological breakthroughs) there is no discussion of policy and feasibility, hence no discussion of how realistic these scenarios can be.
- How is stringency defined? Is it (through abatement costs) part of the scenario inputs for the win-win trajectories or part of the findings?
- Authors mention the use of game theory, but there is no discussion on it, unless it is attributed to the fact that targets for multiple agents (countries) are calculated, based on costs determined at the national level while (climate) benefits at the global.
- I was most confused in lines 96-98: "scenarios under the conditions of medium to high climate damage and low technological costs, accounting for 27.2%, can achieve the 1.5°C target". What exactly is 27.2%, and as percentage of what? What does the overall sentence mean, given that authors specify scenarios of low to high climate damage? Does it mean that if climate damage is low, the 1.5°C cannot be achieved? What about WW1.5L (line 110)?
- Lines 113-114: although Figure 1b shows negative emissions as early as 2035, in the 1.5°C scenario.
- From the description of Figure 1(a), I am confused as to whether, for example, a 10% decline rate means that the costs drop by 10% every five years, or that the decline rate of the costs increases by 10% every five years.
- From the discussion of the method for accounting current NDCs and constructing the policy as usual scenario, do I understand correctly that NDC projections used may be largely incorrect in respect to what is intended in the actual NDCs?
- The manuscript would certainly benefit from a comparative analysis discussion on relevant studies.
- The existing Discussion is more of a Conclusion section, and even so lacks a discussion of limitations and a framing of how the policy prescriptions should be considered. The finding that "higher climate damage, faster technological growth presents opportunities for more benefits for all parties" is rather trivial, and the last paragraph has been written hastily and not as informative.
- On a side note, the figures are well elaborated.

To sum up, I find this manuscript to be worth publishing after some clarifications and revisions.

Response to reviewers' comments

We appreciate the reviewer for his/her insightful review. The comments and suggestions have contributed substantially to improve our paper. We have tried our best to revise the manuscript. Our point-by-point responses are as follows.

■ To Reviewer #1's comments:

Dear authors,

Thank you very much for this interesting work and the nicely presented graphs.

RESPONSE:

Thanks to the reviewer for the positive comments. We tried our best to revise our work and addressed all comments as follows.

1. I see important value in the analysis presented here but the methods are not clearly explained in the main text, and not explained in plain language in the methods section. I do not understand how the equity considerations are mixed with the importance of avoided climate damages. The main text needs to better explain what we are looking at and calling it a win-win strategy is not sufficient. In particular, I understand that the avoided climate damage are used as a metric to allocate mitigation effort. A country that can avoid a lot of damages in a 1.5°C world compared to a 5°C world will contribute more to the mitigation effort. Is it up to the break-even points in terms of expenditure (it would be interesting to have the break-even temperature for each country and compare it to the global warming implied by the ambition of their NDC: <https://www.nature.com/articles/s41467-018-07223-9/>). So the most vulnerable countries, often the poorest with limited adaptation capacity, will have the greatest burden to mitigate. It is deeply unfair and violates the Paris Agreement principle of equity (CBDR-RC principle and article 2), the recognition of the special vulnerability of countries and their priority to receive support (when the win-win is giving them more burden). These points need to be stated clearly or disproven if you disagree. While the win-win strategy can bring an interesting message, it cannot hide its deep unfairness and its incompatibility with the Paris

Agreement and the UNFCCC. Can you please clarify this point and how this win-win strategy is equitable or not?

RESPONSE:

Thank you for your suggestion and encouragement. We have considered the effort-sharing by assigning different responsibilities to different regions in our analysis. Specifically, based on effort-sharing methods, we first construct a social welfare weight for each region by considering its responsibility (grandfathering and historical responsibility), capability (ability to pay) and equality (equal per capita allocation). We further introduce these social welfare weights that can indicate the equity among regions into the cost-benefit analysis. Each region would determine the optimal mitigation effort under the given social welfare weights through the C³IAM model. In this way, all regions' social welfare can be improved compared with that of current NDCs. The principles (responsibility, capability and equality) we used in this study can reflect the CBDR-RC principle used worldwide. As it were, the win-win strategy combines the effort-sharing approaches with cost-benefit analysis of emission mitigation, which is compatible with the Paris Agreement and the UNFCCC. And our results showed that to achieve no matter a well below 2°C or 1.5°C target, more contributions need to be made by Japan, EU, the USA, India, Russia, the Eastern European and Commonwealth of Independent States and China. Though some vulnerable countries (mainly located in the Middle East and Africa, Latin America) still have relative higher burden on emissions mitigation, they can have a quick net benefit if they could experience a rapid reduction of low-carbon technology costs. Therefore, in order to releasing the burden of those vulnerable countries and to achieve warming targets, the global cooperation is required and those vulnerable countries need capital and technology transfer from developed countries, which is in consistent with the Article 11 of the Paris Agreement. To make these points more clear, we added some details of our methods and rewrite the description in the last paragraph of *Introduction* and the first paragraph of *Methods*.

Changes in the manuscript are shown below:

Introduction: To derive a win-win strategy that can outperform the current NDCs and is more acceptable by the parties, we specifically consider the responsibility

(grandfathering and historical responsibility), capability (ability to pay) and equality (equal per capita allocation) to ensure each region's equitable burden sharing in response to climate mitigation (see Methods). An indicator named social welfare weight is constructed to represent the equity following effort-sharing methods. Each region would then determine its optimal emission pathways under its given social welfare weight through the C³IAM (see Methods for detailed process)

Methods: To obtain a win-win strategy, we introduce the effort-sharing approach into the cost-benefit analysis. The first step is to account the current NDCs and construct the policy-as-usual pathways of each region. Then we calculate the effort-sharing indicators of each region following four effort sharing principles, i.e. grandfathering, historical responsibility, ability to pay and equal per capita allocation, and combine these indicators to determine the social welfare weights. The social welfare weights are subsequently applied to simulate the optimal emission pathways in the cost-benefit analysis. In the end, the win-win strategies can be identified through comparing the relative benefits and costs between optimal emission pathways and policy-as-usual pathways.

2. Please note that I see great value in this work even if the results are not equitable. But if they are not, I would not present this strategy as an option for negotiations (it violates basic morality and international agreements). Instead, your results could highlight the potential economic benefits for each country to mitigate up to the allocation you derived (the break-even point). The difference with equitable allocations could then be discussed as the real 'costly effort' that a country could provide, or the net benefit that a country could gain.

RESPONSE:

Thanks for your suggestion and encouragement. As mentioned in the previous reply, the considerations of effort-sharing approaches are used to determine the social welfare weights of each region. And then each region would determine the optimal mitigation effort under the given social welfare weights. Therefore, our win-win strategy combines the effort-sharing approaches with cost-benefit analysis of emission mitigation, which is compatible with the Paris Agreement and the UNFCCC. And our results showed that to achieve no matter a well below 2°C or 1.5°C target, developed countries need to bear more responsibility than backward ones. According to our

results, vulnerable countries (mainly located in MAF and LAM) could have net income following win-win strategy.

As for the well below 2°C target, countries in MAF, which are vulnerable in climate change, could reach break-even point between 2050 and 2060, and the cumulative net income in 2100 obtained is 0.61 to 3.50 trillion dollars, which is 0.45 to 2.62% of cumulative GDP. Countries in LAM could reach break-even point between 2080 and 2085, with the 0.17 to 0.92 trillion dollars cumulative net income in 2100, equivalent of 0.18 to 0.97% of GDP. These countries could reach break-even point earlier and obtain more benefits from achieving 1.5°C target, which is 2050 (2.08 to 4.67 trillion dollars) and 2060 to 2065 (0.70 to 1.48 trillion dollars) of MAF countries and LAM countries, respectively. That is to say, we not only consider the equity but also net income of emissions reduction for each region. Thus, the results could provide reference for ratified parties to enhance their reduction targets.

3. Please explain clearly what is understood by a win-win situation. How is this different from the cost-optimal allocation of mitigation efforts from IAM? How is it different from burden sharing/ equity allocations? Not all readers have this expertise. Please also explain what is the potential advantage of this metric (supposedly more acceptable to parties?), and drawbacks (is that fair to ask countries for mitigation effort that reflects their vulnerability to climate change? I would argue not. Also how can it be combined with cost-optimal modelling from IAMs)? Please discuss the results here and their implications in light of the existing literature instead of simply stating that all previous studies neglects the most important fact.

RESPONSE:

Thanks for your comments and suggestion. In this study, the win-win strategy means that the temperature-limiting goals could be reached (well below 2°C or 1.5°C), the social welfare would be improved and the relative benefits (i.e., the difference between the climate damage avoided and the extra abatement cost) would be increased compared with the current NDCs at both the global and national levels. Considering the uncertainty level of climate change and technology development, there will be a set of specific scenarios. We select nine representative win-win scenarios with higher social welfare for further analysis. Burden sharing or equity

allocations follows the equity principles (i.e., ability to pay, egalitarianism, grandfathering or historical responsibility), however, none of these equity principles could completely equalize the national benefits. Since there exists gap between quota allocation and the actual demand, most equity-based ability allocation schemes are not effective. Some studies apply cost-optimal approaches. But cost optimal method don't consider the national mitigation benefit from the avoided climate risk. In order to overcome the shortcomings of existing allocation plan, compared with current cost-optimal methods, we consider benefits of each region. Compared with traditional cost-benefits methods, we consider both economic benefits and equity of mitigation schemes by introducing effort-sharing principles. We have clarified the contribution of this work in the last paragraph in "Introduction" section. In addition, we add a paragraph in the *Discussion* section to compare the difference with existing literature.

Changes in the manuscript are shown below:

Compared to existing effort-sharing studies, for example Ref. 36 which compared over 40 studies that analyze future GHG emissions allowances for different regions based on a wide range of effort-sharing approaches and long-term concentration stabilization levels, the range of our results is close to that of other studies. For the Organization for Economic Co-operation and Development (OECD) regions (1990 classification) and LAM, proposals in the existing studies (negative allowances in 2030 of -75% to -37% change from 2010 level) and in our study (-66% to -37% in the same year) are close. However, the proposals for Economies in Transition (EIT), ASIA and MAF are more stringent than previous studies. When combining different effort-sharing approaches and cost-benefit analysis, countries could benefit more or less from avoiding the potential climate impacts through more stringent mitigation efforts. Compared with equitable allocations, our win-win strategies provide a real 'costly effort' that a country could provide and point out the net income a country could gain.

4. What are the assumptions for non-party emitters such as IMO and ICAO? Please also explain whether emissions from land use are accounted for. Please detail which GWP is used to aggregate GHG emissions.

RESPONSE:

Thanks for your comments. In this paper, we aim to find win-win strategy for current NDCs, so we only focus on the ratified parties and don't consider the non-party emitters. The exogenous land use is considered and the global warming potential values (GWP) in the 5th IPCC Assessment Report are used in this study. We added these information in *Method for simulating the optimal pathways*.

Changes in the manuscript are shown below:

where $E_i(t)$ is GHG emissions, which are a combination of CO₂, CH₄ and N₂O using the global warming potential values (GWP) in the 5th IPCC Assessment Report, $\mu_i(t)$ and $\sigma_i(t)$ are the emissions reduction rate and GHG intensity, respectively, $E_i^{land}(t)$ is the land-use emissions of each region, which is exogenous.

5. I am also unsure how these results account for adaptation when calculating climate impacts. I can imagine that adaptation is sometimes cheaper than mitigation to avoid climate impacts. How can such considerations change the results presented here?

RESPONSE:

Thanks for your comments. With the focus on climate mitigation, adaptation was not considered in this study yet. We agree that adaptation is sometimes cheaper than mitigation to avoid climate impacts, and that consideration might change our results. But, after applying adaptation measures, the climate change risks might be avoided would also become smaller. Therefore, the allocation results among countries still unclear. This key point could be taken into account in our future study. We have added this research limitation in the end of *Discussion*. Thanks again for your valuable suggestion.

Changes in the manuscript are shown below:

Although this study has contributed to improving the current NDCs by considering both the Paris Agreement targets and economic benefits for each ratified party, some limitations are left to the future work. For example, in recent years, many adaptation strategies have been proposed. More work is needed to assess the adaptation

potentials and costs for managing climate change risks in C³IAM model framework.

6. Please also explain that while a win-win strategy sounds interesting to all parties. It may be that some countries are willing to win even more, which can cause others to lose (prisoner's dilemma), or that fairness or other considerations may influence countries' negotiating positions. In general, please embed the game theory in political/diplomatic reality to highlight the advantage and drawbacks of this approach.

RESPONSE:

Thanks for your comments. In this study, win-win strategy is Pareto optimality instead of game theory, which means even the ratified party make further emissions reduction, the net income will not be improved compared with the win-win plan. The condition for the win-win strategy is that every region should have positive net income compared to their current NDCs, which means every region will win in the win-win strategies. And the fairness of win-win strategy is determined by the social welfare weights that reflect effort-sharing considerations. Therefore, the win-win strategy will be more acceptable for all parties under the global cooperation mechanism. We have corrected the descriptions in the main text.

7. What are the assumptions on negative emissions (there has to be negative CO₂ to compensate non-CO₂ GHG)? Why do none of the win-win strategy feature negative emissions? It seems to result from the modelling hypothesis more than by feasibility considerations. Please discuss implications given that most IAM scenarios find negative emissions to be necessary to achieve the 1.5°C or well below 2°C scenarios at least cost.

RESPONSE:

Thanks for your comments. Actually, we have introduced the negative emissions considering the practical feasibility. In particular, the upper limit is set based on the negative emissions level in the existing scheme under Share Socioeconomic Pathways. The negative emissions level of each region is determined by the global welfare maximization. In our study, all the win-win strategies feature negative emissions in the later period (as shown in Fig. 1d).

8. The literature review is a bit light. In particular, it would be interesting to compare your results with Burke et al.2019 (<https://doi.org/10.1038/s41586-018-0071-9>) who found temperature optima for various economies. In the case of your study and given my limited understanding of your methods, why would a country that is better off with some global warming (Russia or other Nordic countries) cooperate and mitigate emissions? How would it be a win-win for them?

RESPONSE:

Thank you very much for your suggestion. We agree with you that in the short term, Russia and Nordic countries would better off with some global warming. But, in the long term, climate change has negative impact on all regions. This conclusion could also be found in Burke's study (*Burke M, Hsiang S M, Miguel E. Global non-linear effect of temperature on economic production. Nature, 2015*). They suggest that the long-run effects of temperature on economic growth are negative, so that the cooler countries might not benefit on net.

9. In general, much more details and clarity are required on the methods mostly in the main text. So far, the article praises the win-win strategies without detailing their constructions. More details are needed to explain the individual scenarios: what are the warming definitions and likelihood of 1.5°C and 2°C? What are the overshoots etc.

RESPONSE:

Thanks for your comments. We have added some details of methods and rewrite the construction of scenarios in the text. As for warming, we focus on the average atmospheric temperature change in 2100. If the temperature change in 2100 is less than 2°C and every region could gain compared with policy-as-usual, the optimal emission scenario is regarded as win-win scenario for 2°C, and the same for 1.5°C. The overshoots are not allowed for win-win scenario here. The warming targets in this study are in consistent with the Paris Agreement. We have added this explanation in *Introduction-win-win strategies identification*.

Changes in the manuscript are shown below:

The scenario setting considers four aspects, including warming limits targets, low-carbon technology costs, climate damage and equity principles. As for warming, we focus on the average atmospheric temperature change in 2100. If the temperature change in 2100 is less than 2°C and every region could gain compared with policy-as-usual, the optimal emission scenario is regarded as win-win scenario for 2°C, and the same for 1.5°C. The overshoots are not allowed for win-win scenario here. The warming targets in this study are in consistent with the Paris Agreement. Meanwhile, different levels of low-carbon technology costs and climate damage to reflect the uncertainty of low-carbon technology development and climate risk are considered. According to the uncertainty level of climate damage and technology development, there will be a package of specific optimal emission scenarios (Fig. 1a). The changes of low-carbon technology costs and climate damage are set in accordance with previous studies^{19-20, 26, 34}. We define the level of climate damage by using the ratio of economic damage in GDP. The enlarged coefficients (increase times of climate damage shown in Fig. 1a) of damage function in the model is used to characterize different levels of climate damage. The values reported in the reference 19 is set as the reference level, which show a climate damage being approximately 1.6% of GDP at a 2.62°C warming in 2100. With a given temperature rise, the economic damages are assumed to be different times as large as the reference level. Specially, Supplementary Fig. 1 shows the changes in economic damage under different uncertainties with temperature rise for each region. In this study, we define high, medium and low level of climate damage, which is corresponding to the increase times of climate damage coefficients used in the damage function being 4 to 5 times, 2 to 4 times and less than 2 times, respectively. In addition, we define three levels of technological development, i.e., slow development with low decline rate of low-carbon technology costs being less than 10% every five years, medium development with medium decline rate of low-carbon technology costs being 10% to 30% every five years, and rapid development with high decline rate of low-carbon technology costs being 30% to 40% every five years. The capability and responsibility of developing and vulnerable countries are fully considered by introducing equity principles. The social welfare weight of each region are shown in Supplementary Table 1.

10. I do not understand the inclusion of equity, both technically speaking and philosophically speaking. How is equity taking part in a win-win situation? It would be worth explaining.

RESPONSE:

Thanks for your comments. We have considered the effort-sharing by assigning different responsibilities to different regions in our analysis. By introducing responsibility (grandfathering and historical responsibility), capability (ability to pay) and equity (equal per capita allocation) into cost-benefit analysis, the social welfare weights of each region is constructed by adopting effort-sharing methods. After that, each region would determine its optimal mitigation effort under the given social welfare weights through the C³IAM model. In this way, all regions' social welfare can be improved compared with current NDCs. The principles (responsibility, capability and equity) we used in this study can reflect the CBDR-RC principle used worldwide. As it were, the win-win strategy combines the effort-sharing approaches with cost-benefit analysis of emission mitigation, which is compatible with the Paris Agreement and the UNFCCC. We have added the description of our methods in *Introduction* and added some details in *Methods*.

Changes in the manuscript are shown below:

Introduction: To derive a win-win strategy that can outperform the current NDCs and is more acceptable by the parties, we specifically consider the responsibility (grandfathering and historical responsibility), capability (ability to pay) and equality (equal per capita allocation) to ensure each region's equitable burden sharing in response to climate mitigation (see Methods). An indicator named social welfare weight is constructed to represent the equity following effort-sharing methods. Each region would then determine its optimal emission pathways under its given social welfare weight through the C³IAM (see Methods for detailed process).

Methods: To obtain a win-win strategy, we introduce the effort-sharing approach into the cost-benefit analysis. The first step is to account the current NDCs and construct the policy-as-usual pathways of each region. Then we calculate the effort-sharing indicators of each region following four effort sharing principles, i.e. grandfathering, historical responsibility, ability to pay and equal per capita allocation, and combine

these indicators to determine the social welfare weights. The social welfare weights are subsequently applied to simulate the optimal emission pathways in the cost-benefit analysis. In the end, the win-win strategies can be identified through comparing the relative benefits and costs between optimal emission pathways and policy-as-usual pathways.

11. Where is the data from in general? Where are the equity allocations coming from? Where is the climate damage data coming from? Where is the emissions data coming from? Please mention these in a sentence explaining the reasons of your choice.

RESPONSE:

Thanks for your advice. The population, GDP, capital stock, and greenhouse gas emissions data are from UN, IMF, CDIAC and EDGAR. Future population and GDP data are from SSP2 (a more middle-of-the-road development pattern of Shared Socioeconomic Pathways). And the effort-sharing indicators are calculated based on the historical population, GDP and emissions data from previous sources. And the main parameters including the coefficients of climate damage in the C³IAM/EcOp model are from previous studies of Yang and Nordhaus. We have added a paragraph to describe the data source of our study in the end of *Methods*.

Changes in the manuscript are shown below:

Data Sources

The population, GDP, capital stock, and greenhouse gas emissions data used for effort-sharing indicators calculation and model estimation are from UN⁴³, IMF⁴⁴, CDIAC⁴⁵ and EDGAR⁴⁶. Future population and GDP data are from SSP2⁴⁷ (a more middle-of-the-road development pattern of Shared Socioeconomic Pathways). The main parameters including the coefficients of climate damage in the C³IAM/EcOp model are from Yang¹⁸ and Nordhaus¹⁹⁻²⁰.

12. Thanks a lot for your work. I believe that it can give interesting insights and discussions after more clarity is brought to help the reader what (s)he is looking at and how to understand the results.

Line 13 – I do not agree the factual benefits considered in the win-win strategy depend on a subjective attitude (pessimistic). I also would refrain from phrasing such as ‘there will always be’ unless it’s clearly proven in the text. It reads oddly after putting a vague condition on the attitude of countries for that statement. What is a pessimistic attitude? It is not defined in the text.

RESPONSE:

Thanks for your advice. The subjective attitude used here is to describe the possibilities of technological development. We changed the sentence to be “results lights that even the society experience slower decline of emission reduction technologies costs, the win-win strategy also could be found”. Please see *Abstract*.

Changes in the manuscript are shown below:

Results lights that even the society experience slower decline of emission reduction technologies costs, the win-win strategy also could be found.

13. My understanding is that the situation can be win-win based on the climate impacts linked with inaction and the cost of action. I would simply articulate the sentence around these facts.

RESPONSE:

Thanks for your advice. In this study, the win-win strategy is comparing the potential avoided climate impacts and the additional cost of action with the situation following current NDCs. We have clarified some descriptions in the main text.

Changes in the manuscript are shown below:

Under this framework, the win-win strategy means that the temperature-limiting goals could be reached with a net income compared with current NDCs. The net income means the cumulative benefits from the extra avoided climate impacts should exceed the extra mitigation costs compared to situation following current NDCs at both the global and national levels.

14. Please state that this win-win, is a win for some countries only compared to absolute inaction and its climate impacts (so absolute unfairness). However, it is

still a loss compared to potentially fairer allocations (e.g. burden sharing and equity). A ‘win’ is a comparison, and the point of comparison is not defined in the abstract.

RESPONSE:

Thanks for your advice. In our study, the win-win strategy is compared with current NDCs (policy-as-usual scenario). We introduce traditional burden sharing principles into cost-benefit analysis. All regions social welfare can be improved compared with their current NDCs in win-win strategy.

15. Line 15: The added value of this paper in terms national ‘allocations’ is very vague in the abstract. Most earlier studies agree that more contributions are needed from these countries and the Paris Agreement requires them to ratchet up. Please state your results more prominently.

RESPONSE:

Thanks for your advice. Aside from providing the mitigation targets for regions or countries, our win-win strategy could give the potential benefits from employing these mitigation pathways. We have added the related information in abstract. And we added a paragraph in discussion to compare the difference with current effort sharing approaches.

Changes in the manuscript are shown below:

Abstract: A ‘costly effort’ for all ratified parties to enhance current NDCs mitigation targets are provide, which can provide an economically improved action strategy.

Discussion: Compared with equitable allocations, our win-win strategies suggest a real ‘costly effort’ that a country could provide and meanwhile point out the net income a country could gain.

Page 2:

16. Line 21 Why does it indicate that we are at crossroads of governance? Also, the COP24 did not aim at discussing efforts. The ratcheting up of NDC is to be discussed by 2020 and COP26.

RESPONSE:

Thanks for your suggestions. We deleted the sentence “indicating that we at a crossroads of global climate governance” and changed the sentence “but they failed to reach consensus on further emissions cutting efforts” to be “meanwhile agreed to provide detailed information on their climate change mitigation targets”. We changed the sentence to be “Article 14 of the Paris Agreement requires the ratified parties to update their intended nationally determined contributions (INDCs, or NDCs hereafter) by 2020 since the current contributions are insufficient to achieve the 1.5°C or well below 2°C target”.

Changes in the manuscript are shown below:

In the 24th Conference of Parties in Katowice (COP24), the ratified parties settled on most of the difficult elements of the ‘rulebook’ for putting the Paris Agreement into practice¹, meanwhile agreed to provide detailed information on their climate change mitigation targets. Article 14 of the Paris Agreement requires the ratified parties to update their intended nationally determined contributions (INDCs, or NDCs hereafter) by 2020 since the current contributions are insufficient to achieve the 1.5°C or well below 2°C target². The enhanced ambitions will be assessed by the upcoming climate summit³ and are crucial to future mitigation pathways.

17. Line 31: The enhancement of NDC (not simple update) is required by the Paris Agreement, which is only confirmed in the Katowice package.

RESPONSE:

Thanks for your suggestions. We accepted and changed the sentence to be “Article 14 of the Paris Agreement requires the ratified parties to update their intended nationally determined contributions (INDCs, or NDCs hereafter) by 2020 since the current contributions are insufficient to achieve the 1.5°C or well below 2°C target”.

Changes in the manuscript are shown below:

Article 14 of the Paris Agreement requires the ratified parties to update their intended nationally determined contributions (INDCs, or NDCs hereafter) by 2020 since the current contributions are insufficient to achieve the 1.5°C or well below 2°C target². The enhanced ambitions will be assessed by the upcoming climate summit³ and are

crucial to future mitigation pathways.

18. Line 34: There is no 2°C target. At minimum, there is a ‘well below 2°C’. Please state what the upcoming summit is for people reading your article in the future.
Line 38 it’s ‘well below 2°C’

RESPONSE:

Thanks for your suggestions. The warming targets discussed in this study is in consistent with the Paris Agreement, which are well below 2°C and 1.5°C. We changed “2°C target” to be “well below 2°C target” in the whole text. Thanks again for your remind.

19. Line 42: Do you have a reference to back up the statement that policy makers find it harder to adopt the strategy? My experience is that the prefer actually budget allocation that they can use the way they want (and sometimes overshoot).

RESPONSE:

Thanks for your comments and suggestions. Although there is little reference to back up that statement directly, many actual examples can prove that the emissions reduction cost is also an important consideration for policymakers. For example, countries like USA and Canada are afraid that climate change policies would impact the increase of economy, so their attitudes toward climate change are very negative. USA even decide to withdraw the Paris Agreement. Based on such practical consideration, we maker further analysis of cost-benefit on the basis of burden sharing.

20. Line 46: these references derive allocations for all countries to my knowledge.

RESPONSE:

Thanks for your comments and suggestions. We delete the sentence “mainly for global major economics (e.g. G20, OECD and EU)”.

Page 3

21. Line 51: these references are from 1996 and 2008, do you have anything more recent?

RESPONSE:

Thanks for your advice. We have added some recent literatures. Please see *Reference* 19-22.

- Nordhaus, W. D. Economic aspects of global warming in a post-Copenhagen environment. *Proceedings of the National Academy of Sciences* 107, 11721-11726 (2010).
- Nordhaus, W. D. Revisiting the social cost of carbon. *Proceedings of the National Academy of Sciences* 114, 1518-1523 (2017).
- Nordhaus, W. D. Evolution of modeling of the economics of global warming: Changes in the DICE model, 1992–2017. *Climatic change* 148(4), 623-640 (2018)
- Nordhaus, W. D. Climate change: The ultimate challenge for Economics. *American Economic Review* 109(6), 1991-2014 (2019).

22. Line 53: The consequence of the lack of equity of this approach really need to be stated more clearly and deserve a few sentence. The win-win strategy allocates effort to mitigate emissions on the basis of the different exposures of countries to climate impacts. So the most vulnerable countries, often the poorest with limited adaptation capacity, will have the greatest burden to mitigate. It is deeply unfair and violates the Paris Agreement principle of equity (CBDR-RC principle and article 2), the recognition of the special vulnerability of countries and their priority to receive support (when the win-win is giving them more burden). These points need to be stated clearly or disproven if you disagree. While the win-win strategy can bring an interesting message, it cannot hide its deep unfairness and its incompatibility with the Paris Agreement and the UNFCCC.

RESPONSE:

Thanks for your advice. Based on our results, the average improved reduction targets compared with current NDCs of MAF and LAM to achieve 1.5°C target are 41-44% and 45-50% respectively, which are below the global average level (53-56%). To achieve the well below 2°C target, MAF and LAM should improve 19-36% and 12-38% based on current NDCs, which are also below the global average level (29-48%). As it were, the win-win strategy combines the effort-sharing approaches with cost-benefit analysis of emission mitigation, which is compatible with the principle of the Paris

Agreement and the UNFCCC. Although vulnerable countries have relative higher burden to mitigate compared with their specific national conditions. The premise is that these countries should experience quick reduction of low-carbon technology costs. Therefore, they need capital and technology transfer from developed countries, which is in consistent with the Article 11 of the Paris Agreement. For MAF and LAM, the cumulative emissions reduction costs in 2030 are 0.06 to 0.69 and 0.04 to 0.79 trillion dollars respectively. Developed countries should provide financial or technical support under the framework of the Paris Agreement. We added some descriptions about reduction costs of vulnerable countries in *Discussion*.

Changes in the manuscript are shown below:

Financial and technical support from developed countries are necessary for relative vulnerable countries to help them implement the win-win strategy. For regions like MAF and LAM, the cumulative emissions reduction costs in 2030 are 0.06 to 0.69 and 0.04 to 0.79 trillion dollars, respectively. They need capital and technology transfer from developed countries, which is in consistent with the Article 11 of the Paris Agreement.

23. Line 56: This is an overstatement. The article does not prove that consensus only relies on equity and a win situation over climate impacts. Many other factors, including political and diplomatic can prevent consensus. Please avoid such strong statements.

RESPONSE:

Thanks for your advice. We changed the sentence to “the consensus among ratified parties could more easily to be achieved when the updated emissions reduction schemes can bring stakeholders additional gains compared with the current NDCs and meet the equity requirements”.

24. Line 56: who is ‘we’? And this statement sound obvious. If there is a lack of will, obviously a stronger willingness would help.

RESPONSE:

Thanks for your comments. We changed this sentence into “what is needed for

meeting the 1.5°C or well below 2°C target is a widely beneficial strategy that can stimulate a stronger inherent willingness of the parties to participate the climate agreement”.

25. Line 62: do you have a reference? I strongly disagree, the willingness or not to tackle climate change is driven by many factors and the economics is only a small part. Many negative cost options are not implemented, which shows that it’s not the main drive.

RESPONSE:

Thanks for your suggestion. We changed this sentence to “Normally, one important factor of the parties’ willingness to reduce emissions is the associated costs and benefits”.

26. Line 64 why using ‘would’? I do not see a conditionality in that statement. I would use ‘will’.

RESPONSE:

Thanks for your suggestion. We changed it into “will”.

27. Line 65: why ‘would’?

RESPONSE:

Thanks for your suggestion. We changed it into “will”.

Page 4:

28. Line 70: please define what is 1.5°C and 2°C here. What likelihood? What overshoot? By when? Compared to what?

RESPONSE:

Thanks for your suggestion. In the main text, we have clarified that “As for warming,

we focus on the average atmospheric temperature change in 2100. If the temperature change in 2100 is less than 2°C and every region could gain compared with policy-as-usual, the optimal emission scenario is regarded as win-win scenario for 2°C, and the same for 1.5°C. The overshoots are not allowed for win-win scenario here. The warming targets in this study are in consistent with the Paris Agreement.”

29. Line 70: The statement implies that the only possible win-win strategy is the one suggested here, and that is compared to inaction. Please indicate that other win-win strategies could exist. Many argue that equitable solutions would be beneficial for all. Some could argue that the status quo is by definition the win-win strategy, all factors accounted for (not just win over climate impacts) as it reflects the choices of governments. Some countries can consider a relative win over a given competitor more important than an absolute win (and could accept an absolute loss as relative win).

RESPONSE:

Thanks for your suggestion. We agree that other win-win scenarios may exist from the viewpoint of political or diplomatic thinking. Those kinds of win-win are not discussed in our study. Here our focus is to discuss the win-win situation under the principle of economic benefits and fairness of each country. We have modified our descriptions to clarify our focus. We also mentioned this as a limitation of our study, which is shown in the end of *Discussion*: “In addition, the win-win strategy defined here is under the principle of economic benefits with the consideration of fairness for each country. Successful implementation of the win-win strategy is premised on the improved understanding on climate damages and the breakthroughs of low-carbon technologies. Furthermore, in addition to economic benefits, factors such as political attitudes, diplomacy policies and environmental capacity, are thought to be important determinants on the climate mitigation actions of each country. This can be further discussed in the future study”. Thanks again for your valuable suggestions.

30. Line 74: The choice of the term ‘will’ reflect the legal language of the Paris Agreement but is here understood as a prediction that is not backed up (and actually unlikely, research shows that many countries are not on track to achieve their

NDC:

<https://www.sciencedirect.com/science/article/pii/S030142151830750X>).

RESPONSE:

Thanks for your suggestion. We changed the sentence to be “Based on the Paris Agreement, each ratified party are supposed to fulfill domestic mitigation measures to realize its NDC²⁷.”

31. Line 75: which studies? Please provide references.

RESPONSE:

Thanks for your suggestion. We clarified three references here to support this point.

- **Ref.2:** United Nations Environment Programme (UNEP). The Emissions Gap Report 2018. An annual assessment tracking climate policy action over the past six years, which provided the basis for the analysis presented in this Perspective. http://uneplive.unep.org/media/docs/theme/13/EGR_2015_301115_lores.pdf (2018).
- **Ref.4:** Rogelj, J., et al. Paris Agreement climate proposals need a boost to keep warming well below 2°C. Nature 534, 631-639 (2016).
- **Ref.28:** Zoi Vrontisi et al. Enhancing global climate policy ambition towards a 1.5°C stabilization: a short-term multi-model assessment. Environ. Res. Lett. 13 044039 (2018).

32. Line 78: Are the UNFCCC and the Paris Agreement not global cooperation strategies?

RESPONSE:

Thanks for your suggestion. Since NDCs are submitted by each ratified party, we regard it as a non-cooperative scenario. Current studies showed that cooperative emissions reduction mechanism can promote economic efficiency (Ref.18, Ref. 28-30). Therefore, we simulate a global cooperative situation to obtain the optimal emission trajectories toward the warming targets. To make it more clearly, we changed the sentence to be “an updated NDC strategy in response to global climate change is appealed” in the text.

Changes in the manuscript are shown below:

Hence, an updated NDC strategy in response to global climate change is appealed¹⁸,

33. Line 80: Suggestion: the next two sentences could be inverted to reflect chronological order.

RESPONSE:

Thanks for your suggestion. We have modified these two sentences.

Page 5

34. Before stating results, there need to be a description of what the scenarios are made of and what are the assumptions. Beyond the absence of discussion of the parameters, the text does not event introduce what the scenarios entail. They are just called win-win, and consider climate impacts, costs and equity somehow. There are no more details at this point. The reader needs to understand that before reading about the results.

RESPONSE:

Thanks for your suggestion. We add a paragraph to describe the scenario before starting results. Please see the first paragraph of *win-win strategies identification*: “ The scenario setting considers four aspects, including warming limits targets, low-carbon technology costs, climate damage and equity principles. As for warming, we focus on the average atmospheric temperature change in 2100. If the temperature change in 2100 is less than 2°C and every region could gain compared with policy-as-usual, the optimal emission scenario is regarded as win-win scenario for 2°C, and the same for 1.5°C. The overshoots are not allowed for win-win scenario here. The warming targets in this study are in consistent with the Paris Agreement. Meanwhile, different levels of low-carbon technology costs and climate damage to reflect the uncertainty of low-carbon technology development and climate risk are considered. According to the uncertainty level of climate damage and technology development, there will be a package of specific optimal emission scenarios (Fig. 1a). The changes of low-carbon technology costs and climate damage are set in accordance with previous studies^{19, 20, 26, 33}. We define the level of climate damage by using the ratio of economic damage in GDP. The enlarged coefficients (increase times of climate damage shown in Fig. 1a) of damage function in the model is used to characterize different levels of climate damage. The values reported in the reference

19 is set as the reference level, which show a climate damage being approximately 1.6% of GDP at a 2.62°C warming in 2100. With a given temperature rise, the economic damages are assumed to be different times as large as the reference level. Specially, Supplementary Fig. 1 shows the changes in economic damage under different uncertainties with temperature rise for each region. In this study, we define high, medium and low level of climate damage, which is corresponding to the increase times of climate damage coefficients used in the damage function being 4 to 5 times, 2 to 4 times and less than 2 times, respectively. In addition, we define three levels of technological development, i.e., slow development with low decline rate of low-carbon technology costs being less than 10% every five years, medium development with medium decline rate of low-carbon technology costs being 10% to 30% every five years, and rapid development with high decline rate of low-carbon technology costs being 30% to 40% every five years. The capability and responsibility of developing and vulnerable countries are fully considered by introducing equity principles. The social welfare weight of each region are shown in Supplementary Table 1.”

35. Line 89: I am not sure to understand. The uncertainty does not result in more scenarios. It just reflects the uncertainty of a given set of scenarios. While the results may yield a greater range, it does not imply that there is a broader ‘package of specific scenarios’.

RESPONSE:

Thanks for your comments. We define the level of climate damage by using the ratio of economic damage in GDP. The enlarged coefficients (increase times of climate damage shown in Fig. 1a) of damage function in the model is used to characterize different levels of climate damage. The values reported in the reference 19 is set as the reference level, which show a climate damage being approximately 1.6% of GDP at a 2.62°C warming in 2100. With a given temperature rise, the economic damages are assumed to be different times as large as the reference level. Following Ref. 26, we set the upper limit of climate damage. According to Ref. 20, we define the decline rate of low-carbon technology cost keeps constant as base year 2015. With the combination of different uncertainty level of climate damage and technology development, a set of optimal emissions scenarios are obtained. In all scenarios, those

who are consistent with the well below 2°C or 1.5°C targets are chosen as the win-win scenarios (the percentage could reach 64.2%).

36. Line 92: Please briefly explain which uncertainties are covered and how.

RESPONSE:

Thanks for your suggestions. In our study, the uncertainties include climate damage and low-carbon technology development. The scenario matrix was constructed based on different level of climate damage and the decline rate of low-carbon technology cost. We obtained these uncertainties by adjusting the coefficients of climate damage function and emissions reduction cost equation (Ref. 19, Ref. 20, Ref. 26). We have added one paragraph in the section of “*win-win strategies identification*”.

37. Line 92: as a percentage of what? Of all the scenarios that the paper quantifies as win-win? Based on what range of assumptions? This is unclear. Please start by a paragraph on your model, its goals and modelling assumptions. At this stage, the reader does not know what the scenarios are made of.

RESPONSE:

Thanks for your comments. In all the optimal emission scenarios (Fig. 1a), the percentage of win-win scenarios consistent with well below 2°C and 1.5°C targets could reach 64.2%, the majority of which are under conditions of medium to high (i.e., equal or greater than 20%) decline rate of low-carbon technology costs. We have added the above information into the main text.

The scenario setting considers four aspects, including warming limits targets, low-carbon technology costs, climate damage and equity principles. As for warming, we focus on the average atmospheric temperature change in 2100. If the temperature change in 2100 is less than 2°C and every region could gains compared with policy-as-usual, the optimal emission scenario is regarded as win-win scenario for 2°C, and the same for 1.5°C. The overshoots are not allowed for win-win scenario here. The warming targets in this study are in consistent with the Paris Agreement. Meanwhile, different levels of low-carbon technology costs and climate damage to reflect the uncertainty of low-carbon technology development and climate risk are considered. According to the uncertainty level of climate damage and technology

development, there will be a package of specific optimal emission scenarios (Fig. 1a). The changes of low-carbon technology costs and climate damage are set in accordance with previous studies^{19, 20, 26, 33}. We define the level of climate damage by using the ratio of economic damage in GDP. The enlarged coefficients (increase times of climate damage shown in Fig. 1a) of damage function in the model is used to characterize different levels of climate damage. The values reported in the reference 19 is set as the reference level, which show a climate damage being approximately 1.6% of GDP at a 2.62°C warming in 2100. With a given temperature rise, the economic damages are assumed to be different times as large as the reference level. Specially, Supplementary Fig. 1 shows the changes in economic damage under different uncertainties with temperature rise for each region. In this study, we define high, medium and low level of climate damage, which is corresponding to the increase times of climate damage coefficients used in the damage function being 4 to 5 times, 2 to 4 times and less than 2 times, respectively. In addition, we define three levels of technological development, i.e., slow development with low decline rate of low-carbon technology costs being less than 10% every five years, medium development with medium decline rate of low-carbon technology costs being 10% to 30% every five years, and rapid development with high decline rate of low-carbon technology costs being 30% to 40% every five years.

The win-win scenarios are selected from the optimal emission pathways under different uncertainty levels of climate damages and technology development. To simulate the optimal mitigation path, we apply a revised version of the global multiregional economic optimum growth model (C³IAM/EcOp), which is a submodule of the C³IAM³¹; it is established based on the theory of optimal economic growth and consists of an economic module and a climate module. The economic module of C³IAM/EcOp is a modified version of a standard neoclassical optimal growth model. The climate module of C³IAM/EcOp links GHG emissions to concentration, radiative forcing and temperature. In addition, C³IAM/EcOp take into account the interaction between economic module and climate module through introducing climate damage function and abatement cost function. We have added these details into *Method*.

Changes in the manuscript are shown below:

Introduction: It is found that in all the optimal emission scenarios (Fig. 1a), the percentage of win-win scenarios consistent with well below 2°C and 1.5°C targets could reach 64.2%, the majority of which are under conditions of medium to high (i.e., equal or greater than 20%) decline rate of low-carbon technology costs.

Methods: To simulate the optimal mitigation path, we apply a revised version of the global multiregional economic optimum growth model (C³IAM/EcOp), which is a submodule of the C³IAM³¹; it is established based on the theory of optimal economic growth and consists of an economic module and a climate module. The economic module of C³IAM/EcOp is a modified version of a standard neoclassical optimal growth model. The climate module of C³IAM/EcOp links GHG emissions to concentration, radiative forcing and temperature. In addition, C³IAM/EcOp take into account the interaction between economic module and climate module through introducing climate damage function and abatement cost function.

38. Line 95: ‘if we would’ is not grammatically correct. I suggest ‘if society does not have a slower technology development,...’

RESPONSE:

Thanks for your suggestion. We changed the sentence to “this implies that if the society would experience a medium to rapid technology development (the decline rate every five years of the low-carbon technology costs can reach 10% or more), the win-win strategy could always be found” in the text.

39. Line 95 slower than what? This section is very unclear and comparison are left without objects.

RESPONSE:

Thanks for your comments. “Slower” means the decline rate every five years of the low-carbon technology costs lower than 10%. Since this kind of expression is unclear, we deleted this sentence and changed it to be “this implies that if the society would experience a medium to rapid technology development (the decline rate every five years of the low-carbon technology costs can reach 10% or more), the win-win strategy could always be found.”

40. Line 97: please define what are these medium to high damages and low costs. These labels are introduced in the article but do not have a universal definition. The article needs to explain what is understood by that.

RESPONSE:

Thanks for your comments. “Medium to high damage” means more than twice relative to the benchmark level. “Low costs” refers to the decline rate of low-carbon technology being 20% every five years. We changed this sentence to be “only the scenarios under the conditions of more than twice climate damage relative to the reference level and more than 20% decline rate of low-carbon technology costs, accounting for 27.2% in the all optimal emissions scenarios, can achieve the 1.5°C target”. We also added the definition of these degree words in the text (see the first paragraph in the section of “*win-win strategies identification*”).

41. Line 102: The pathway does not seem to show that it is challenging. The pathway confirms that NDCs are not in line with 2°C.

RESPONSE:

Thanks for your comments. We changed this sentence to “the pathway under the policy as usual scenario shows that current NDCs are not in line with the well below 2°C target” in the text.

42. Line 103: This section needs to come before the Results section (see my prior comments on the lack of explanation on the scenarios). Again, please explain what is understood by ‘2°C/1.5°C’. Is that a 50% chance? Through the century or in 2100 with possible overshoot?

RESPONSE:

Thanks for your suggestion. “2°C/1.5°C” means “2°C or 1.5°C”, we changed it into “2°C or 1.5°C”. And we moved the win-win strategies identification before the *Results* section.

43. Line 105: what are the characteristics of the different levels of climate damages? Could there be a table summarizing the main differences?

RESPONSE:

Thanks for your suggestion. The increase of climate damage (times) means different climate damage assumptions relative to benchmark climate damage function and the amount of global climate damage of policy-as-usual scenario in 2100. We define the level of climate damage by using the ratio of economic damage in GDP. The enlarged coefficients (increase times of climate damage shown in Fig. 1a) of damage function in the model is used to characterize different levels of climate damage. The values reported in the reference 19 is set as the reference level, which show a climate damage being approximately 1.6% of GDP at a 2.62°C warming in 2100. With a given temperature rise, the economic damages are assumed to be different times as large as the reference level. Specially, Supplementary Fig. 1 shows the changes in economic damage under different uncertainties with temperature rise for each region. We added a figure in Supplementary material to show the climate damage associated with different temperature rise of each region. And we added a part to describe how the uncertainty of climate damage are simulated in our model before **Results**. Thanks again for your valuable advice.

Changes in the manuscript are shown below:

We define the level of climate damage by using the ratio of economic damage in GDP. The enlarged coefficients (increase times of climate damage shown in Fig. 1a) of damage function in the model is used to characterize different levels of climate damage. The values reported in the reference 19 is set as the reference level, which show a climate damage being approximately 1.6% of GDP at a 2.62°C warming in 2100. With a given temperature rise, the economic damages are assumed to be different times as large as the reference level. Specially, Supplementary Fig. 1 shows the changes in economic damage under different uncertainties with temperature rise for each region.

Supplementary Figure 1|The climate damage associated with different temperature rise.

44. Line 198: 90% of what? Of their current levels? Of a reduction compared to BaU? Compared to their 2030 NDC emissions? Please be more specific.

RESPONSE:

Thanks for your comments. It's 90% of their current NDCs. And we have mentioned in the beginning of this sentence.

Changes in the manuscript are shown below:

To reach the 1.5°C target, all parties need to further reduce their emissions in 2030 compared with the current NDCs, and much more improvements need to be made by the following regions, including Japan (on average 90% extra emissions reduction

required for WW1.5s), EU (84% extra), the USA (63% extra), India (63% extra), Russia Federation (59% extra), EES(58% extra) and China (55% extra) (Fig. 4b, Supplementary Figures 4 and 6).

45. Line 202: a guide may be an overstatement. Information may be more appropriate.

RESPONSE:

Thanks for your suggestion. We changed “guide” into “information”.

Changes in the manuscript are shown below:

The updated NDCs in all win-win scenarios for achieving the 2°C and 1.5°C targets are given in Supplementary Tables 2-3, respectively, providing a straightforward information for the next round of pledging in the post-Katowice Climate Package.

46. Line 203, Fig4a: given that the policy as usual path is the same in each graph, there might be better and clearer representations that help comparing the different scenarios. Perhaps bundle in 2 graphs for 1.5°C and 2°C.

RESPONSE:

Thanks for your suggestion. We accepted your advice and bundled fig.4a in 2 graphs which show the GHG emissions gap between win-win strategy and current NDCs.

Changes in the manuscript are shown below:

Fig. 4|GHG emission gaps between policy-as-usual scenario following the current NDCs and win-win scenarios at the global, regional and national levels.

Page 12

47. Line 215: what would a relaxing of policies imply? What does it mean to increase the stringency? If the article is not discussing the policies, it may be better to only discuss the needed emissions trends. The whole paragraph is unclear.

RESPONSE:

Thanks for your suggestion. Here, we think that “To realize the win-win strategy, effective policies are always required. The marginal abatement cost (MAC) is an important factor that can influence the stringency of climate change policy. In order to improve the feasibility, the stringency of climate change policies should be consistent with the corresponding marginal abatement cost shown in Fig. 5. Compared with

other studies containing MAC analysis, our results are within the existing interval. Moreover, higher marginal costs do not necessarily imply higher total policy costs³⁵. Thus, the win-win strategies are feasible from this point of view.”

Page 13:

48. Line 231: The PA is not really about climate actions. It is more about.

Line 232: Please explain how it's a continued success? The first test will really be at the COP26 with the outcome of the ratcheting up process. Also, how can a success 'require' countries to do something?

RESPONSE:

Thanks for your comments. Here is “Its continued success”, means that if the Paris Agreement want to be effective in the global climate change governance, ratified parties need to make more reduction efforts. Both the update and stock will be carried out under the framework of the Paris Agreement, which can be regarded as the continuation of the Paris Agreement. We have deleted the sentence “the Paris Agreement is widely viewed as the best understanding of the climate actions that countries intended to pursue after 2020” and deleted “its continued success”. This part has been reorganized in the text.

Changes in the manuscript are shown below:

The post-Katowice climate governance requires parties to undertake greater emissions mitigation in the subsequent round of pledging for achieving warming targets³⁴. In line with this, our study presents win-win strategies for updating current NDCs to achieve a well below 2°C or 1.5°C target while satisfying the ratified parties. All Parties would gain from raising their target and align with a 1.5°C or well below 2°C objective.

49. Line 240: Given the win-win narrative, I would phrase this as ‘Most Parties would gain from raising their target and align with a 2°C objective’.

RESPONSE:

Thanks for your suggestion. We accepted your advice and reorganized this sentence. Since in win-win strategies, all parties would have net income. So we changed the

sentence to be “All Parties would gain from raising their target and align with a 1.5°C or well below 2°C objective”.

50. Line 246: again, it is important to explain that the net positive income is compared to inaction. And that the situation would be very different fi compared to a more equitable approach, in particular for developing countries. Likewise, I would rather speak of avoided damages than net incomes.

RESPONSE:

Thanks for your suggestion. Compared with current NDCs targets, the win-win strategy would let all parties obtain the cumulative positive net income. We have added the clear definition of net income in the text.

Changes in the manuscript are shown below:

Under this framework, the win-win strategy means that the temperature-limiting goals could be reached with a net income compared with current NDCs. The net income means the cumulative benefits from the extra avoided climate impacts should exceed the extra mitigation costs compared to situation following current NDCs at both the global and national levels.

51. Line 253: how do you define what is acceptable?

RESPONS:

Thanks for your comments. We deleted this word and modified the sentence as “despite the negative net income for some countries at the early stage, the amount is less than 0.4% of annual GDP”.

52. Line 254: it may be best to avoid overstatements. Science did not conclude that societies would be destroyed. Also, destruction is presented as the alternative to your win-win scenario. It comes across as threatening. The advantages of your scenario should speak for themselves.

RESPONSE:

Thanks for your comments. We changed this sentence to be “therefore, to avoid the

threatening climate damage, all the ratified parties in the world are encouraged to take the climate mitigation actions following our win-win strategy, and each of them could obtain 0.37- 5.88% GDP gains in 2100”.

Page 15

53. Line 260: It may be better to state that you have set these targets, rather than you can set them. Perhaps provide them in a separate table.

RESPONSE:

Thanks for your comments. We have set the updated emissions reduction target for each country under well below 2°C and 1.5°C targets, shown in supplement materials. In order to make it more clear, we added a sentence in the main text: “The exact emissions reduction targets for each country are shown in Supplementary Tables 2 and 3” in the text”. Thanks again for your suggestion.

Changes in the manuscript are shown below:

We have set the emission reduction target for each country following the derived win-win strategy that is consistent with public perceived climate damage on the socio-economic and earth system. The exact emissions reduction targets for each country are shown in Supplementary Tables 2 and 3.

54. The last paragraph of the discussion could come as a second paragraph. It reads oddly as a concluding statement and ends with numbers.

RESPONSE:

Thanks for your suggestion. We have added a paragraph about future research prospects and research limitation in the end of *Discussion*.

Changes in the manuscript are shown below:

Although this study has contributed to improving the current NDCs by considering both the Paris Agreement targets and economic benefits for each ratified party, some limitations are left to the future work. For example, in recent years, many adaptation strategies have been proposed. More work is needed to assess the adaptation potentials and costs for managing climate change risks in C³IAM model framework.

In addition, the win-win strategy defined here is under the principle of economic benefits with the consideration of fairness for each country. Successful implementation of the win-win strategy is premised on the improved understanding on climate damages and the breakthroughs of low-carbon technologies. Furthermore, in addition to economic benefits, factors such as political attitudes, diplomacy policies and environmental capacity, are thought to be important determinants on the climate mitigation actions of each country. This can be further discussed in the future study.

55. Methods: Please introduce a general sequence of the steps before going into details.

RESPONSE:

Thank you for your advice. We have added a paragraph at the beginning of *Methods*.

Changes in the manuscript are shown below:

To obtain a win-win strategy, we introduce the effort-sharing approach into the cost-benefit analysis. The first step is to account the current NDCs and construct the policy-as-usual pathways of each region. Then we calculate the effort-sharing indicators of each region following four effort sharing principles, i.e. grandfathering, historical responsibility, ability to pay and equal per capita allocation, and combine these indicators to determine the social welfare weights. The social welfare weights are subsequently applied to simulate the optimal emission pathways in the cost-benefit analysis. In the end, the win-win strategies can be identified through comparing the relative benefits and costs between optimal emission pathways and policy-as-usual pathways.

56. Line 389: 'in consistent with' reads oddly.

RESPONSE:

Thank you for your advice. We changed it into "Based on".

Changes in the manuscript are shown below:

Based on our definition of win-win strategy, we could find the win-win strategies from all optimal emissions pathways under different uncertainties of climate damage

and low-carbon technology cost.

■ To Reviewer #2's comments

Thank you for giving me the chance to review manuscript NCOMMS-19-28971
I find it well-written yet with numerous potential (mostly minor and a few major) improvements.

RESPONSE:

Thanks to the reviewer for the positive comments. We tried our best to revise our work and addressed all comments as follows.

1. Although, to begin with, I am not very comfortable with the authors' non-disclosure of (socioeconomic/input) data, the employed model (C3IAM) is referred to as a model established for evaluating INDCs based on the SSPs. Are we to assume a specific SSP is used for this analysis? If so, how does that scenario relate to the technological and marginal abatement costs assumed here?

RESPONSE:

Thanks for your comments. In this study, future population and GDP data are from SSP2 (a more middle-of-the-road development pattern of Share Socioeconomic Pathways). This scenario could not impact our model's assumption about the costs of emissions reduction. We have clarified the data sources in *Methods*.

Changes in the manuscript are shown below:

Data Sources

The population, GDP, capital stock, and greenhouse gas emissions data used for effort-sharing indicators calculation and model estimation are from UN⁴¹, IMF⁴², CDIAC⁴³ and EDGAR⁴⁴. Future population and GDP data are from SSP2⁴⁵ (a more middle-of-the-road development pattern of Share Socioeconomic Pathways). The main parameters including the coefficients of climate damage in the C³IAM/EcOp model are from Yang¹⁸ and Nordhaus¹⁹⁻²⁰.

2. Regarding the determination of the cooperative scheme, the EcOp module of the C3IAM model is selected. As a standard neoclassical optimal growth model, it calculates parties' willingness to reduce emissions as a function of associated costs and benefits. I am trying to understand what the regional/national benefits for each Party are: if these benefits (climate damage aversion) are at the Party scale, then Parties choose their emissions reductions based on how vulnerable they are? If so, this is not in line with the Paris Agreement equity principles.

RESPONSE:

Thank you for your suggestion. Compared with other cost-benefit research, we have considered the effort-sharing by assigning different responsibilities to different regions in our analysis. Through introducing responsibility (grandfathering and historical responsibility), capability (ability to pay) and equity (equal per capita allocation) into cost-benefit analysis, the social welfare weights of each region from effort-sharing methods is constructed. After that, each region would determine the optimal mitigation effort under the given social welfare weights through the C³IAM model. In this way, all regions' social welfare can be improved compared with current NDCs. The principles (responsibility, capability and equity) we used in this study can reflect the CBDR-RC principle used worldwide. And our results showed that to achieve no matter a well below 2°C or 1.5°C target, more contributions need to be made by Japan, EU, the USA, India, Russia, the Eastern European and Commonwealth of Independent States and China. Vulnerable countries mainly located in the Middle East and Africa (MAF) and Latin America (LAM). Based on our results, the average improved reduction targets compared with current NDCs of MAF and LAM to achieve 1.5°C target are 41-44% and 45-50% respectively, which are below the global average level (53-56%). To achieve the well below 2°C target, MAF and LAM should improve 19-36% and 12-38% based on current NDCs, which are also below the global average level (29-48%). As it were, the win-win strategy combines the effort-sharing approaches with cost-benefit analysis of emission mitigation, which is compatible with the principle of the Paris Agreement and the UNFCCC. To make these points more clear, we added some details of our methods and rewrite the description in the last paragraph of *Introduction* and the first paragraph of *Methods*.

Changes in the manuscript are shown below:

Introduction: To derive a win-win strategy that can outperform the current NDCs and is more acceptable by the parties, we specifically consider the responsibility (grandfathering and historical responsibility), capability (ability to pay) and equality (equal per capita allocation) to ensure each region's equitable burden sharing in response to climate mitigation (see Methods). An indicator named social welfare weight is constructed to represent the equity following effort-sharing methods. Each region would then determine its optimal emission pathways under its given social welfare weight through the C3IAM (see Methods for detailed process). Before doing so, since the existing NDCs submitted are very chaotic in terms of their definition and coverage³², we first develop a uniform accounting criterion for the current NDCs and construct the policy-as-usual pathways (see Methods). Through comparing the benefits and costs between optimal emission pathways and policy-as-usual pathways, the optimal emission pathways that could realize the temperature warming limits and bring net incomes to every region are derived (see Methods).

Methods: To obtain a win-win strategy, we introduce the effort-sharing approach into the cost-benefit analysis. The first step is to account the current NDCs and construct the policy-as-usual pathways of each region. Then we calculate the effort-sharing indicators of each region following four effort sharing principles, i.e. grandfathering, historical responsibility, ability to pay and equal per capita allocation, and combine these indicators to determine the social welfare weights. The social welfare weights are subsequently applied to simulate the optimal emission pathways in the cost-benefit analysis. In the end, the win-win strategies can be identified through comparing the relative benefits and costs between optimal emission pathways and policy-as-usual pathways.

3. Nordhaus' 2010 paper is referred to for representing climate damage in terms of (global) GDP in 2100; is this estimate up to date?

RESPONSE:

Thanks for your comments. This study considers the uncertainty level of climate damage. Specifically, we used the Nordhaus' 2010 paper as the reference level of climate damage and used the Burke' 2015 paper as the upper limit level of climate damage. The climate damage function in Nordhaus's recent research is the same with the one in 2010's paper.

4. The authors select nine ‘win-win’ scenarios, relating to levels of technological costs and climate damage (and assuming given stringency), which help explore future pledges targets, but (aside from technological breakthroughs) there is no discussion of policy and feasibility, hence no discussion of how realistic these scenarios can be.

RESPONSE:

Thanks for your comments. Based on our analysis, the updated emissions reduction target for each country under 2°C and 1.5°C targets is obtained, shown in supplement materials. We also output the results of marginal abatement cost (MAC) for achieving the temperature limit goals. MAC is an important factor that can influence the climate policy efforts. The stringency of emission reduction policies should be consistent with the corresponding MAC. We have provided each region’s MAC in fig.5. Compared with other studies containing MAC analysis, our results are within the existing interval. Moreover, higher marginal costs do not necessarily imply higher total policy costs³⁵. Thus, the win-win strategies are feasible from this point of view. Thanks again for your suggestion. This is where we can going to expand our model in the future.

5. How is stringency defined? Is it (through abatement costs) part of the scenario inputs for the win-win trajectories or part of the findings?

RESPONSE:

Thanks for your suggestion. The marginal abatement cost (MAC) is an important factor that can influence the stringency of climate change policy. Thus, we apply the MAC to indicate the implementation efforts of policy. MAC is a part of our findings. Here increase the stringency means intensify the implementation of policy. The strength of policy is determined by the cost of low-carbon technology and the rate of emissions reduction. Among them, the decline rate of low-carbon technology is the scenario input, the emissions reduction rate is the scenario output. We have reorganized the language of this paragraph.

Changes in the manuscript are shown below:

To realize the win-win strategy, effective policies are always required. The marginal abatement cost (MAC) is an important factor that can influence the stringency of

climate change policy. In order to improve the feasibility, the stringency of climate change policies should be consistent with the corresponding marginal abatement cost shown in Fig. 5. Compared with other studies containing MAC analysis, our results are within the existing interval. Moreover, higher marginal costs do not necessarily imply higher total policy costs³⁵. Thus, the win-win strategies are feasible from this point of view. All parties need to start by tightening their policies, and most of them can relax their stringency at the latter stage. The timings for policy relaxation differ among regions.

6. Authors mention the use of game theory, but there is no discussion on it, unless it is attributed to the fact that targets for multiple agents (countries) are calculated, based on costs determined at the national level while (climate) benefits at the global.

RESPONSE:

Thanks for your comments. In this study, we simulate a global cooperative situation with the consideration of equitable effort sharing indicated by social welfare weights of each region. It's not a game theory, so we have deleted this sentence.

7. I was most confused in lines 96-98: "scenarios under the conditions of medium to high climate damage and low technological costs, accounting for 27.2%, can achieve the 1.5°C target". What exactly is 27.2%, and as percentage of what? What does the overall sentence mean, given that authors specify scenarios of low to high climate damage? Does it mean that if climate damage is low, the 1.5°C cannot be achieved? What about WW1.5L (line 110)?

RESPONSE:

Thanks for your comments. It's 27.2% of all optimal emissions pathways. If climate damage is low, the temperature change of optimal emission scenarios would not achieve 1.5° C target. And the WW1.5L means the win-win scenarios with relative low level of climate damage in all WW1.5s, which is different with the matrix. To avoid confusion, we changed the previous names of each win-win strategy to WW 2.0 A, WW 2.0 B, WW 2.0 C, WW 2.0 D, WW 1.5 A, WW 1.5 B, WW 1.5 C, WW 1.5 D and WW 1.5 E.

Changes in the manuscript are shown below:

We select nine representative win-win scenarios that have higher welfare for further analysis; of the nine, four would reach the well below 2°C target under different levels of climate damage (named WW2.0s, including WW2.0 A, WW2.0 B, WW2.0 C and WW2.0 D) and five would reach the 1.5°C target (named WW1.5s, including WW1.5 A, WW1.5 B, WW1.5 C, WW1.5 D, and WW1.5 E) (Figs. 1a, c).

8. Lines 113-114: although Figure 1b shows negative emissions as early as 2035, in the 1.5°C scenario. From the description of Figure 1(a), I am confused as to whether, for example, a 10% decline rate means that the costs drop by 10% every five years, or that the decline rate of the costs increases by 10% every five years.

RESPONSE:

Thanks for your comments. Because the reference level (i.e., 0) means the decline rate of low-carbon technology cost keeps constant as the base year 2015, a 10% decline rate means that the costs drop by 10% every five years. We have clarified the meanings in the first paragraph in Section of “*Win-win strategies identification*”.

9. From the discussion of the method for accounting current NDCs and constructing the policy as usual scenario, do I understand correctly that NDC projections used may be largely incorrect in respect to what is intended in the actual NDCs?

RESPONSE:

Thanks for your comments. To project the actual NDC emissions amount is a complicated process. Based on previous study, to overcome the difficulties of accounting for emissions in the BaU scenario and to draw the NDC path for each ratified party, we use CEEP-S accounting model to provide a transparent projection of future emissions in the BaU scenario by considering uncertain economic development (GDP) and dynamic emission intensity (GHG emissions per unit GDP); then, the NDCs are further quantified based on the BaU emissions. We add up the NDCs of each country to get the regional NDCs, and then construct the policy as usual scenarios (PAU) to find the win-win strategies for current NDCs. The future emissions reduction level of PAU scenario is assumed based on the current NDCs. As

for the regions whose NDCs is lower than BaU emissions, we could obtain the mitigation rate of the target year and construct the policy as usual emissions pathway by assuming the same mitigation rate with NDCs during the whole model period. As for the regions whose NDCs is higher than BaU emissions, we construct their policy as usual emissions pathway by using their BaU emissions pathways so as to avoid the situation that some countries actually do more than they have committed. In this sense, the policy as usual scenario is more close to the real situation of each country.

10. The manuscript would certainly benefit from a comparative analysis discussion on relevant studies.

RESPONSE:

Thanks for your suggestion. We add a paragraph in the *Discussion* section to imply the difference with existing literature.

Changes in the manuscript are shown below:

Compared to existing effort-sharing studies, for example Ref. 37 which compared over 40 studies that analyze future GHG emissions allowances for different regions based on a wide range of effort-sharing approaches and long-term concentration stabilization levels, the range of our results is close to that of other studies. For the Organization for Economic Co-operation and Development (OECD) regions (1990 classification) and LAM, proposals in the existing studies (negative allowances in 2030 of -75% to -37% change from 2010 level) and in our study (-66% to -37% in the same year) are close. However, the proposals for Economies in Transition (EIT), ASIA and MAF are more stringent than previous studies. When combining different effort-sharing approaches and cost-benefit analysis, countries could benefit more or less from avoiding the potential climate impacts through more stringent mitigation efforts. Compared with equitable allocations, our win-win strategies provide a real ‘costly effort’ that a country could provide and point out the net income a country could gain.

11. The existing Discussion is more of a Conclusion section, and even so lacks a discussion of limitations and a framing of how the policy prescriptions should be

considered. The finding that “higher climate damage, faster technological growth presents opportunities for more benefits for all parties” is rather trivial, and the last paragraph has been written hastily and not as informative.

RESPONSE:

Thanks for your suggestion and comments. We have rewrite the last part of *Discussion* and add some research limitations. We will keep enriching our model and simulating specific policies in future work.

Changes in the manuscript are shown below:

Although this study has contributed to improving the current NDCs by considering both the Paris Agreement targets and economic benefits for each ratified party, some limitations are left to the future work. For example, in recent years, many adaptation strategies have been proposed. More work is needed to assess the adaptation potentials and costs for managing climate change risks in C³IAM model framework. In addition, the win-win strategy defined here is under the principle of economic benefits with the consideration of fairness for each country. Successful implementation of the win-win strategy is premised on the improved understanding on climate damages and the breakthroughs of low-carbon technologies. Furthermore, in addition to economic benefits, factors such as political attitudes, diplomacy policies and environmental capacity, are thought to be important determinants on the climate mitigation actions of each country. This can be further discussed in the future study.

12. On a side note, the figures are well elaborated.

RESPONSE:

Thank you very much.

13. To sum up, I find this manuscript to be worth publishing after some clarifications and revisions.

RESPONSE:

Thank you very much.

Reviewers' comments:

Reviewer #1 (Remarks to the Author):

Dear Authors, thank you very much for your revisions.

I could not re-read the manuscript and focused on the response to referees.

My main point remains the same, and it remains unaddressed. You wrote: "Under this framework, the win-win strategy means that the temperature-limiting goals could be reached with a net income compared with current NDCs. The net income means the cumulative benefits from the extra avoided climate impacts should exceed the extra mitigation costs compared to situation following current NDCs at both the global and national levels."

Again, this cannot be as an acceptable solution. Allocating effort based on exposure to climate contradicts the Paris Agreement. This is not stated in the text despite my numerous mentions of this problem. Again, this article could reframe the analysis, keeping the results, to make it informative. It would be much more useful than shaping it as a deeply unfair suggestion for action. An interesting analysis could be framed to show how some countries would lose more welfare by inaction, compared to 1.5°C/2°C commensurate action. I have suggested that in previous iterations of revisions.

Another issue with the current framing is to study additional efforts compared to current NDCs. This disfavors countries that provided ambitious and costly NDCs. All the effort already part of their NDCs are not accounted for. Conversely, countries with BaU NDCs are favored. This is also deeply unfair. Again this work brings interesting insights on what additional efforts could be beneficial to countries in the current context, but certainly not be presented as a way forward.

Therefore, I restate my previous comment:

"The win-win strategy allocates effort to mitigate emissions on the basis of the different exposures of countries to climate impacts. So the most vulnerable countries, often the poorest with limited adaptation capacity, will have the greatest burden to mitigate. It is deeply unfair and violates the Paris Agreement principle of equity (CBDR-RC principle and article 2), the recognition of the special vulnerability of countries and their priority to receive support (when the win-win is giving them more burden). "

Reviewer #2 (Remarks to the Author):

Thank you very much for giving me the change to go through the revised version of the manuscript.

I can see that the inconsistency with the equity principles of the Paris Agreement was both highlighted by all reviewers and, to some extent, addressed and clarified.

Based on the collective feedback, the authors have significantly improved their manuscript. Drawing from both the responses and the revised version of the manuscript, I understand that this is more than a cost-benefit analysis resulting from the neoclassical setting of economic benefits of action outperforming economic damages of inaction translating from vulnerability, as was the case in the first version of the submitted manuscript.

In particular, the employed (and now described) equity approach considers historical responsibility, ability to pay (or capability), equal per capita allocation and grandfathering.

The weights are also included as Supplementary data.

However, I cannot help but ask the following:

- How were these criteria selected? Most make sense, but aren't historical responsibility and grandfathering contradicting, in the sense that the former refers to past emissions weakening the claim for future emissions, while the latter to past emissions strengthening the claim for future emissions.
- In fact, and not to take sides, how is grandfathering in line with equity, if not suggesting no structural changes to the globe, in terms of national economies?
- Where did you find these data? Or, how did you calculate these social weights?
- To me, it seems that your (originally-described) costs-benefits/vulnerability approach (not framed in terms of equity) was not designed to allow for claiming an equitable solution, but was amended as such. In other words, this is a corrective description of the approach, which makes this significantly improved, but then again leads to the odd result of Middle Eastern, African and Latin American countries carrying relatively higher burdens.
- The justification of this cannot be the potential of rapid reduction of technological costs (which, in fact, are part of the scenario design).
- In turn, this leads to arguing for global cooperation, technological flows and climate finance, which cannot be implied by a weak result (or limitation of the method).

To make things clearer, it is my understanding that the above translates to "in order to determine how countries should cooperate, we assume an approach that is almost equitable, leads to non-major emitters bearing large burdens, but it is likely tech costs drop and maybe countries cooperate a bit more in other terms".

If anything (and if not changing the approach), the authors should clearly discuss this in the conclusions as a limitation. If I understand incorrectly and this is not the case, the reader might do so similarly, and the authors should explain this better.

Furthermore, I acknowledge that after the first review round the authors have discussed some (very rational and welcome) limitations to their research, more should be included so that the results can be traced back to the theoretical foundations and mathematical structure of the IAM used.

Finally, there are many grammar, syntax, spelling issues throughout the manuscript (including in the modifications of the revised version). Although most of them do not make reading the manuscript harder, there are some that do (by the way, "in consistent with" probably means "in consistency with" but could be read "inconsistent with", which is the opposite -- there was one instance removed after a comment, but one more included in the revisions). I would recommend that a native English speaker thoroughly read and review the manuscript prior to submission.

Again, I do believe that this research deserves publication after the suggested (little more than) minor revisions are carried out.

Response to reviewers' comments

We appreciate the reviewer for his/her insightful review. The comments and suggestions have contributed substantially to improve our paper. We have tried our best to revise the manuscript. Our point-by-point responses are as follows.

■ To Reviewer #1's comments:

1. My main point remains the same, and it remains unaddressed. You wrote: "Under this framework, the win-win strategy means that the temperature-limiting goals could be reached with a net income compared with current NDCs. The net income means the cumulative benefits from the extra avoided climate impacts should exceed the extra mitigation costs compared to situation following current NDCs at both the global and national levels." Again, this cannot be as an acceptable solution. Allocating effort based on exposure to climate contradicts the Paris Agreement. This is not stated in the text despite my numerous mentions of this problem. Again, this article could reframe the analysis, keeping the results, to make it informative. It would be much more useful than shaping it as a deeply unfair suggestion for action. An interesting analysis could be framed to show how some countries would lose more welfare by inaction, compared to 1.5 °C/2 °C commensurate action. I have suggested that in previous iterations of revisions.

RESPONSE:

Thank you very much for your valuable comments and sorry for the misunderstanding in the previous revision. Instead of introducing the win-win strategy as the updated NDC action, we reframed our story by presenting win-win strategies in order to improve current emissions reduction efforts to achieve a well below 2 °C or 1.5 °C target, while clearly showing the long-term economic loss due to insufficient action against climate warming. Following your suggestions, we show how some countries would lose by inaction or continuing the current policy efforts, compared to 1.5 or 2 °C commensurate action, and answer if countries would have net income (avoid climate damage minus abatement cost) if they achieve the 1.5 °C or well below 2 °C target. The breakeven point between mitigation costs and benefits for each country are also identified.

We have revised the whole descriptions of the manuscript especially the introduction part, and added the comparison of the net income between win-win strategy and inaction situation (see Fig. 4). Moreover, based on our analysis, some

non-major emitters bear large burdens, therefore, we recognize the special vulnerability of countries and their priority to receive technical and financial support. For doing so, the ceiling costs of win-win strategies for each country has also been highlighted. Thanks again for your valuable suggestions.

Changes in the manuscript are shown below:

Title: The title of this paper has been changed to be **Win-win strategy for approaching global warming targets in the post-Paris Agreement era.**

All revisions in the Introduction

Results: If only following the current reduction efforts (Fig.4a-b), the whole world would experience a washout of benefit, which is estimated to be as high as 126.68-616.12 trillion dollars and 264.11-610.16 trillion dollars until 2100, as compared to well below 2 °C and 1.5 °C commensurate action respectively and about 1.21-5.86 times and 2.51-5.80 times of global GDP in 2015 respectively. Therein, India and the Middle East and Africa (MAF) will have larger net income compared to their own current national GDP. More worse, if even the current NDCs cannot be achieved (the USA quit from the Paris Agreement) or if some countries are unable to implement their NDCs, the whole world would tend to lose out on more benefit, ranging between 149.78 and 791.98 trillion dollars until 2100, which is about 1.42-7.53 times of the current global GDP (2015) (Fig.4c-d).

Fig. 4/Net income of win-win strategies at the regional levels (Unit: times of the regional or national GDP in 2015). a, Net income following the current reduction efforts (policy as usual scenario) of 2 °C. b, Net income following the current reduction efforts (policy as usual scenario) of 1.5 °C. c, Net income following the business as usual (BaU) efforts of 2 °C. d, Net income following the business as usual (BaU) efforts of 1.5 °C.

2. Another issue with the current framing is to study additional efforts compared to current NDCs. This disfavor countries that provided ambitious and costly NDCs. All the effort already part of their NDCs are not accounted for. Conversely, countries with BaU NDCs are favored. This is also deeply unfair. Again this work brings interesting insights on what additional efforts could be beneficial to countries in the current context, but certainly not be presented as a way forward.

RESPONSE:

Thank you very much for your comments. All the effort already part of their NDCs have been accounted for in our model. For countries with ambitious NDCs, they suffer higher abatement cost for achieving their NDCs, but they are also closer to their emission reduction goals for achieving global temperature limiting targets, indicating less abatement cost after NDCs time period. We have included this kind of balance and tradeoffs in the model. As shown in Eq. 20 and 21, the abatement costs of each region reflecting their different levels of existing NDC efforts have already been introduced in the C³IAM/EcOp model through calculating the reduction rate of current NDCs relative to the BaU scenario.

3. Therefore, I restate my previous comment:

"The win-win strategy allocates effort to mitigate emissions on the basis of the different exposures of countries to climate impacts. So the most vulnerable countries, often the poorest with limited adaptation capacity, will have the greatest burden to mitigate. It is deeply unfair and violates the Paris Agreement principle of equity (CBDR-RC principle and article 2), the recognition of the special vulnerability of countries and their priority to receive support (when the win-win is giving them more burden). "

RESPONSE:

Thank you very much for your valuable comments and sorry for the misunderstanding in the previous revision. Instead of introducing the win-win strategy as the updated NDC action, we reframed our story by presenting win-win strategies in order to

improve current emissions reduction efforts to achieve a well below 2 °C or 1.5 °C target, while clearly showing the long-term economic loss due to insufficient action against climate warming. Following your suggestions, we show how some countries would lose by inaction or continuing the current policy efforts, compared to 1.5 or 2 °C commensurate action, and answer if countries would have net income (avoid climate damage minus abatement cost) if they achieve the 1.5 °C or well below 2 °C target. The breakeven point between mitigation costs and benefits for each country are also identified.

We have revised the whole descriptions of the manuscript especially the introduction part. Moreover, based on our analysis, some non-major emitters bear large burdens, therefore, we recognize the special vulnerability of countries and their priority to receive technical and financial support. For doing so, the ceiling costs of win-win strategies for each country has also been highlighted. Thanks again for your valuable suggestions.

■ To Reviewer #2's comments:

1. Thank you very much for giving me the change to go through the revised version of the manuscript.

I can see that the inconsistency with the equity principles of the Paris Agreement was both highlighted by all reviewers and, to some extent, addressed and clarified. Based on the collective feedback, the authors have significantly improved their manuscript. Drawing from both the responses and the revised version of the manuscript, I understand that this is more than a cost-benefit analysis resulting from the neoclassical setting of economic benefits of action outperforming economic damages of inaction translating from vulnerability, as was the case in the first version of the submitted manuscript. In particular, the employed (and now described) equity approach considers historical responsibility, ability to pay (or capability), equal per capita allocation and grandfathering. The weights are also included as Supplementary data.

RESPONSE:

Thank you very much for your support and positive comments. We have tried our best to revise the manuscript. Our point-by-point responses are as follows.

2. How were these criteria selected? Most make sense, but aren't historical responsibility and grandfathering contradicting, in the sense that the former refers to past emissions weakening the claim for future emissions, while the latter to past emissions strengthening the claim for future emissions. In fact, and not to take sides, how is grandfathering in line with equity, if not suggesting no structural changes to the globe, in terms of national economies?

RESPONSE:

Thank you very much for your comments. We selected four mainstream principles widely used for defining effort sharing in the existing literature (Rose et al., 1998; Metz, 2000; Bas et al., 2012) so as to provide a more equitable weight to allocate the global emission reduction burden.

The reason we included both grandfathering and historical responsibility is that, both of them reflect the main appeals of different stakeholders, including developing countries and developed countries. Thus to make the study more objective without taking sides, we took into account all the four mainstream effort-sharing principles

when determining the integrated social welfare weights. Besides, grandfathering regime is a direct outcome of the sovereignty principle, which is regarded as a kind of equity principles, as described in several related literatures (Rose et al., 1998; Metz, 2000; Ruijven et al., 2012). To make it more clearly, we adjusted the related descriptions in this paper.

References:

- Rose A, Stevens B, Edmonds J, et al. International Equity and Differentiation in Global Warming Policy [J]. *Environmental and Resource Economics*, 1998.
- Metz B. International equity in climate change policy [J]. *Integrated Assessment*, 2000, 1(2):111-126.
- Bas J. van Ruijven, Matthias Weitzel, Michel G.J. den Elzen, et al. Emission allowances and mitigation costs of China and India resulting from different effort-sharing approaches [J]. *Energy Policy*, 2012, 46:116-134.

Changes in the manuscript are shown below:

Introduction: To derive a win-win strategy that can outperform the current policy efforts from the real long-term benefits, we considered responsibility (grandfathering and historical responsibility) defined by multiple entities such as developing countries and developed countries, capability (ability to pay) to assign more affluent countries with more efforts, and equality (equal per capita allocation) to ensure each region's equitable burden sharing in response to climate mitigation (see Methods). An integrated social welfare weight indicator was constructed by combining the estimated social welfare weights obtained from the aforementioned mainstream effort-sharing principles.

Methods: To obtain a win-win strategy, we introduce an effort-sharing approach into the cost-benefit analysis. The first step is to account for the current NDCs and construct a policy as usual pathway for each region. Then, we calculate the effort-sharing indicators of each region by following the four mainstream effort-sharing principles that is grandfathering, historical responsibility, ability to pay, and equal per capita allocation, and combine these indicators to define the integrated social welfare weights. The integrated social welfare weights are subsequently applied to simulate the optimal emission pathways in the cost-benefit analysis. Finally, the win-win strategies can be identified through comparing the relative benefits and costs between optimal emission pathways and policy as usual pathways.

3. Where did you find these data? Or, how did you calculate these social weights?

RESPONSE:

Thank you very much for your comments.

First of all, we calculate the weight under each principle. Specifically, for grandfathering principle, GHG emissions (a combination of CO₂, CH₄ and N₂O) of each region at the 1990 level are regarded as the baseline. For historical principle, we used cumulative GHG emissions of each region from the period 1990 to 2017. For ability to pay principle, we used each region's per capita GDP in the year 2017. For equal per capita principle, we used each region's population in the year 2017. The population, GDP, capital stock, and greenhouse gas emissions data used for effort-sharing indicators calculation and model estimation are from UN, IMF, CDIAC and EDGAR, which has been illustrated in *Data Sources*. Future population and GDP data are from SSP2 (a more middle-of-the-road development pattern of Share Socioeconomic Pathways).

After getting the weights of each principle, we calculated the average of the weights of these four different methods to generate the integrated social weight.

To make these points more clear, we added some details in the "Methods".

Changes in the manuscript are shown below:

Methods: Furthermore, we calculated the average of the weights of these four different methods. The resulting average weights were then used as social welfare weights in the simulation of optimal pathways. For the grandfathering principle, the permits are distributed equivalent to the baseline year emission, indicating that more emissions in baseline year would lead to lesser share of reduction burden. In this paper, GHG emissions, which are a combination of CO₂, CH₄, and N₂O of each region at the 1990 level are regarded as the baseline. For historical principle, the permits are distributed equivalent to the contribution of global temperature increase over a certain period of time. This principle suggests that the reductions towards an overall emissions ceiling were to be shared among countries proportional to their relative share of responsibility for climate change. We use cumulative GHG emissions of each region from the period 1990 to 2017. For the ability to pay principle, the permits are distributed equivalent to per capita GDP, indicating that richer countries should have heavier reduction burden. We use each region's per capita GDP in the year 2017. For the equal per capita principle, the distribution of permits is in proportion to population. In this regime, the more people there are, the lesser

responsibility there is to reduce emissions. We use each region's population for the year 2017.

4. To me, it seems that your (originally-described) costs-benefits/vulnerability approach (not framed in terms of equity) was not designed to allow for claiming an equitable solution, but was amended as such. In other words, this is a corrective description of the approach, which makes this significantly improved, but then again leads to the odd result of Middle Eastern, African and Latin American countries carrying relatively higher burdens. The justification of this cannot be the potential of rapid reduction of technological costs (which, in fact, are part of the scenario design). In turn, this leads to arguing for global cooperation, technological flows and climate finance, which cannot be implied by a weak result (or limitation of the method). To make things clearer, it is my understanding that the above translates to "in order to determine how countries should cooperate, we assume an approach that is almost equitable, leads to non-major emitters bearing large burdens, but it is likely tech costs drop and maybe countries cooperate a bit more in other terms". If anything (and if not changing the approach), the authors should clearly discuss this in the conclusions as a limitation. If I understand incorrectly and this is not the case, the reader might do so similarly, and the authors should explain this better. Furthermore, I acknowledge that after the first review round the authors have discussed some (very rational and welcome) limitations to their research, more should be included so that the results can be traced back to the theoretical foundations and mathematical structure of the IAM used.

RESPONSE:

Thank you very much for your comments. What you have summed up is what we want to express: rapid cost reduction in various regions, as assumed in this study (following Ref. 21 and 30), is an important prerequisite for achieving win-win. However, technological breakthroughs and rapid cost drops are not a natural process, which could largely depend on the diversified and practical cooperation between countries. In our conclusions, we added that *"... implementing such a win-win strategy in a real word requires countries to recognize the gravity of global warming and to make breakthroughs in low-carbon technologies. Financial and technical support from developed countries is necessary for relatively vulnerable countries to implement the win-win strategy. In order to determine how countries should cooperate, we assume an approach that takes into account the equitable effort sharing*

for emission reduction. However, it leads to some non-major emitters bearing larger burdens. Therefore, we should recognize the special vulnerability of countries and prioritize them to receive technical and financial support, which need further analysis on how to implement it in the practice.”

Due to the data and model limitation, it is unable for us to simulate the specific path of implementing the win-win strategies like how much money is needed, who will provide it and how to support the technology transfer. Doing so requires further analysis about capacity building, international finance and technology transfer. We have mentioned this in the manuscript. Thanks again for your valuable suggestions.

Changes in the manuscript are shown below:

***Discussions:** Most importantly, implementing such a win-win strategy in a real word requires countries to recognize the gravity of global warming and to make breakthroughs in low-carbon technologies. Financial and technical support from developed countries is necessary for relatively vulnerable countries to implement the win-win strategy. In order to determine how countries should cooperate, we assume an approach that takes into account the equitable effort sharing for emission reduction. However, it leads to some non-major emitters bearing larger burdens. Therefore, we should recognize the special vulnerability of countries and prioritize them to receive technical and financial support, which need further analysis on how to implement it in the practice.*

***Discussions:** In addition, the win-win strategy defined here is under the principle of economic benefits with the consideration of fairness for each country. Successful implementation of the win-win strategy is premised on improving the understanding of climate damages and the breakthroughs of low-carbon technologies. In addition to economic benefits, factors such as political attitudes, diplomacy policies, and environmental capacities are thought to be important determinants of climate mitigation actions of each country. This can be discussed in a future study.*

5. Finally, there are many grammar, syntax, spelling issues throughout the manuscript (including in the modifications of the revised version). Although most of them do not make reading the manuscript harder, there are some that do (by the way, "in consistent with" probably means "in consistency with" but could be read "inconsistent with", which is the opposite -- there was one instance removed after a comment, but one more included in the revisions). I would recommend that a native English speaker thoroughly read and review the manuscript prior to

submission.

RESPONSE:

Thank you very much for your comments and suggestion. We apologize for this error, and we have corrected the text as suggested. Our manuscript has been revised by professional English proofing company again. Thanks again for your advice.

6. Again, I do believe that this research deserves publication after the suggested (little more than) minor revisions are carried out.

RESPONSE:

Thank you very much for your great effort on improving our paper.

Reviewers' comments:

Reviewer #1 (Remarks to the Author):

Thanks for the revision, the text is more balance, although the claims of a 'win-win' solution is erroneous given the existence of the Paris Agreement. While it does not suggest an acceptable normative solution, this article can make a timely contribution to highlight the self-inflicted losses that countries commit to by not enhancing their NDC sufficiently. The term 'win-win' is disserving the paper as it is misleading the reader and does not reveal the more balanced and useful contribution it makes. Additionally, there needs to be a clearer description of the methods in plain language (combining equity and cost consideration is not clear enough).

Thank you for your work, please find my comments below.

Best with the revisions.

Main comment:

- Again, I cannot support the misleading term 'win-win' to describe a strategy that would be a major loss for many vulnerable countries compared to the existing universal and binding Paris Agreement. This strategy is only a win compared to inaction, but it contradicts the Paris Agreement. I made clearer suggestions on what I see could be name for this approach 'no-regret' or 'avoided loss' strategy, even though I do not think it is the reviewer's role to shape the writing. Feel free to think of another term that best depicts your approach.

- A quick description of the role of equity in the main text would be useful. How is equity working in your approach? The original model with each country (or region) contributing to mitigation effort on the basis of its exposure and vulnerability (what I understand as your 'win-win') is clear. But integrating equity then would 'disrupt' this allocation. If it is then a mix of equity and cost consideration, it becomes neither purely equitable, nor purely 'win-win'. How is it working? (see my comments on the methods section). Is mitigation effort calculated on the basis of avoided climate damages or equity? How could it be both (in my view, combining them results in the allocation doing neither of them)? Please bring out your conclusions in plain language for main emitters. In conclusion, it could be useful to highlight which key emitting countries (G7 or G20) have an economically rationale approach if they do not increase their NDC. Having such concluding sentences in plain language could help the dissemination of the paper.

- Please briefly describe the types of climate damages accounted for.

- See suggestions for graphs to display breakeven dates and amount of investments (net costs ahead of breakeven point) needed.

Page 1

Suggested title: "Enhancing national emissions pledges in light of climate vulnerability"

I do not think it is my reviewer's role to suggest title or wording, but including 'win-win' in the title in spite of all the comments duly addressed by the author is a problem. This approach is NOT more win-win than any of the equity literature. The authors have agreed with that point. The term 'win-win' is an overstatement of the research findings and, importantly, vague. A better term than win-win for this approach could be 'no-regret', 'avoided loss' or 'self-preserving' strategy.

Line 8: This first sentence is too conclusive in absence of supporting reference. Please consider either adding a reference or changing language to: "A strategy that informs on countries' potential losses due to lack of climate action may facilitate global governance." Furthermore, what is an informative strategy? For whom? And how would a strategy facilitate governance? Do you mean increase global action? The introduction could be a bit shorter and crisper before the results. I suggest reading:

https://cbs.umn.edu/sites/cbs.umn.edu/files/public/downloads/Annotated_Nature_abstract.pdf

Line 13: Again, I cannot support the term win-win strategy when it clearly represents a loss for most vulnerable countries compared to the current binding international agreement. In that sense, this sentence is also misleading by presenting benefits to the 'win-win' strategy without a point of reference (which is the absence of action, and not the Paris Agreement that all countries signed). This article and abstract seem to start from the idea that there is no agreement to 1) address climate change and 2) do it in an equitable manner. If your hypothesis is that the Paris Agreement is ineffective, please state so clearly and justify it.

Instead of using such vague catchy term, I suggest simply explaining the method you are using: 'Here, we quantify a distribution of mitigation effort whereby each country is economically better off than under current climate pledges and the associated climate impacts. This effort-sharing approach applied to a 1.5°C and 2°C global warming threshold suggests 'no-regret' emissions trajectories to inform NDC enhancement and Long-Term Strategies for 2020.'

Line 15: I am not sure how important this global result is in the abstract given than a lot of a research is available on that matter (see emissions gap report). How is this result crucial for the abstract? If it is not, consider putting it in the body of the text.

Line 17: the reference to the umbrella group is vague. Are greater NDC needed from all G20 countries? Or all except a few you could mention? Can you tell which countries has interest in increasing its NDC by the most percentage?

Line 20: "If even NDCs are difficult to be in position" does not make grammatic sense to me. Please consider revising language, in particular for the abstract. I understand that English may not be the primary language of the authors, but precise language is crucial to the value and impact of the paper in my experience.

Line 23: Again, I do not find that the term 'win-win' reflect the concept of the study.

Line 22-23: isn't that the whole purpose of the strategy and part of its design? If so, it may not be necessary to re-state it in the first paragraph.

Line 23: I do not understand the last sentence. Is that important?

Page 2:

Line 34: only one country enhanced its ndc, right? Consider stating it.

The term INDC can also be changed to NDC throughout the text.

Which IPCC special report?

Line 42: broader costs than what?

Line 43: Do you have any evidence that countries do not understand their potential losses? Please avoid such simplifying statements. Countries understand the risks, short-term economic decisions imply a range of priorities, sometimes competing with climate considerations. Even if addressing is a benefit over the century, the short term investments may be untenable for some developing countries. Even the term 'no-regret' that I suggested instead of 'win-win' is an overstatement. The manuscript needs to recognise than even a win-win strategy is not a silver bullet to solve climate negotiations.

Lie 44: grammar "if they had"

Line 46: I would argue that NDCs are not based on intuitions. Also, the term 'better' is subjective and undefined here. Better than what? In which regards?

Page 3

Line 49: language: "some studies focused on the emissions gap"

Line 64: I would not state that that the strategy of this article is needed. This seems to be an overstatement. What is identified here is a gap, or a lack, not a need.

Line 67: Again, I disagree with win-win.

Page 4

Line 73: do you have evidence that CCS and NET are progressing or working? If so please provide references. I understood that great uncertainties remain with respect to the future availability. The

use of the future tense, rather than a conditional mode does not reflect that uncertainty.

Page 5:

Line 90: It would be helpful to have a description of the climate damages that are modelled here, and how this model compares with other existing climate impact models.

Lin 92: How is equity used to drive the effort sharing? The methods are unclear in the main text. How is social welfare distributed? What is social welfare here? Avoided climate impacts? GDP? Please explain in language that the board readership of nature can understand.

What does the word 'region' refer to here? Is effort share across countries? World regions? Other?

Line 99: what does it mean that each country can choose its emissions pathways? Is that a possibility within the model?

Line 101: chaotic is not an appropriate term, please be more specific. What is the criterion? What does it mean to account for NDCs? Is the sentence referring to emissions? costs?

Line 108: I would refer to warming thresholds rather than warming targets

Page 6

Line 121: Language: "Nordhaus'" not "Nordhaus's"

Line 127: why these coefficients? Why these values? Is that supported by the literature? Is that an arbitrary choice

Line 129: same question, why choosing 15%, 30%, 40% etc..

Page 7:

Line 139: "a winwin strategy is found"

The results are hard to understand as the coefficients discussed here come across as arbitrary.

Expressing a percentage of scenarios when the range is based on unjustified criteria does not bring useful information for the reader.

Line 147: why selecting 9 scenarios?

Page 8:

Line 151: what do the letters stand for? Which scenario is which or how do damages increase with the letter?

Line 152: is that GHG emissions? CO2 only?

Figure 1: Why are the business as usual scenarios oscillating? Going up and down? That makes little physical sense.

Page 9:

Line 178: the cumulative efforts implies by the scenarios you derived? Isn't that the structuring criterion of the scenarios?

I might have asked before and forgot the answer, but why using USD PPP? That seems to me that PPP accounts for the relative differences between countries' economies at a given point, and may not be relevant for an integrated figure over time. Can you please explain briefly the choice?

Page 10

Figure 2: Please name panels individually. On the 'a' column, isn't the light green the cost of climate impacts and dark green the costs of mitigations? If so, please name it that way, it may be clearer.

Column b, please consider more different colors

Column c: this is nice and informative, but it could be easier and clearer to have a graph showing the date for break even points, and net-costs until that break-even points (that would indicate the amount of upfront investment needed and inform discussions around finance transfers across countries).

Page 11

Figure 3: why changing colors on column a.

Following my suggestion for figure 2, having the breakeven dates would enable to compare the effect of the 2°C and 1.5°C scenarios.

Page 12:

Line 233: Language: 'more worse' is not correct.

Page 14

Figure 5: panel b: can you please comment on countries/regions with negative values? Does that imply that their NDCs is overachieving their 'win-win' scenario? That would imply an irrational behaviour according to your depiction of a 'win-win' case, and this supports my comment to change this 'win-win' name.

Page 15:

Line 272: 'all countries need to tighten their policies'? Isn't that contradictory with the negative results of figure 5b mentioned just above?

Figure 6 is interesting. Can you please discuss the implications of your findings for international cooperation? When using equity and effort-sharing consideration, an international emissions trading scheme is usually hypothesise to allow for emissions transfer across countries. Here, since countries are seemingly guided by self-interest, no such scheme seems needed. In addition, the mitigation cost presented in figure 6 are very different across countries, which would make such international market ineffective. Can you please comment on that in the text?

It would be useful to have a table in the main text with for at least the top 4 emitters (maybe more):
1) 2030 emissions for countries, 2) net-zero emissions dates, 3) amount of negative emissions

Page 16

Line 295: it seems obvious that vulnerable countries have the most to gain. I do not understand that the sentence starts with "even the relatively vulnerable countries"

This paragraph seems to belong to the results section more than discussion.

Page 17: It would be useful to use the term 'investment' to mention the net costs early in the century, it may be clearer to the reader.

Line 323:

Page 18

Line 324: larger than what? Larger than developing countries? Isn't that the whole point of that redistribution?

Line 325: you may want to add a reference to <https://doi.org/10.1007/s11558-019-09370-0> to highlight that countries may not be acting early enough to avoid climate change, despite their interest to do so.

Line 334: The criterion o 15% improvement in negative emissions tech cost seems crucial to the results. Referring back to my earlier comment, please justify why this 15% value was chosen and how it aligns with the literature.

Line 336: this study is an interesting reference but is over 5 years old. Please consider using (Robiou du Pont & Meinshausen, 2018) for a recent point of comparison that provides single values (instead of ranges) for each country. Alternatively, (Pan, Elzen, Höhne, Teng, & Wang, 2017; Robiou du Pont et al., 2017) from 2017 provide recent results for each equity category, which results in a range is less precise.

Page 19

Line 347: This article is not telling what countries should do, please change the word 'should'. It informs at most.

Line 348: I do not understand the use of the past tense: 'this study contributes to...'

In conclusion, it could be useful to highlight which key emitting countries (G7 or G20) have an economically rationale approach if they do not increase their NDC. Having such concluding sentences in plain language could help the dissemination of the paper.

Methods:

Line 430: Please introduce why and how can equity be blended in the win-win approach without changing its win-win nature. Is mitigation effort across countries derived on the basis of avoided climate damages, or on the basis of equity? Or both, and then it is neither 100%.

Line 430: 'effort-sharing' of what (costs? Emissions rights) across what actors. It is not clear to the reader. I would also argue against using grandfathering that can be used to assess as a metric of inequity (Robiou du Pont et al., 2017), but not suggested as an equitable solution (Kantha et al., 2018).

Line 433: how are social welfare wights used, how do they influence allocations. Please bring some background for a wide audience to understand how equity is influencing the supposedly win-win strategy. Please bring such insights into the main text: is mitigation effort calculated on the basis of avoided climate damages or equity? How could it be both (in my view, combining them results in the allocation doing neither of them)

Reviewer #2 (Remarks to the Author):

Thank you once more for giving me the opportunity to review NCOMMS-19-28971B.

The authors have done a fairly good job responding to the points previously raised and, although I believe their manuscript cannot adequately address all of the reviewers' concerns in relation to the aforementioned equity issues (e.g. the adopted modelling approach, the justification of non-equitable results, etc.), I think they have put a lot of effort in framing this accordingly or more appropriately.

In other words, the research presented in this manuscript cannot possibly go "the reviewers' way" in all aspects but can certainly contribute to progressing or influencing thinking in this field.

Indicatively, I still believe that the authors' post-review modified/corrective approach resulted in outcomes, the justification of which cannot be the potential of rapid reduction of technological costs (which, in fact, are part of the scenario design), in turn leading to arguing for global cooperation, technological flows and climate finance; however, this concern has fruitfully led to highlighting these issues for the case of relatively vulnerable and/or developing countries.

I am still a bit concerned with the grandfathering principle, in that its selection is underpinned by literature prior to the Paris Agreement. I would urge the authors to back their argument with more recent pieces of work suggesting that grandfathering is part of the Paris Agreement equity principles.

A helpful justification in the literature was provided by Du Pont et al. (2017): "The fairness of the 'grandfathering' approach is criticized in the literature and not supported as such by any Party. However, we include it in the average because it represents one of the five IPCC equity categories, stressing national circumstances regarding current emissions levels, and is implicitly followed by many of the developed countries."

- Du Pont, Y. R., Jeffery, M. L., Gütschow, J., Rogelj, J., Christoff, P., & Meinshausen, M. (2017).

Equitable mitigation to achieve the Paris Agreement goals. *Nature Climate Change*, 7(1), 38.

However, please keep in mind that this research led to comments, e.g.:

- Kartha, S., Athanasiou, T., Caney, S., Cripps, E., Dooley, K., & Dubash, N. K. (2017). Response to Robiou du Pont et al on climate equity. *Nature Climate Change*.

Kindly see other discussions here:

- Du Pont, Y. R., & Meinshausen, M. (2018). Warming assessment of the bottom-up Paris Agreement emissions pledges. *Nature communications*, 9(1), 4810.

- Rogelj, J., & Schleussner, C. F. (2019). Unintentional unfairness when applying new greenhouse gas emissions metrics at country level. *Environmental Research Letters*, 14(11), 114039.

- Doukas, H., Nikas, A., González-Eguino, M., Arto, I., & Anger-Kraavi, A. (2018). From integrated to integrative: Delivering on the Paris Agreement. *Sustainability*, 10(7), 2299.

-Klinsky, S., Roberts, T., Huq, S., Okereke, C., Newell, P., Dauvergne, P., ... & Keck, M. (2017). Why equity is fundamental in climate change policy research. *Global Environmental Change*, 44, 170-173.

I therefore recommend that this manuscript be accepted after a minor revision, so as to better consider, discuss/justify, or frame this grandfathering issue. This does not mean that the authors should change their approach, but rather defend it despite its acknowledged weakness, as they have successfully done in this revision process so far.

Response to reviewers' comments

We appreciate the reviewer for his/her insightful review. The comments and suggestions have contributed substantially to improve our paper. We have tried our best to revise the manuscript. Our point-by-point responses are as follows.

■ To Reviewer #1's comments:

Note that the italic words in the **RESPONSE** are the descriptions directly copied from the manuscript.

Thanks for the revision, the text is more balance, although the claims of a 'win-win' solution is erroneous given the existence of the Paris Agreement. While it does not suggest an acceptable normative solution, this article can make a timely contribution to highlight the self-inflicted losses that countries commit to by not enhancing their NDC sufficiently. The term 'win-win' is disserving the paper as it is misleading the reader and does not reveal the more balanced and useful contribution it makes. Additionally, there needs to be a clearer description of the methods in plain language (combining equity and cost consideration is not clear enough).

Thank you for your work, please find my comments below.

Best with the revisions.

RESPONSE:

Thank you very much for your valuable comments for improving our paper. We have tried our best to revise the manuscript. First of all, we replaced 'win-win strategy' with 'self-preservation strategy' and revised the whole descriptions of the manuscript, and changed the title to be '**Self-preservation strategy for approaching global warming targets in the post-Paris Agreement era**'. Additionally, we have revised the description of the methods, especially the combination of equity and benefit-cost analysis. In the original cost-benefit analysis for the climate change, the objective of our model is to maximize the global social welfare, which use social welfare weight of each region to aggregate the regional and national social welfare. And the social welfare weights represent the relative importance in the utility and the relative mitigation burden of each region or country. In order to improve the equity of original cost-benefit analysis, we introduce the effort-sharing approaches to determine the social welfare weights of each region in the objective function. Based on this, the cost-benefit analysis is further conducted. Therefore, in our study, the mitigation effort across regions are derived on the basis of avoided climate damages and

abatement costs of each region by meanwhile considering the equity in each region's social welfare weight. Thanks again for your valuable advice. Our point-by-point responses are as follows.

Main comment:

1. Again, I cannot support the misleading term 'win-win' to describe a strategy that would be a major loss for many vulnerable countries compared to the existing universal and binding Paris Agreement. This strategy is only a win compared to inaction, but it contradicts the Paris Agreement. I made clearer suggestions on what I see could be name for this approach 'no-regret' or 'avoided loss' strategy, even though I do not think it is the reviewer's role to shape the writing. Feel free to think of another term that best depicts your approach.

RESPONSE:

Thank you very much for your valuable suggestions and sorry for the misunderstanding in the previous revision. **We replaced 'win-win strategy' with 'self-preservation strategy' and revised the whole descriptions of the manuscript especially the introduction part.** The concept of 'self-preservation strategy' has been described as *'The self-preservation strategy could contribute to straightforward benefits that countries would otherwise lose by inaction or insufficient action, compared to 1.5 °C or 2 °C commensurate action.'* And we changed the title of this article to be *'Self-preservation strategy for approaching global warming targets in the post-Paris Agreement era'*.

2. A quick description of the role of equity in the main text would be useful. How is equity working in your approach? The original model with each country (or region) contributing to mitigation effort on the basis of its exposure and vulnerability (what I understand as your 'win-win') is clear. But integrating equity then would 'disrupt' this allocation. If it is then a mix of equity and cost consideration, it becomes neither purely equitable, nor purely 'win-win'. How is it working? (see my comments on the methods section). Is mitigation effort calculated on the basis of avoided climate damages or equity? How could it be both (in my view, combining them results in the allocation doing neither of them)? Please bring out your conclusions in plain language for main emitters. In conclusion, it could be useful to highlight which key emitting countries (G7 or G20) have an economically rationale approach if they do not increase their NDC. Having such concluding sentences in plain language could help the dissemination of the paper.

RESPONSE:

Thank you very much for your comments. In the original cost-benefit analysis for the climate change, the objective of our model is to maximize the global social welfare, which use social welfare weight of each region to aggregate the regional and national social welfare. And the social welfare weights represent the relative importance in the utility and the relative mitigation burden of each region or country. In order to improve the equity of original cost-benefit analysis, we introduce the effort-sharing approaches to determine the social welfare weights of each region in the objective function. Based on this, the cost-benefit analysis is further conducted. Therefore, in our study, the mitigation effort across regions are derived on the basis of avoided climate damages and abatement costs of each region by meanwhile considering the equity in each region's social welfare weight. This part has been revised in the method section and we have added the brief description in the main text.

We have added a paragraph and Figure 4 to highlight the key emitting countries (G7 or G20) that have an economically rationale approach if they do not increase their NDC. But it might be only economically rationale for them in short run. Additionally, we added the discussions about key emitter's (G20) upfront investment in discussions section.

3. Please briefly describe the types of climate damages accounted for. See suggestions for graphs to display breakeven dates and amount of investments (net costs ahead of breakeven point) needed.

RESPONSE:

Thank you very much for your comments. We briefly list some examples of climate damages in the first paragraph in the introduction section: *'global temperatures are likely to reach 1.5 °C between 2030 and 2052, which would cause dramatic damage, including such as rising seas levels, intense flooding, wildfires, and drought'*.

In our model, we are using the climate damage function given by Nordhaus (2010), which can evaluate *the collective impact of many types of climate damages, such as damages to major sectors (e.g. agriculture), adverse impacts on health, non-market damages, and estimates of the potential costs of catastrophic damages*. We have added these descriptions in the method.

Regarding the breakeven dates and investments, we have added a paragraph and Figure 4 to discuss the amount of upfront investment needed and timing of break-even points for G20 economics and vulnerable countries in all self-preservation scenarios for achieving the 2 °C and 1.5 °C targets. Thanks again for your valuable advice.

Page 1

1. Suggested title: “Enhancing national emissions pledges in light of climate vulnerability”

I do not think it is my reviewer’s role to suggest title or wording, but including ‘win-win’ in the title in spite of all the comments duly addressed by the author is a problem. This approach is NOT more win-win than any of the equity literature. The authors have agreed with that point. The term ‘win-win’ is an overstatement of the research findings and, importantly, vague. A better term than win-win for this approach could be ‘no-regret’, ‘avoided loss’ or ‘self-preserving’ strategy.

RESPONSE:

Thank you very much for your valuable comments and suggestions. We have revised the whole descriptions of the manuscript especially the introduction part, and changed the title to be ‘*Self-preservation strategy for approaching global warming targets in the post-Paris Agreement era*’. Thanks again for your advice.

2. Line 8: This first sentence is too conclusive in absence of supporting reference. Please consider either adding a reference or changing language to: “A strategy that informs on countries’ potential losses due to lack of climate action may facilitate global governance.” Furthermore, what is an informative strategy? For whom? And how would a strategy facilitate governance? Do you mean increase global action? The introduction could be a bit shorter and crisper before the results. I suggest reading:

https://cbs.umn.edu/sites/cbs.umn.edu/files/public/downloads/Annotated_Nature_abstract.pdf

RESPONSE:

Thank you very much for your comments and suggestions. We have revised the first sentence in **Abstract** to be ‘*A strategy that informs on countries’ potential losses due to lack of climate action may facilitate global governance.*’ Here an informative strategy means a solution that can be provided for countries to increase the global action. Thanks again for your advice and sorry for the misunderstanding in the previous revision. Additionally, thanks for your kind remind and recommendation, we have read the guidance and tried our best to shorten some part of the introduction section.

3. Line 13: Again, I cannot support the term win-win strategy when it clearly represents a loss for most vulnerable countries compared to the current binding

international agreement. In that sense, this sentence is also misleading by presenting benefits to the ‘win-win’ strategy without a point of reference (which is the absence of action, and not the Paris Agreement that all countries signed). This article and abstract seem to start from the idea that there is no agreement to 1) address climate change and 2) do it in an equitable manner. If your hypothesis is that the Paris Agreement is ineffective, please state so clearly and justify it. Instead of using such vague catchy term, I suggest simply explaining the method you are using: ‘Here, we quantify a distribution of mitigation effort whereby each country is economically better off than under current climate pledges and the associated climate impacts. This effort-sharing approach applied to a 1.5°C and 2°C global warming threshold suggests ‘no-regret’ emissions trajectories to inform NDC enhancement and Long-Term Strategies for 2020.’

RESPONSE:

Thank you very much for your valuable comments and sorry for the misunderstanding in the previous revision. We accepted your kind advice and changed the sentence to be ‘*Here, we quantify a distribution of mitigation effort whereby each country is economically better off than under current climate pledges and the associated climate impacts. This effort-sharing optimizing approach applied to a 1.5°C and 2°C global warming threshold suggests ‘self-preservation’ emissions trajectories to inform NDC enhancement and long-term strategies.*’ Thanks again for your valuable advice.

4. Line 15: I am not sure how important this global result is in the abstract given than a lot of a research is available on that matter (see emissions gap report). How is this result crucial for the abstract? If it is not, consider putting it in the body of the text.

RESPONSE:

Thank you very much for your comments. We have removed this sentence from the abstract. Thanks again for your advice.

5. Line 17: the reference to the umbrella group is vague. Are greater NDC needed from all G20 countries? Or all except a few you could mention? Can you tell which countries has interest in increasing its NDC by the most percentage?

RESPONSE:

Thank you very much for your comments. The umbrella group countries here refer to other branches of umbrella group countries except USA, i.e., Canada, Australia, and New Zealand. We have revised the description and listed the main countries that need relative greater NDCs in the abstract. ‘*To be in line with no matter a 2 °C or 1.5 °C*

target, more contributions are needed from Japan, the USA, Russia, China, India, the EU, Canada, Australia, and New Zealand. Following the current emissions reduction efforts, the whole world would experience a washout of benefit, amounting to almost 126.68-616.12 trillion dollars until 2100 compared to 1.5 °C or well below 2 °C commensurate action.'

6. Line 20: "If even NDCs are difficult to be in position" does not make grammatic sense to me. Please consider revising language, in particular for the abstract. I understand that English may not be the primary language of the authors, but precise language is crucial to the value and impact of the paper in my experience.

RESPONSE:

Thank you very much for your comments and suggestion. We have corrected the descriptions. This sentence has been changed to '*If countries are even unable to implement their current NDCs, the whole world would lose more benefit, almost 149.78-791.98 trillion dollars until 2100.*' We have checked the language of the whole text again. Thanks again for your valuable advice.

7. Line 23: Again, I do not find that the term 'win-win' reflect the concept of the study.

RESPONSE:

Thank you very much for your comments and suggestion. We have changed the term for this approach to be 'self-preservation strategy'. Thanks again for your advice.

8. Line 22-23: isn't that the whole purpose of the strategy and part of its design? If so, it may not be necessary to re-state it in the first paragraph.

RESPONSE:

Thank you very much for your comments and suggestion. We want to emphasize the contribution of the self-preservation strategy after introducing the losses of current emissions reduction efforts or even unable to implement current NDCs compared to 1.5 °C or well below 2 °C commensurate action. Hence, we still keep this sentence.

9. Line 23: I do not understand the last sentence. Is that important?

RESPONSE:

Thank you very much for your comments and suggestion. We have deleted the last sentence in abstract.

Page 2:

10. Line 34: only one country enhanced its ndc, right? Consider stating it. The term INDC can also be changed to NDC throughout the text.

RESPONSE:

Thank you very much for your comments and suggestions. We have shortened the description of the first paragraph in Introduction section as *'To facilitate global climate governance, Paris Agreement requires the ratified parties to update their nationally determined contributions (NDCs) every five years¹. However, the recent 24th Conference of Parties in Katowice, Poland (COP24) and 25th Conference of Parties in Madrid, Spain (COP25) ended with limited progress².'* And we deleted the term INDC in Introduction section and corrected the whole text as suggested.

11. Which IPCC special report?

RESPONSE:

Thank you very much for your comments. The IPCC Special Report cited here is the Special Report: Global Warming of 1.5 °C, published in 2018. We have added detailed information about this report in the text as below *'According to the IPCC Special Report on the impacts of global warming of 1.5°C, at the current rate, global temperatures are likely to reach 1.5 °C between 2030 and 2052, which would cause dramatic damage, such as rising seas levels, intense flooding, wildfires, and drought³.'*

12. Line 42: broader costs than what?

RESPONSE:

Thank you very much for your comments. This indicates a broader cost of inaction than sufficient action. We added this information as *'Therefore, inaction to climate change will lead to substantial socio-economic losses, implying the occurrence of a broader cost than sufficient action.'*

13. Line 43: Do you have any evidence that countries do not understand their potential losses? Please avoid such simplifying statements. Countries understand the risks, short-term economic decisions imply a range of priorities, sometimes competing with climate considerations. Even if addressing is a benefit over the century, the short term investments may be untenable for some developing countries. Even the term 'no-regret' that I suggested instead of 'win-win' is an overstatement. The manuscript needs to recognise that even a win-win strategy is not a silver bullet to solve climate negotiations.

RESPONSE:

Thank you very much for your comments and suggestion. We reorganized this part as *‘In this sense, providing information for countries about their own widespread economic losses due to insufficient action against climate change and check if they had net income (avoided climate damage minus abatement cost) when they achieve the 1.5 °C or well below 2 °C target would be helpful for countries to make a self-preservation decision.’*

14. Line 44: grammar “if they had”

RESPONSE:

Thank you very much for your comments and suggestion. We have corrected the text as suggested. The sentence has been changed to be *‘if they had net income (avoided climate damage minus abatement cost)’*.

15. Line 46: I would argue that NDCs are not based on intuitions. Also, the term ‘better’ is subjective and undefined here. Better than what? In which regards?

RESPONSE:

Thank you very much for your comments. We have clarified the description and revised the sentence to be *‘Thus, a better emission-reduction strategy than current NDCs in terms of the potential net income from climate mitigation would be more informative for countries to reset their goals and update their NDCs in the post-Paris Agreement era.’*

Page 3

16. Line 49: language: “some studies focused on the emissions gap”

RESPONSE:

Thank you very much for your comments and suggestion. We apologize for this error, and we have corrected the text as suggested.

17. Line 64: I would not state that that the strategy of this article is needed. This seems to be an overstatement. What is identified here is a gap, or a lack, not a need.

RESPONSE:

Thank you very much for your comments and suggestion. We have changed this sentence to be *‘Therefore, what is lack to reach the 1.5 °C or well below 2 °C target is a beneficial strategy that can balance the long-term benefits obtained by reducing*

global warming and the short-term abatement costs for each country, and take into account the equitable effort sharing among countries.'

18. Line 67: Again, I disagree with win-win.

RESPONSE:

Thank you very much for your comments and suggestion. We have changed 'win-win' to be 'self-preservation'.

Page 4

19. Line 73: do you have evidence that CCS and NET are progressing or working? If so please provide references. I understood that great uncertainties remain with respect to the future availability. The use of the future tense, rather than a conditional mode does not reflect that uncertainty.

RESPONSE:

Thank you very much for your comments and suggestion. According to the *20 Years of Carbon Capture and Storage* report published by IEA, without CCS, the cost of achieving atmospheric concentrations in the range of 430-480 ppm CO₂-eq would be 138% higher. And this report also points that in the 20th year (2016) of operation of the Sleipner CCS Project in Norway, which has captured almost 17 million tonnes of CO₂ from an offshore natural gas production facility and permanently stored them in a sandstone formation deep under the seabed. This report could prove that CCS are progressing and working. Additionally, in the *Special Report: Global Warming of 1.5°C*

published by IPCC, both BECCS (480[0-1000] GtCO₂ in 1.5°C pathways with no or limited overshoot) and AFOLU CDR measures including afforestation and reforestation (210[10-540] GtCO₂ in 1.5°C pathways with no or limited overshoot) can play a major role. This report also point that BECCS development is still limited in 2030, but ramps up median levels of 3 (Below-1.5°C), 5 (1.5°C-low-OS) and 7 GtCO₂yr⁻¹(1.5°C-high-OS) in 2050, and 6 (Below-1.5°C), 12 (1.5°C-low-OS) and 15 GtCO₂yr⁻¹(1.5°C-high-OS) in 2100, respectively. We have added these two references in the manuscript. We added these two reports in reference and revised this part as '*In addition, if the low-carbon technologies (such as Carbon Capture and Storage, renewable energy utilization, and negative emissions technologies) could be rapidly developed, it will result in a lower cost for emission reduction, which will make*

countries more capable in mitigating climate change^{3, 28}.

Changes in the manuscript are shown below:

Reference:

3. *IPCC Special Report: Global Warming of 1.5°C (Cambridge Univ. Press, 2018)*
28. *International Education Association (IEA). 20 years of carbon capture and storage. <https://webstore.iea.org/20-years-of-carbon-capture-and-storage> (2016).*

Page 5:

20. Line 90: It would be helpful to have a description of the climate damages that are modelled here, and how this model compares with other existing climate impact models.

RESPONSE:

Thank you very much for your comments. *'We adopt the climate damage function derived from Ref. 20, based on which the collective impact of many types of climate damages are included, such as damages to major sectors such as (e.g. agriculture), adverse impacts on health, non-market damages, and estimates of the potential costs of catastrophic damages. We further compare the difference of the degrees of climate damage with the results of other existing climate impact models to define the uncertainty of climate damage.'* We have added this information in the Method section of manuscript.

21. Line 92: How is equity used to drive the effort sharing? The methods are unclear in the main text. How is social welfare distributed? What is social welfare here? Avoided climate impacts? GDP? Please explain in language that the board readership of nature can understand.

RESPONSE:

Thank you very much for your comments. In the original cost-benefit analysis, the objective of our model is to maximize the global social welfare, which use social welfare weights of each region to aggregate regional social welfare. And the social welfare weights represent the relative importance in the utility and the relative mitigation burden of each region or country. In order to improve the equity of original cost-benefit analysis, we introduce the effort-sharing approaches to change the social welfare weights of each region in the objective function. After this combination, the mitigation effort across regions will be derived on the basis of equity as well as avoided climate damages and abatement costs of each region. We have revised the description in the main text and Methods Section. Thanks again for your valuable

advice. As for social welfare, it's a discounting summation of utility, which is a function of consumption and is neither avoided climate impacts nor GDP. And we maximize social welfare in our model, instead of distributing social welfare. We have clarified this in the manuscript and in the Methods section.

Changes in the manuscript are shown below:

To take into account the equity between countries or regions when simulating the self-preservation strategy, we introduce the effort-sharing approaches to determine the social welfare weights that can represent the relative importance in the utility and the relative mitigation burden of each region or country. An integrated social welfare weight indicator is constructed for each region by combining the estimated social welfare weights obtained from the existing mainstream effort-sharing principles, including responsibility (grandfathering and historical responsibility) defined by multiple entities such as developing countries and developed countries, capability (ability to pay) to assign more affluent countries with more efforts, and equality (equal per capita allocation) to ensure each region's equitable burden sharing in response to climate mitigation (see Methods). And then the integrated social welfare weight is used in the global welfare maximization function to improve the equity of allocation results in the cost-benefit analysis. The optimal emission pathways for each region will then be determined under its given integrated social welfare weight and its own climate damage and abatement cost functions through the C³IAM (see Methods for detailed process).

22. What does the word 'region' refer to here? Is effort share across countries? World regions? Other?

RESPONSE:

Thank you very much for your comments. In this model, we first consider the effort-sharing and cost-benefit analysis at regional level, i.e. USA (the United States), CHN (China), JPN (Japan), IND (India), EU (the European Union), Asia (Asia excluding China, India and Japan), RUS (Russia Federation), MAF (the Middle East and Africa), EES (Eastern European and Commonwealth of Independent States countries except the Russian Federation), LAM (Latin America), OBU (other branches of umbrella group, i.e., Canada, Australia, and New Zealand) and OWE (other developed countries in Western Europe). And then the allocation results and net income are downscaled to country level. We added this information in the main text as 'In C³IAM, we first implement the effort-sharing and cost-benefit analysis at regional level (in total 12 regions); and then the allocation results and net income are further downscaled to the country level.'

23. Line 99: what does it mean that each country can choose its emissions pathways?
Is that a possibility within the model?

RESPONSE:

Thank you very much for your comments. Sorry for the misunderstandings. Actually, the model will optimize the emission pathways for each region. We have revised the description as *'The optimal emission pathways for each region will then be determined under its given integrated social welfare weight and its own climate damage and abatement cost functions through the C³IAM (see Methods for detailed process)'*.

24. Line 101: chaotic is not an appropriate term, please be more specific. What is the criterion? What does it mean to account for NDCs? Is the sentence referring to emissions? costs?

RESPONSE:

Thank you very much for your comments. We replaced the term 'chaotic' with 'ambiguous'. This sentence refer to emissions. Following NDCs documents and Ref. 30, the current submissions were taken very literally. Virtually every aspect of the submitted NDCs was decided nationally, and little to no guidance or requirements were given that could clarify their scope or enable comparability and quantifications of the pledged actions. More than 70% of the ratified parties choose business as usual (BaU) scenarios as the emissions reduction reference. Few parties (only 48) have indicated their methodology for quantifying the BaU scenario, with none providing the data source. Even worse, some parties have proposed only mitigation and adaptation actions, which makes it difficult to precisely determine their future emissions. This led published estimates of the overall emissions implications of current NDCs until 2030 to vary widely. Therefore, to overcome the difficulties of accounting for emissions in the BaU scenario and to draw the NDC path for each country, we developed a Carbon Emission Extended Principle based on Structure (CEEP-S) method. First, the CEEP-S provides a transparent projection of future emissions in the BaU scenario by considering uncertain economic development (GDP) and dynamic emission intensity (GHG emissions per unit GDP); then, the NDCs are further quantified based on the BaU emissions. The detailed implication of this criterion are shown in the section of "Method for accounting current NDCs and constructing policy as usual scenario".

25. Line 108: I would refer to warming thresholds rather than warming targets

RESPONSE:

Thank you very much for your comments and suggestion. We have changed the sentence to be '*including warming thresholds, low-carbon technology costs, climate damage and equity principles*'. Thanks again for your valuable advice.

Page 6

26. Line 121: Language: "Nordhaus'" not "Nordhaus's"

RESPONSE:

Thank you very much for your comments. We apologize for this error, and we have corrected the text as suggested.

27. Line 127: why these coefficients? Why these values? Is that supported by the literature? Is that an arbitrary choice

RESPONSE:

Thank you very much for your comments. Following Nordhaus (2010), we define the level with climate damage to be 1.6% of the global GDP at a 2.62 °C warming in 2100 as the reference level of climate damage (i.e., 1). The increase in climate damage (times) means the times of climate damage coefficients used in the damage function compared to the reference level of climate damage for the given temperature rise. And the maximum increase level was following Burke (2015). And then we divided equally to obtain other level of climate damage under the uniform distribution assumption. The values for defining the high, medium and low level of climate damage are set by ourselves with the consideration of the identified self-preservation scenarios (SP scenarios). The corresponding climate damage could be found in Supplementary Fig. 1.

28. Line 129: same question, why choosing 15%, 30%, 40% etc..

RESPONSE:

Thank you very much for your comments. 0-40% means the decline rate of low-carbon technology cost every five years, and the reference level (i.e., 0) means the decline rate of low-carbon technology cost keeps constant as the base year 2015 according to Nordhaus (2017). And 40% was following NREL (2017). And then we divided equally to obtain other level of low carbon technology cost under the uniform distribution assumption. The values for defining slow, medium and rapid development

of low-carbon technology are set by ourselves with the consideration of the identified self-preservation scenarios (SP scenarios).

Page 7:

29. Line 139: “a winwin strategy is found”. The results are hard to understand as the coefficients discussed here come across as arbitrary. Expressing a percentage of scenarios when the range is based on unjustified criteria does not bring useful information for the reader.

RESPONSE:

Thank you very much for your comments. As mentioned before, the coefficients of uncertainty level of climate damages and low carbon technology cost were set according to the existing literatures under the uniform distribution assumption. Therefore, we used the percentage of scenarios to reflect the conditions of scenarios that could realize the temperature warming targets. Though we group the scenarios according to different climate damage change levels and technology development speeds, we also indicate the exact numbers for the damage change and the decline rate of the cost. This can make the readers easier to understand the characteristics of the scenarios as well as how large is the damage change and how fast is the technology development.

30. Line 147: why selecting 9 scenarios?

RESPONSE:

Thank you very much for your comments. We selected 9 scenarios for further analysis according to the different level of low carbon technology cost and maximum social welfare criterion. We added the sentence ‘*We selected nine representative self-preservation scenarios that have the highest welfares under each level of low-carbon technology cost for further analysis.*’ in this part.

Page 8:

31. Line 151: what do the letters stand for? Which scenario is which or how do damages increase with the letter?

RESPONSE:

Thank you very much for your comments. The letters indicate 9 selected scenarios in SP 2.0s and SP 1.5s. And the level of damages decreases and the level of decline rate of low carbon technology cost increase when the letter goes from A to E. You can see Fig.1c for the characteristic of each scenario and its names.

32. Line 152: is that GHG emissions? CO₂ only? Figure 1: Why are the business as usual scenarios oscillating? Going up and down? That makes little physical sense.

RESPONSE:

Thank you very much for your comments. The emissions were GHG emissions which includes CO₂, CH₄ and N₂O. Regarding the business as usual scenario, the up and down results are because different regions peak their different types of GHG emissions (CO₂, CH₄ and N₂O) at different time.

Page 9:

33. Line 178: the cumulative efforts implies by the scenarios you derived? Isn't that the structuring criterion of the scenarios? I might have asked before and forgot the answer, but why using USD PPP? That seems to me that PPP accounts for the relative differences between countries' economies at a given point, and may not be relevant for an integrated figure over time. Can you please explain briefly the choice?

RESPONSE:

Thank you very much for your comments. We have revised the sentence to be '*Fig. 2a and Fig. 3a suggest that, as compared to the current reduction efforts, the global cumulative benefits would outweigh the additional costs before 2100*'. Regarding the PPP, it can reflect the difference among regions more correctly, especially for developing countries. Similar with GDP data of SSP database, we used PPP levels of 2011 to calculate GDP and other monetary values for every period. Therefore, USD PPP could be used for an integrated figure over time.

Page 10

34. Figure 2: Please name panels individually. On the 'a' column, isn't the light green the cost of climate impacts and dark green the costs of mitigations? If so, please name it that way, it may be clearer. Column b, please consider more different colors Column c: this is nice and informative, but it could be easier and clearer to have a graph showing the date for break even points, and net-costs until that break-even points (that would indicate the amount of upfront investment needed and inform discussions around finance transfers across countries).

RESPONSE:

Thank you very much for your comments and suggestions. We named column **a, b, c** as 'Global cumulative relative costs and benefits under 2 °C target', 'Regional cumulative relative benefits and costs in 2100 under 2 °C target' and 'National net income from 2020 to 2100 under 2 °C target' respectively. Please see the figure

caption. The short name (due to space limitation) of each column has been bold emphasized on the top of each panel. Our research regards the avoided loss due to climate change (costs of climate) as the benefits. Since the ‘a’ column is used to display the global net income which is the result of benefit and cost, so we keep our original label of this two lines. And we make a detailed explanation in the caption of Fig 2 and Fig 3: ‘The cumulative relative benefits (light green line) mean the avoided cost of climate impact. Cumulative relative costs (dark green line) mean the cost of climate mitigation.’ To make the difference between this two lines more obvious, we changed dark green line (cumulative relative costs) to be dotted line. The colors in column **b** has been changed to make the regional difference more obvious. In column **c**, we added the year of breakeven points in parentheses. Thanks again for your valuable advice.

We added a new figure (Figure 4) to indicate the amount of upfront investment needed and inform discussions around finance transfers across countries.

Changes in the manuscript are shown below:

Fig. 2|Net income between policy as usual scenario following the current reduction efforts and SP 2.0s at the global, regional and national levels. a, Global cumulative relative costs and benefits under 2 °C target. The circles indicate the turning point where the benefits exceed the costs. The cumulative relative benefits (light green line) mean the avoided cost of climate impact. Cumulative relative costs (dark green dotted line) mean the cost of climate mitigation. **b,** Regional cumulative relative benefits and costs in 2100 under 2 °C target. The black line indicates that the benefits and costs are equal. Countries or regions on the right side of the black line have positive net income. The size of the bubble refers to regions' cumulative net income in 2100. Different colors represent different regions. USA, the United States; CHN, China; JPN, Japan; IND, India; EU, the European Union; Asia, Asia excluding China, India and Japan; RUS, Russia Federation; MAF, the Middle East and Africa; EES, Eastern European and Commonwealth of Independent States countries (except the Russian Federation); LAM, Latin America; OBU, other branches of umbrella group, i.e., Canada, Australia, and New Zealand; OWE, other developed countries in Western Europe. **c,** National net income from 2020 to 2100 under 2 °C target. Unit, trillion dollars per year. The emission gap in **c** means the difference in the GHG emissions between current NDCs and self-preservation scenarios. The positive emissions gap indicates the further required GHG emissions reduction. Numbers in parentheses refer to year of break-even points.

Fig. 4|The upfront investment and timing of break-even points for G20 Economies and selected vulnerable countries following self-preservation strategy.
a. The upfront investment and timing of break-even points for G20 and selected vulnerable countries under 2.0 °C target. **b.** The upfront investment and timing of break-even points for G20 and selected vulnerable countries under 1.5 °C target. Different colors represent different countries. G20 Economies: IDN, Indonesia; KOR, the Republic of Korea; CHN, China; EU, the European Union; IND, India; JPN, Japan; ARG, Argentina; BRA, Brazil; MEX, Mexico; SAU, Saudi Arabia; ZAF, South Africa; AUS, Australia; CAN, Canada; TUR, Turkey; RUS, the Russian Federation; USA, the United States. Selected vulnerable countries: COL, Colombia; VEN, Venezuela; ETH, Ethiopia. Since the other four G20 members, i.e. the United Kingdom, France, Germany and Italy belong to the EU, the related information does not display respectively.

Page 11

35. Figure 3: why changing colors on column a. Following my suggestion for figure 2, having the breakeven dates would enable to compare the effect of the 2°C and 1.5°C scenarios.

RESPONSE:

Thank you very much for your comments and suggestions. Because in the whole text, green (from light to dark) color is used to stand for the four 2.0s strategies (SP 2.0A, SP 2.0B, SP 2.0C, SP 2.0D) and purple (from light to dark) color is used to stand for the five 1.5s strategies (SP 1.5A, SP 1.5B, SP 1.5C, SP 1.5D, SP 1.5E). So in Figure 2, we use green on column **a** and in Figure 3 we changed the color to be purple, which are consistent with the other figures in the whole manuscript. Following your previous suggestion, the colors in column **b** has been changed to make the regional difference more obvious. In column **c**, we added the year of breakeven points in parentheses. Thanks again for your valuable advice.

Changes in the manuscript are shown below:

Fig. 3|Net income between policy as usual scenario following the current reduction efforts and SP 1.5s at the global, regional and national levels. a, Global cumulative relative costs and benefits under 1.5 °C target. The circles indicate the turning point where the benefits exceed the costs. The cumulative relative benefits (light purple line) mean the avoided cost of climate impact. Cumulative relative costs (dark purple dotted line) mean the cost of climate mitigation. b, Regional cumulative

relative benefits and costs in 2100 under 1.5 °C target. The black line indicates that the benefits and costs are equal. Countries or regions on the right side of the black line have positive net income. The size of the bubble refers to regions' cumulative net income in 2100. Different colors represent different regions. **c**, National net income from 2020 to 2100 under 1.5 °C target. Unit, trillion dollars per year. The emission gap in **c** means the difference in the GHG emissions between current NDCs and self-preservation scenario. The positive emissions gap indicates the further required GHG emissions reduction. Number in parentheses refers to year of break-even points.

Page 12:

36. Line 233: Language: 'more worse' is not correct.

RESPONSE:

Thank you very much for your comments. We have corrected the text to be 'what's worse'.

Page 14

37. Figure 5: panel b: can you please comment on countries/regions with negative values? Does that imply that their NDCs is overachieving their 'win-win' scenario? That would imply an irrational behaviour according to your depiction of a 'win-win' case, and this supports my comment to change this 'win-win' name.

RESPONSE:

Thank you very much for your comments. Panel b is only for 2030. As for some regions in some scenarios, they do not need to cut more GHG emissions in 2030, but they need to cut more emissions in the future. No matter the values was negative or positive, their NDCs was not enough to achieving 2 or 1.5 target according to our optimal emissions mitigation scenarios.

Page 15:

38. Line 272: 'all countries need to tighten their policies'? Isn't that contradictory with the negative results of figure 5b mentioned just above?

RESPONSE:

Thank you very much for your comments. They are not contradicted as they were showing different aspects. The negative values of Figure 5b are derived from comparing the optimal mitigation rate with their current NDCs. However, 'all countries need to tighten their policies' was from the time perspective, and the

marginal abatement cost of all region become larger year by year at the early stage. In order to not confuse readers, we have revised the description in the manuscript as below, *‘From the time perspective, all regions need to start by tightening their policies year by year at the early stage.’*

39. Figure 6 is interesting. Can you please discuss the implications of your findings for international cooperation? When using equity and effort-sharing consideration, an international emissions trading scheme is usually hypothesise to allow for emissions transfer across countries. Here, since countries are seemingly guided by self-interest, no such scheme seems needed. In addition, the mitigation cost presented in figure 6 are very different across countries, which would make such international market ineffective. Can you please comment on that in the text? It would be useful to have a table in the main text with for at least the top 4 emitters (maybe more): 1) 2030 emissions for countries, 2) net-zero emissions dates, 3) amount of negative emissions.

RESPONSE:

Thank you very much for your comments. Under the assumption of international emissions trading scheme, the marginal abatement cost of each region will be the same. However, our results were obtained under no international emission trading scheme and no transfers among countries. Therefore, the marginal mitigation costs presented in previous Figure 6 (now Figure 7) are different across regions, because of the different mitigation efforts of regions. The big difference in marginal abatement cost wouldn't make the international market ineffective, but further verifies the necessity of establishing international emissions trading scheme in order to reduce the total abatement cost.

We have added a table in the main text to present the 2030 GHG emissions, net-zero year and cumulative negative GHG emissions for six major emitters, i.e. China, India, the EU, the USA, Russia Federation and Japan (shown in Table 1). The explanation of this table are added in the results section as *‘The average GHG emissions of China, the USA, the EU, RUS and Japan need to become negative before mid-century under both SP 2.0 and SP 1.5 scenarios. India’s average GHG emissions need to be negative before 2065 for achieving 2 °C target, which is almost 10 years later compared with the timing for 1.5 °C target. Among these major emitters, the timing of net-zero emissions of the USA and Japan (2035-2040) is 10 years earlier than China (2045-2050) and 23 years earlier than India (2060-2065) for achieving 2 °C target. The gap of net-zero points between these countries has narrowed for 1.5 °C target (Table 1).’*

Table 1| GHG emissions in 2030, timing of net-zero emissions and cumulative negative emissions of selected countries for the SP 2.0s and SP 1.5s, average over the four SP 2.0 strategies and five SP 1.5 strategies.

Country	Strategy	GHG emissions in 2030 (GtCO ₂ -eq)	Net-zero year	Cumulative negative emissions (GtCO ₂ -eq)
China	SP 2.0s	5.62 (4.53 to 6.56)	2045-2050	-49.48 (-46.71 to -51.68)
	SP 1.5s	4.61 (4.38 to 4.77)	2040-2045	-61.85 (-58.48 to -67.09)
India	SP 2.0s	3.49 (3.26 to 3.70)	2060-2065	-22.66 (-20.96 to -25.09)
	SP 1.5s	3.26 (3.22 to 3.29)	2050-2055	-30.15 (-25.28 to -33.09)
EU	SP 2.0s	1.63 (0.93 to 2.25)	2040-2045	-26.85 (-25.25 to -28.87)
	SP 1.5s	0.97 (0.85 to 1.06)	2035-2040	-31.45 (-30.85 to -32.46)
USA	SP 2.0s	1.37 (0.28 to 2.39)	2035-2040	-50.33 (-40.52 to -56.80)
	SP 1.5s	0.37 (0.22 to 0.47)	2035	-47.79 (-42.04 to -52.62)
RUS	SP 2.0s	0.63 (0.33 to 0.92)	2040-2045	-13.38 (-14.11 to -12.75)
	SP 1.5s	0.33 (0.29 to 0.37)	2035	-15.28 (-15.97 to -14.73)
JPN	SP 2.0s	0.19 (0.01 to 0.36)	2035-2040	-6.38 (-6.89 to -5.83)
	SP 1.5s	-0.01 (-0.02 to 0.01)	2030-2035	-7.24 (-7.32 to -7.10)

Page 16

40. Line 295: it seems obvious that vulnerable countries have the most to gain. I do not understand that the sentence starts with “even the relatively vulnerable countries” This paragraph seems to belong to the results section more than discussion.

RESPONSE:

Thank you very much for your comments. Yes, the first paragraph is a brief conclusion of the main results in this study. We’d like to keep it as a summary at the end.

41. Page 17: It would be useful to use the term ‘investment’ to mention the net costs early in the century, it may be clearer to the reader. Line 323:

RESPONSE:

Thank you very much for your comments. We have added a part to discuss the financial transfer in the results section. In the discussion section, we discussed the upfront investment of selected countries that are relatively vulnerable. The description has been revised as *‘They need capital and technology transfer from developed countries, which is consistent with Article 11 of the Paris Agreement. Relative vulnerable countries, for instance, Algeria and Colombia need 2.48-13.02 and 104.56-797.57 billion dollars of upfront investment for approaching the global warming targets respectively, and turn into profit in 2030-2035 and 2060-2075.’*

Page 18

42. Line 324: larger than what? Larger than developing countries? Isn't that the whole point of that redistribution?

RESPONSE:

Thank you very much for your comments. In our study, *‘some non-major emitters bearing large burdens and some countries may not be acting early enough to avoid climate change, despite their interest to do so’*, which need further analysis on how to redistribute and implement our study in the practice. Sorry for the misunderstanding in previous version. We have revised this part in the main text.

43. Line 325: you may want to add a reference to <https://doi.org/10.1007/s11558-019-09370-0> to highlight that countries may not be acting early enough to avoid climate change, despite their interest to do so.

RESPONSE:

Thank you very much for your suggestions. We added this article in the text. Shown in Reference section as Ref. 34.

Changes in the manuscript are shown below:

Reference:

34. Emmerling, J., Kornek, U., Bosetti, V., Lessmann, K. *Climate thresholds and heterogeneous regions: Implications for coalition formation. The Review of International Organizations, published online, doi: 10.1007/s11558-019-09370-0. (2020).*

44. Line 334: The criterion of 15% improvement in negative emissions tech cost seems crucial to the results. Referring back to my earlier comment, please justify why this 15% value was chosen and how it aligns with the literature.

RESPONSE:

Thank you very much for your comments. As we have answered before, 0-40% means the decline rate of low-carbon technology cost every five years, and the reference level (i.e., 0) means the decline rate of low-carbon technology cost keeps constant as the base year 2015 according to Nordhaus (2017). And 40% was following NREL (2017). And then we divided equally to obtain other levels of low carbon technology cost under the uniform distribution assumption to further reflect the uncertainty of low carbon technology development.

45. Line 336: this study is an interesting reference but is over 5 years old. Please consider using (Robiou du Pont & Meinshausen, 2018) for a recent point of comparison that provides single values (instead of ranges) for each country. Alternatively, (Pan, Elzen, Höhne, Teng, & Wang, 2017; Robiou du Pont et al., 2017) from 2017 provide recent results for each equity category, which results in a range is less precise.

RESPONSE:

Thank you very much for your comments and suggestions. We use Robiou du Pont et al. (2017) for a recent point of comparison and reorganized this part as *‘Compared to the existing effort-sharing studies, such as Ref. 8, which collected over 70 studies that analyzed future GHG emissions allowances for different regions based on a wide range of equity allocation approaches, our results are more stringent. For example, in Ref. 8, the enhanced strategy of the USA in terms of its GHG emissions reduction in 2030 is on average 44% and 64% compared with 2010 level for 2°C and 1.5°C targets, respectively, which are less stringent than the result of our study (79% and 94%, respectively). The reason for such difference may be because (1) the deterministic warming targets, i.e. 2 °C and 1.5 °C in 2100 applied in this study are more stringent than the targets of Ref.8 with a likelihood; and (2) our results are economically optimum for each involved region rather than at the global scale. When combining different effort-sharing approaches and cost-benefit analysis, countries could benefit from avoiding potential climate impacts through more stringent mitigation efforts. Compared to equitable allocations, our self-preservation strategies suggest a real ‘costly effort’ that a country could put in and point out the net income a country could stand to gain.’*

Page 19

46. Line 347: This article is not telling what countries should do, please change the word ‘should’. It informs at most.

RESPONSE:

Thank you very much for your suggestions. We have changed “should” to “could”.

47. Line 348: I do not understand the use of the past tense: ‘this study contributes to...’In conclusion, it could be useful to highlight which key emitting countries (G7 or G20) have an economically rationale approach if they do not increase their NDC. Having such concluding sentences in plain language could help the dissemination of the paper.

RESPONSE:

Thank you very much for your suggestions. We changed this sentence to ‘*Although this study contributes to displaying the real economic benefits for each country and has provided some insights for countries to reform their actions and update the NDCs in the post-Paris Agreement era, there are still a few limitations.*’

Following your great suggestion, we also added ‘*Our analysis indicates that, the upfront investment before break-even points of G20 Economies is approximately 16.38 to 103.53 trillion dollars for achieving the temperature limiting targets. In particular, the USA has to invest 5.41-33.27 trillion dollars. For Canada and Australia, the upfront investment is also relatively higher than other G20 Economies. And the break-even points for the USA, Canada and Australia will occur in the end of this century. This is a severe obstacle in implementing the proposed self-preservation strategies in the real world.*’ in the second paragraph in discussion section. Thanks again for your valuable advice.

Methods:

48. Line 430: Please introduce why and how can equity be blended in the win-win approach without changing its win-win nature. Is mitigation effort across countries derived on the basis of avoided climate damages, or on the basis of equity? Or both, and then it is neither 100%.

RESPONSE:

Thank you very much for your comments. In the original cost-benefit analysis for the climate change, the objective of our model is to maximize the global social welfare, which use social welfare weight of each region to aggregate the regional and national social welfare. And the social welfare weights represent the relative importance in the utility and the relative mitigation burden of each region or country. In order to improve the equity of original cost-benefit analysis, we introduce the effort-sharing approaches to determine the social welfare weights of each region in the objective

function. Based on this, the cost-benefit analysis is further conducted. Therefore, in our study, the mitigation effort across regions are derived on the basis of avoided climate damages and abatement costs of each region by meanwhile considering the equity in each region's social welfare weight. We have revised the description in the main text and Methods Section. Thanks again for your valuable advice.

49. Line 430: 'effort-sharing' of what (costs? Emissions rights) across what actors. It is not clear to the reader. I would also argue against using grandfathering that can be used to assess as a metric of inequity (Robiou du Pont et al., 2017), but not suggested as an equitable solution (Kantha et al., 2018).

RESPONSE:

Thank you very much for your comments. In our study, it is effort-sharing of social welfares of each region. We used different effort-sharing approaches to calculate the average social welfare weight for each region and include it as the weight of each region's welfare in the global welfare maximization function (Eq.(1))

'Noted that although the fairness of the 'grandfathering' approach is criticized in the literature and not supported as such by any Party. Following Ref. 7, we choose to include it in the average because it represents one of the five IPCC equity categories, stressing national circumstances regarding current emissions levels, and is implicitly followed by many of the developed countries (Du Pont et al. (2017))'. Also, as mentioned in Du Pont et al.(2018), the grandfathering approach, a status-quo approach that allocates equal emissions mitigation rates to all countries, is considered unfair and not openly supported by any country but implicitly matches many developed countries' targets, which they often declare as fair. Therefore, we include it in the average. We have described this in the Methods section.

50. Line 433: how are social welfare wights used, how do they influence allocations. Please bring some background for a wide audience to understand how equity is influencing the supposedly win-win strategy. Please being such insights into the main text: is mitigation effort calculated on the basis of avoided climate damages or equity? How could it be both (in my view, combining them results in the allocation doing neither of them)

RESPONSE:

Thank you very much for your comments. In the original cost-benefit analysis for the climate change, the objective of our model is to maximize the global social welfare, which use social welfare weight of each region to aggregate the regional and national

social welfare. And the social welfare weights represent the relative importance in the utility and the relative mitigation burden of each region or country.. In order to improve the equity of original cost-benefit analysis, we introduce the effort-sharing approaches to determine the social welfare weights of each region in the objective function. Based on this, the cost-benefit analysis is further conducted. Therefore, in our study, the mitigation effort across regions are derived on the basis of avoided climate damages and abatement costs of each region by meanwhile considering the equity in each region's social welfare weight. We have revised the description in the main text and Methods Section. Thanks again for your valuable advice.

Reviewer #2 (Remarks to the Author):

Note that the italic words in the RESPONSE are the descriptions directly copied from the manuscript.

Thank you once more for giving me the opportunity to review NCOMMS-19-28971B.

The authors have done a fairly good job responding to the points previously raised and, although I believe their manuscript cannot adequately address all of the reviewers' concerns in relation to the aforementioned equity issues (e.g. the adopted modelling approach, the justification of non-equitable results, etc.), I think they have put a lot of effort in framing this accordingly or more appropriately.

In other words, the research presented in this manuscript cannot possibly go "the reviewers' way" in all aspects but can certainly contribute to progressing or influencing thinking in this field.

Indicatively, I still believe that the authors' post-review modified/corrective approach resulted in outcomes, the justification of which cannot be the potential of rapid reduction of technological costs (which, in fact, are part of the scenario design), in turn leading to arguing for global cooperation, technological flows and climate finance; however, this concern has fruitfully led to highlighting these issues for the case of relatively vulnerable and/or developing countries.

RESPONSE:

Thank you very much for your support and positive comments. We have tried our best to revise the manuscript.

I am still a bit concerned with the grandfathering principle, in that its selection is underpinned by literature prior to the Paris Agreement. I would urge the authors to back their argument with more recent pieces of work suggesting that grandfathering is part of the Paris Agreement equity principles.

A helpful justification in the literature was provided by Du Pont et al. (2017): "The fairness of the 'grandfathering' approach is criticized in the literature and not supported as such by any Party. However, we include it in the average because it represents one of the five IPCC equity categories, stressing national circumstances regarding current emissions levels, and is implicitly followed by many of the developed countries."

- Du Pont, Y. R., Jeffery, M. L., Gütschow, J., Rogelj, J., Christoff, P., & Meinshausen,

M. (2017). Equitable mitigation to achieve the Paris Agreement goals. *Nature Climate Change*, 7(1), 38.

However, please keep in mind that this research led to comments, e.g.:

- Kartha, S., Athanasiou, T., Caney, S., Cripps, E., Dooley, K., & Dubash, N. K. (2017). Response to Robiou du Pont et al on climate equity. *Nature Climate Change*.

Kindly see other discussions here:

- Du Pont, Y. R., & Meinshausen, M. (2018). Warming assessment of the bottom-up Paris Agreement emissions pledges. *Nature communications*, 9(1), 4810.

- Rogelj, J., & Schleussner, C. F. (2019). Unintentional unfairness when applying new greenhouse gas emissions metrics at country level. *Environmental Research Letters*, 14(11), 114039.

- Doukas, H., Nikas, A., González-Eguino, M., Arto, I., & Anger-Kraavi, A. (2018). From integrated to integrative: Delivering on the Paris Agreement. *Sustainability*, 10(7), 2299.

-Klinsky, S., Roberts, T., Huq, S., Okereke, C., Newell, P., Dauvergne, P., ... & Keck, M. (2017). Why equity is fundamental in climate change policy research. *Global Environmental Change*, 44, 170-173.

RESPONSE:

Thank you very much for your comments and suggestions. We have added the supplement statement in the Method section: *'Noted that although the fairness of the 'grandfathering' approach is criticized in the literature³⁵⁻³⁶ and not supported as such by any Party. Following Ref. 8 and Ref. 37, we choose to include it in the average because it represents one of the five IPCC equity categories, stressing national circumstances regarding current emissions levels, and is implicitly followed by many of the developed countries.'* to support selection of grandfathering principle. The reference Du Pont et al. (2017) was already in our reference list (Ref. 7), and we reemphasized the point of view of this research. Additionally, we added Du Pont et al. (2018) in this part to support our analysis. Thanks again for your valuable advice.

I therefore recommend that this manuscript be accepted after a minor revision, so as to better consider, discuss/justify, or frame this grandfathering issue. This does not mean that the authors should change their approach, but rather defend it despite its acknowledged weakness, as they have successfully done in this revision process so far.

RESPONSE:

Thank you very much for your great effort on improving our paper.

REVIEWERS' COMMENTS:

Reviewer #1 (Remarks to the Author):

Thanks for the great improvements and the explanations.

I am only puzzled by one element: why combining vulnerability and equity to derive mitigation allocations?

I cannot make sense of such a combination whereby the regional effort is allocated on the basis of climate impacts while the allocation across countries of that region is made on the basis of equity. As a results, countries' allocations are a mix of vulnerability and equity that makes little sense to me. Could you please explain how you could interpret such results?

Otherwise, I would simply limit the results to feature the vulnerability component as seems intended by the narrative of the text (win-win/self-preserving strategy), even if that implies limiting the results' granularity to the regional level. I understand that vulnerability and climate impact data may not always be available at the national level. Having regional results, compared to regionally aggregated NDCs, is still interesting and allows for a clearer understanding than mixing equity and vulnerability. Please explain how you see these two potential options: (1) keeping the mix equity/vulnerability or (2) using only vulnerability potentially at the cost of spatial resolution.

Best

Reviewer #2 (Remarks to the Author):

Thank you for the chance to review the revised version of the manuscript.

Again, I have to recommend acceptance after minor revision, as the revision based on my previous comment (and on one of the other reviewer's comments), was simply a copy-and-paste action, right off Du Pont et al.

The authors can and should use the employed justification to their benefit but should kindly rephrase (if anything, to make more sense in terms of syntax and semantics) and reinforce it a little (I kindly refer to a list of multiple discussions/papers on equity, offered in my previous review report, but they should feel free to expand):

"However, please keep in mind that this research led to comments, e.g.: - Kartha, S., Athanasiou, T., Caney, S., Cripps, E., Dooley, K., & Dubash, N. K. (2017). Response to Robiou du Pont et al on climate equity. *Nature Climate Change*. Kindly see other discussions here: - Du Pont, Y. R., & Meinshausen, M. (2018). Warming assessment of the bottom-up Paris Agreement emissions pledges. *Nature communications*, 9(1), 4810. - Rogelj, J., & Schleussner, C. F. (2019). Unintentional unfairness when applying new greenhouse gas emissions metrics at country level. *Environmental Research Letters*, 14(11), 114039. - Doukas, H., Nikas, A., González-Eguino, M., Arto, I., & Anger-Kraavi, A. (2018). From integrated to integrative: Delivering on the Paris Agreement. *Sustainability*, 10(7), 2299. -Klinsky, S., Roberts, T., Huq, S., Okereke, C., Newell, P., Dauvergne, P., ... & Keck, M. (2017). Why equity is fundamental in climate change policy research. *Global Environmental Change*, 44, 170-173."

Response to reviewers' comments

We appreciate the reviewer for his/her insightful review. The comments and suggestions have contributed substantially to improve our paper. We have tried our best to revise the manuscript. Our point-by-point responses are as follows.

■ To Reviewer #1's comments:

Note that the italic words in the RESPONSE are the descriptions directly copied from the manuscript.

Thanks for the great improvements and the explanations. I am only puzzled by one element: why combining vulnerability and equity to derive mitigation allocations? I cannot make sense of such a combination whereby the regional effort is allocated on the basis of climate impacts while the allocation across countries of that region is made on the basis of equity. As a results, countries' allocations are a mix of vulnerability and equity that makes little sense to me. Could you please explain how you could interpret such results? Otherwise, I would simply limit the results to feature the vulnerability component as seems intended by the narrative of the text (win-win/self-preserving strategy), even if that implies limiting the results' granularity to the regional level. I understand that vulnerability and climate impact data may not always be available at the national level. Having regional results, compared to regionally aggregated NDCs, is still interesting and allows for a clearer understanding than mixing equity and vulnerability. Please explain how you see these two potential options: (1) keeping the mix equity/vulnerability or (2) using only vulnerability potentially at the cost of spatial resolution.

RESPONSE:

Thank you very much for your valuable comments for improving our paper. We have tried our best to revise the manuscript. As you understand, we first use the cost-benefit analysis by considering the regional climate damage to optimize the emission mitigation pathway for each region. The regions are basically defined based on the geographical locations. Because the cost-benefit analysis may cause unfair allocation among regions, for example, climate vulnerable regions may undertake more emissions reduction efforts which lead to the internal countries refusing to accept, therefore, we specifically adjusted the social welfare weights to improve the equity of allocation results and considered the differentiated abatement cost in different regions. In this way, the emission effort for each region would be more cost-effective for the countries inside the region. However, because the vulnerability and climate impact

data are not available at the national level and it is also difficult to conduct the cost-benefit analysis for hundreds of countries together, thus we assume the countries in the same region may facing similar climate impact in the model. On the basis of this, we further allocate the regional efforts to the countries inside the region based on the equity principle. Following previous effort-sharing studies, we apply the common but differentiated responsibilities and respective capabilities (CBDR-DC) to ensure a fair and efficient assignment of improved strategy. Indicators of responsibility, capability, and equality are used to downscale the gap for countries. The self-preservation strategy can be used to assess the regional current NDCs and apply on the regional level. However, since NDCs are submitted by each ratified parties, the benchmark on the national level to guide countries in boosting their reduction ambitions is also important. Thus, we still choose to maintain our original expression on country level results. Thanks again for your great effort on improving our paper.

■ To Reviewer #2's comments:

Note that the italic words in the RESPONSE are the descriptions directly copied from the manuscript.

Thank you for the chance to review the revised version of the manuscript. Again, I have to recommend acceptance after minor revision, as the revision based on my previous comment (and on one of the other reviewer's comments), was simply a copy-and-paste action, right off Du Pont et al. The authors can and should use the employed justification to their benefit but should kindly rephrase (if anything, to make more sense in terms of syntax and semantics) and reinforce it a little (I kindly refer to a list of multiple discussions/papers on equity, offered in my previous review report, but they should feel free to expand):

"However, please keep in mind that this research led to comments, e.g.: - Kartha, S., Athanasiou, T., Caney, S., Cripps, E., Dooley, K., & Dubash, N. K. (2017). Response to Robiou du Pont et al on climate equity. *Nature Climate Change*. Kindly see other discussions here: - Du Pont, Y. R., & Meinshausen, M. (2018). Warming assessment of the bottom-up Paris Agreement emissions pledges. *Nature communications*, 9(1), 4810. - Rogelj, J., & Schleussner, C. F. (2019). Unintentional unfairness when applying new greenhouse gas emissions metrics at country level. *Environmental Research Letters*, 14(11), 114039. - Doukas, H., Nikas, A., González-Eguino, M., Arto, I., & Anger-Kraavi, A. (2018). From integrated to integrative: Delivering on the Paris Agreement. *Sustainability*, 10(7), 2299. -Klinsky, S., Roberts, T., Huq, S., Okereke, C., Newell, P., Dauvergne, P., ... & Keck, M. (2017). Why equity is fundamental in climate change policy research. *Global Environmental Change*, 44, 170-173."

RESPONSE:

Thank you very much for your support and positive comments. And thank you for recommending useful literatures to us. We have studied them and referred to the literatures related to grandfathering approach. The descriptions in the Methods have been modified as *‘Noted that the grandfathering approach determines the national efforts relying on the current emissions and is not conducive to countries with relatively low emissions in the base year. Thus, it is often criticized in the literature³⁵⁻³⁶. However, we choose to include it in the average because it represents one of the five IPCC equity categories and is implicitly followed by many of the developed countries^{8, 37}’*. Thanks again for your great effort on improving our paper.